

# What Drives Plate Motion?

Yongfeng Yang

Bureau of Water Resources of Shandong Province, No. 127, Lishan Road, Jinan, China

*Correspondence to*: Yongfeng Yang (roufeng_yang@outlook.com)

**Abstract.** Plate motion is a remarkable Earth process that is widely ascribed to two primary driving forces: ridge push and slab pull. With the release of the first- and second-order stress fields in 1989, it was found that the observed stresses are mainly distributed on the uppermost brittle part of the lithosphere. A modeling analysis, however, reveals that the stress produced by ridge push is mainly distributed in the lower part of the lithosphere. Doglioni and Panza recently showed that slab pull was

inconsistent with the geometry and kinematics of plate. These findings suggest that other force is possibly responsible for plate motion and the observed stress. Here, we propose that the pressure of deep ocean water against the continental wall exerts enormous force (i.e., ocean-generated force) on the continent. The continent is fixed on top of the lithosphere, this attachment allows the ocean-generated force to laterally transfer to the lithospheric plate. We show that this force may combine the ridge push, collisional, and shearing forces to form force balances for the lithospheric plate; the calculated movements for the South

American, African, North American, Eurasian, Australian, and Pacific plates are well consistent with the observed movements in both speed and azimuth, the RMS of the calculated speed against the observed speed for these plates is 0.91, 3.76, 2.77, 2.31, 7.43, and 1.95 mm/yr, respectively.

## 1 Introduction

One of the most significant achievements in the 20[th] century was the establishment of plate tectonics, which developed from

the previous concept of continental drift (Wegener, 1915 and 1924). Plate tectonics mainly describes the motion of a dozen different-sized plates that connect with each other to form a giant "jigsaw puzzle" over the Earth's surface. The evidence supporting this motion includes shape fitting of the African and American continents, a coal belt crossing from North America to Eurasia, identical directions of ice sheet movement in southern Africa and India, and Global Positioning System (GPS) speed measurements. In addition, paleomagnetic reversals in oceans (Hess, 1962; Vine and Matthews, 1963) reflect seafloor

spreading, and studies of the Hawaii-Emperor seamount chain have shown that the chain is actually a trace of the lithosphere rapidly moving over relatively motionless hotspots (Wilson, 1963; Raymond et al., 2000), which further confirms the Earth's surface motion. During the past 50 years, the understanding of plate motion has expanded greatly. Plates were found to have been periodically dispersed and aggregated in the Mesozoic period, accompanied by 5-6 significant astronomical events (Cande and Kent, 1992; Cande et al., 1989; Ma et al., 1996; Wan, 1993; Hibsch et al., 1995). The speed and direction of plate





motion supported by paleomagnetism and deformation in the intraplate regions exhibited various styles over geological time (Wan, 2018). Global measurements of tectonic stresses reveal a strong correlation with plate motion, and the observed stresses may be used to constrain the forces that act on the plates (Zoback et al., 1989; Zoback, 1992; Bott and Kusznir, 1984; Zoback& Magee, 1991; Richards, 1992; Sperner et al., 2003; Heidbach et al., 2016; Heidbach et al., 2007; Heidbach et al., 2010; Heidbach et al., 2018).

Exploring the plate driving forces is important because it provides the first insights into the processes that yield plate tectonics. Throughout the history of plate tectonics, a large number of forces have been postulated to account for plate motion. Forces include centrifugal and tidal forces, ridge push, slab pull, basal drag, slab suction, mantle plume, geoid deformation, and the Coriolis force (Wegener, 1915; Hales, 1936; Holmes, 1931; Pekeris, 1935; Runcorn, 1962a,b; Wilson, 1963; McKenzie, 1968; McKenzie, 1969; Morgan, 1971; Morgan, 1972; Turcotte and Oxburgh, 1972; Forsyth & Uyeda, 1975; Oxburgh and

Turcotte, 1978; Spence, 1987; White & McKenzie, 1989; Richards, 1992; Vigny et al., 1992; Bott, 1993; Tanimoto& Lay, 2000; Conrad & Lithgow-Bertelloni, 2002; Turcotte and Schubert, 2014). Slab pull is derived from a cold and dense sinking plate that uses its weight to pull the remaining plate to which it is attached. Ridge push is usually treated either as a boundary force or a body force. As a boundary force, ridge push is derived from a "gravity wedging" effect of a warm, buoyant mantle upwelling beneath the ridge crest and acts at the edge of the lithospheric plate. In contrast, as a body force, ridge push is derived

from the horizontal pressure gradient of the cooling and thickening of the oceanic lithosphere and acts over the area of the oceanic portion of a given plate. As these two forces act on the edges of plates, they are often termed boundary forces. Basal drag (i.e., basal shear traction) is thought to be caused by the viscous moving asthenosphere along the bottom of the lithosphere, the moving asthenosphere originates from the mantle convection. Mantle plume represents the rising hot mantle flow that originates from the core-mantle boundary (Morgan, 1971; Morgan, 1972; Wilson, 1963).

Early studies on deformation modeling and torque balance analysis tended to agree that ridge push and slab pull are important for plate motion, whereas basal drag provides resistance instead of a driving force (Forsyth & Uyeda, 1975; Solomon et al., 1975; Chapple and Tullis, 1977; Richardson et al., 1979; Wortel and Cloetingh, 1981; Cloetingh and Wortel, 1986; Richardson and Cox, 1984; Richardson and Reding, 1991; Stefanick and Jurdy, 1992). Subsequent studies with complicated physical models yielded an improved understanding: buoyancy anomalies within the lithosphere, crust, and mantle act as the principal

drivers, whereas viscous dissipation within the lithosphere and at its base and shear along thrust faults at collision zones resist plate motion (Conrad and Hager, 1999;Conrad and Lithgow-Bertelloni, 2002; Stadler et al., 2010; Lithgow-Bertelloni and Richards, 1995; Becker and O'Connell, 2001; Zhong, 2001; Bird et al., 2008; Becker and Faccenna, 2011; Ghosh et al., 2013; Coltice et al., 2017). That is, in addition to slab pull and ridge push, the lithosphere and mantle feed plate move in some way. For example, the lithosphere's density variation forms a lateral pressure gradient by which plate motion is driven. The sinking

slab inserts into the deeper mantle, while the hot mantle flows (i.e., plumes) originating from the core-mantle boundary rise up to the top of the asthenosphere; this process of upwelling and downwelling causes the large-scale circulation of the plate and mantle (i.e., whole mantle convection). A more detailed description of whole mantle convection is discussed in previous





studies (Coltice et al., 2017; Bercovici et al., 2015). On the whole, the research from the past 40 years tends to agree that slab pull and ridge push are the primary plate driving forces, whereas the question of whether mantle plumes act as a driving force

remains controversial.

## 2 What is the problem of the primary driving forces?

### 2.1 Slab pull

Slab pull is considered as a "negative" buoyancy to drive the oceanic plate (e.g., Conrad & Lithgow-Bertelloni, 2003). However, since this force was proposed, its validity remains debated. Doglioni and Panza (2015) recently carried out an in-depth

investigation on slab pull, and some of their findings include the following:

1) As demonstrated by Cruciani et al. (2005), the slab dip is unrelated to the age of the oceanic lithosphere; consequently, the negative buoyancy that increases with age and is determined by the cooling oceanic lithosphere cannot control the slab dip.

2) It is assumed that eclogitization within a slab would increase the density of the lithosphere. Nevertheless, eclogitization is mostly distributed in the oceanic crust at depths of 6~8 km, and this transformation does not occur in the remaining lithospheric

mantle at depths of 60~80 km. It is possible that a small part of the slab density can increase, but the majority of the slab density does not change.

3) It is often asked why the lithosphere would subduct. This issue arises when an oceanic hydrated and serpentinized lithosphere is involved (Ulmer & Trommsdorff, 1995). Without being metamorphosed by the subduction process, the oceanic lithosphere would not be denser than the surrounding rocks. As pointed out by Panza et al. (2007), serpentinized LID often

occurs along transform faults and ridges; therefore, it is lighter than the asthenospheric mantle and elicits the question of how the plates can be pulled.

4) Most of the slabs are affected by down dip compression, and this influence is limited to depths below 300 km (Isacks& Molnar, 1971). Frepoli et al. (1996) showed that most slabs may appear at shallower depths; this situation requires a slab to be forced to sink rather than to naturally sink.

5) Although there is no slab for continental plates, these plates still move, for example, as shown in the movements of North America, South America, and Africa (Gripp & Gordon, 2002). Trench suction is widely used to account for these movements, but the mantle beneath both South and North America is moving eastward (Russo & Silver, 1994; Bokelmann, 2002). This trend is objective to the kinematics required by the trench suction model.

6) In the hotspot reference frame, the plate velocities seem to be inversely proportional to the low velocity zone's viscosity

and not related to both the age of the downgoing lithosphere and the length of the subduction zones. For example, the Pacific Plate is the fastest moving plate, but the viscosity of the asthenosphere beneath this plate is rather low (Pollitz et al., 1998; Gripp & Gordon, 2002).



7) The vertical velocity of plates (subduction-related uplift or subsidence along plate boundaries) is far slower than the horizontal velocity (Kreemer et al., 2002); this situation implies that the vertical motions of plates are rather passive.

Additionally, a kinematics analysis reveals that the subduction rate appears to be controlled by horizontal plate motion. For instance, along E- (or NE-) directed slabs, the convergence rate is faster than the subduction rate; therefore, the subduction cannot be the energetic source of plate motion.

8) When the plate motion relative to the underlying mantle is addressed, the slab might move out of the mantle and sink just because there is a faster upper plate overriding it (El Gabryet al., 2013).

9) The strength of the oceanic lithosphere is low (e.g., approximately $8\times10^{12}$ N m$^{-1}$) (Liu et al., 2004), which means that the oceanic lithosphere is able to resist a force that is smaller than slab pull (approximately $3.3\times10^{13}$ N m$^{-1}$) (Turcotte & Schubert, 2002). If slab pull is the primary driving force for the Pacific Plate, this argument of strength above would require a stretching for the Pacific Plate before slab pull drives this plate to move.

10) Brandmayr et al. (2011) and El Gabry et al. (2013) recently investigated geodynamics in the Mediterranean region. Their
findings of Vs and $\rho$ distributions with depth suggest that the slabs are less dense than the surrounding mantle, and no evidence is found to support slab pull.

These arguments on slab pull lead to the conclusion that this force cannot drive oceanic plate (Doglioni and Panza, 2015). This conclusion is further strengthened by Faccincani et al. (2021). These authors revealed that the lithospheric mantle density structure can be affected by variations in thermal regimes and bulk composition, and their results suggest that the lithospheric
mantle is not denser than the underlying asthenospheric mantle. A difference in density between the lithospheric mantle and the underlying asthenospheric mantle means that the oceanic plate, which consists of the lithospheric crust and mantle, is unlikely to sink, forming a "negative" buoyancy to drive plate motion.

## 2.2 Ridge push

### 2.2.1 Resultant stress

Tectonic stresses are caused by the forces that act on the plates (Middleton and Wilcock, 1996), and they in turn provide constraints on the plate driving forces. With the first release of the first- and second-order stress fields (Zoback et al., 1989; Zoback, 1992; Zoback & Magee, 1991), it became evident in the World Stress Map (WSM) that the maximum horizontal compressional stress $S_H$ in North America, South America and Europe has an orientation that is predominantly subparallel to either the relative or absolute plate motions (Richardson, 1992; Müller et al., 1992; Zoback, 1992). Due to this coupling of
stress orientations and plate motions, the first-order intraplate stress patterns are concluded, mainly by means of torque analysis, to be caused by the same forces that drive plate motion, especially ridge push, slab pull, collisional force, trench suction, and traction at the base of the lithosphere (Richardson, 1992; Zoback, 1992; Grünthal and Stromeyer, 1992; Gölke and Coblentz, 1996; Zoback and Zoback, 1991; Zoback and Burke, 1993; Zoback et al., 1989). Subsequent releases of the stress field (Heidbach et al., 2016; Sperner et al., 2003; Heidbach et al., 2010; Heidbach et al., 2007; Heidbach et al., 2018) and modeling



studies that reproduce plate tectonics (Ghosh et al., 2013; Richards, 1992; Stadler et al., 2010; Becker and O'Connell, 2001; Bird et al., 2008; Ghosh and Holt, 2012; Lithgow-Bertelloni and Guynn, 2004; Alisic et al., 2012) support this conclusion. These modeling studies examined the stress's orientation and style (i.e., compressional or extensional), but they are limited to the lithosphere's surface, i.e., the lithosphere is treated as a thin "shell" that is similar to the membrane, the related forces act at the edges of the lithospheric plates and their base to produce the stresses, the resultant stresses are projected onto the surface

of the plate, and then these stresses are compared with the observed stresses in the WSM. Consequently, an examination of the consistency between the modeled stresses and the observed stresses across the entire thickness of the lithosphere is lacking. The first release of the first- and second-order stress fields (Zoback et al., 1989; Zoback, 1992; Zoback & Magee, 1991) revealed another important feature of the tectonic stresses: the observed stresses are mainly concentrated on the uppermost brittle part of the lithosphere (which is ~ 40 km in depth), except for some portions of the continent that are dominated by high

topography. This vertical distribution of tectonic stresses is not included in these modeling studies (i.e., Ghosh and Holt, 2012; Lithgow-Bertelloni and Guynn, 2004). As mentioned above, ridge push is treated either as a boundary force or a body force. As a boundary force, it is derived from a "gravity wedging" effect of a warm, buoyant mantle upwelling and acts at the edge of the lithospheric plate. Turcotte and Schubert (2014) suggested that the ridge push force may be outlined in Figure 1(top) and written as $F_{RP} = F_1 - F_2 - F_3$, and $F_1 = F_5$, $F_2 = F_4$. Since each of $F_3$, $F_4$, and $F_5$ relates to pressure P that linearly increases with

depth, we would expect that the minimal ridge push force would appear at the uppermost part of the oceanic ridge, whereas the maximal ridge push force would appear at the lowermost part.





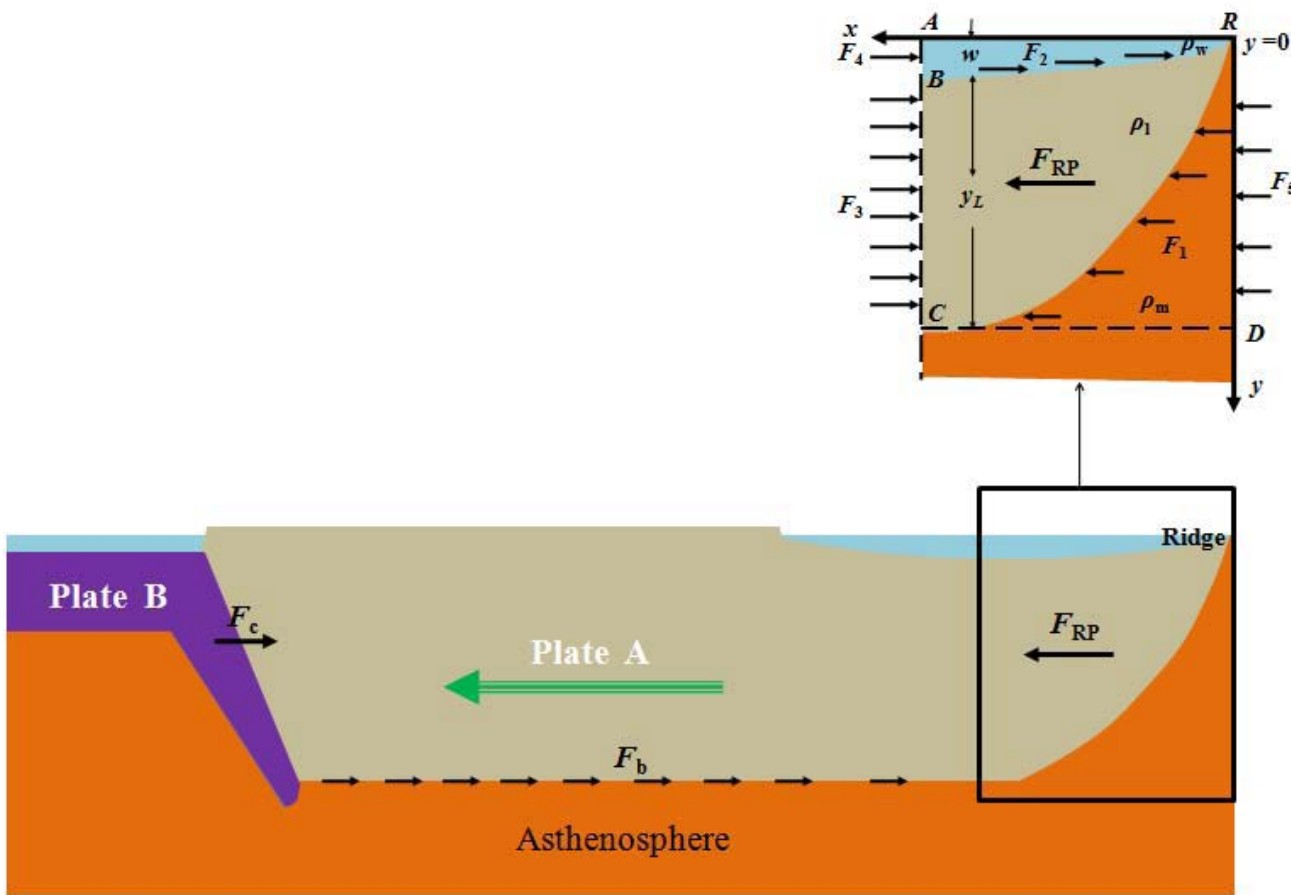

**Figure 1: Modeling the distribution of ridge push force $F_{RP}$, basal friction force $F_b$, and collisional force $F_c$ around a sample continental plate.**

The distribution of ridge push, basal friction, and collisional forces around a continental plate is outlined in Figure 1, in which trench suction is neglected. These forces represent the primary driving forces. It is assumed that Plate A moves towards the left, Plate B exerts a collisional force (i.e., $F_c$) on its left side, the oceanic ridge exerts a push force (i.e., $F_{RP}$) on its right side, and the asthenosphere exerts a friction force (i.e., $F_b$) on its base. This force distribution requires the stress produced by these forces to be distributed mostly in the lowermost part of the plate.

We developed a simplified model to verify this expectation (Figure 2 (top)). The model is made of rocks materials and is straight, meaning that the Earth's curvature is not considered. The model is assumed to be homogeneous and isotropic, and its thickness and length are 100 km and 290 km, respectively. Along the horizontal direction, it receives a collisional force $F_c$, and this force is resistive and uniformly exerted on its left side; the oceanic ridge exerts a push force $F_{RP}$ on its right side, this force is driving and increases with depth; the mantle exerts a frictional force $F_b$ on its base, and this force is resistive. These





forces realize a horizontal force balance. Along the vertical direction, it is supported by the mantle, and its gravity is balanced
out by the supporting from the mantle.

Finite element analysis software (i.e., Abaqus) is used to resolve the stress caused by these forces. The model's bottom is given
a remote boundary condition. As the upper part of the lithospheric plate is elastic and brittle, whereas the lower part is plastic
and ductile, this requires to assume that the physical property of the model is vertically transited from elasticity to plasticity.

The inputs include the vertical pressure caused by the rock's weight and the lateral pressures caused by the loads (i.e., $F_{RP}$, $F_b$,
and $F_c$). The outputs include the stress produced by the vertical pressure alone and the stress produced by a combination of the
vertical and lateral pressures. The two-dimensional frame allows to know the horizontal stress (i.e., S11) and the vertical stress
(i.e., S22).

However, in order to expedite the deduction, we will only discuss the horizontal stress (i.e., S11) in the following. The elastic

modulus, Poisson ratio, and rock density of the model are set to 100,000 MPa, 0.3, and 2,690 kg/m³, respectively. The pressure
caused by the rock's weight yields Set I data; The $F_{RP}$ is given as $4.0 \times 10^{12}$ N m⁻¹, which is generally accepted by the scientific
community (Turcotte and Schubert, 2004). It is assumed that $F_b$ and $F_c$ are 80% and 20% of $F_{RP}$, respectively. The pressures
caused by these loads yield Set II data; To test the stress variation when the resistive forces are moderately adjusted, we again
assume $F_b$ and $F_c$ to be 50% and 50% of $F_{RP}$. The pressures caused by these revised loads yield Set III data.

To gain a more accurate understanding of the resultant stress, we cut a rectangular area *GHIJ* to specifically discuss. The stress
clouds of this area are compared in Figure 2 (left). Please note that any of these loads (i.e., $F_{RP}$, $F_b$, and $F_c$) is too small relative
to the rock's weight. For example, when $F_{RP} = 4.0 \times 10^{12}$ N m⁻¹ is applied to the model's right side (which is 85 km length), its
resultant mean pressure is 47.06 MPa, while the mean lithostatic pressure of the model (which is 100 km depth) is 1318.1 MPa.
This reality means that, if we use a stress cloud to compare the stress caused by a combination of the rock's weight and the

loads with the stress caused by the rock's weight alone, both of them may be indistinguishable. To create a visual impression,
we magnify these loads 50 times, which yields Set II' data and Set III' data (see Figure 2(right)). We find that the horizontal
stress caused by these loads is compressional and mainly concentrated on the lower part of section *GHIJ*.

We use three sections (i.e., M₁N₁, M₂N₂, and M₃N₃) from that rectangular area to quantify the comparison. Each section keeps
a span of 50 km relative to one another. The stress diagrams for three sections are compared in Figure 3. When we subtract the

stress caused by the rock's weight from the stress caused by the rock's weight and these loads, we obtain the stress caused by
the loads, which is exhibited in Set II (III) - Set I. These stress diagrams agree the result exhibited in the stress clouds.

Kusznir and Bott (1977) argued that, due to the ductile nature of the lower part of the lithosphere, there would be a redistribution
of any stress applied to the whole lithosphere that would result in stress amplification in the upper brittle part of the lithosphere.
This view is based on the assumption that force is uniformly exerted on the side of the lithospheric plate, but the reality is that

the ridge push force would increase with depth; consequently, the redistribution of the stress and its amplification are not
applicable to the ridge push force. In contrast, we have considered this ductile nature in the modeling, but no evidence was



found for stress amplification in the upper part of section *GHIJ*. Our modeling analysis suggests that the stress caused by a combination of the ridge push, collisional, and basal friction forces, cannot be in accordance with the observed stress.

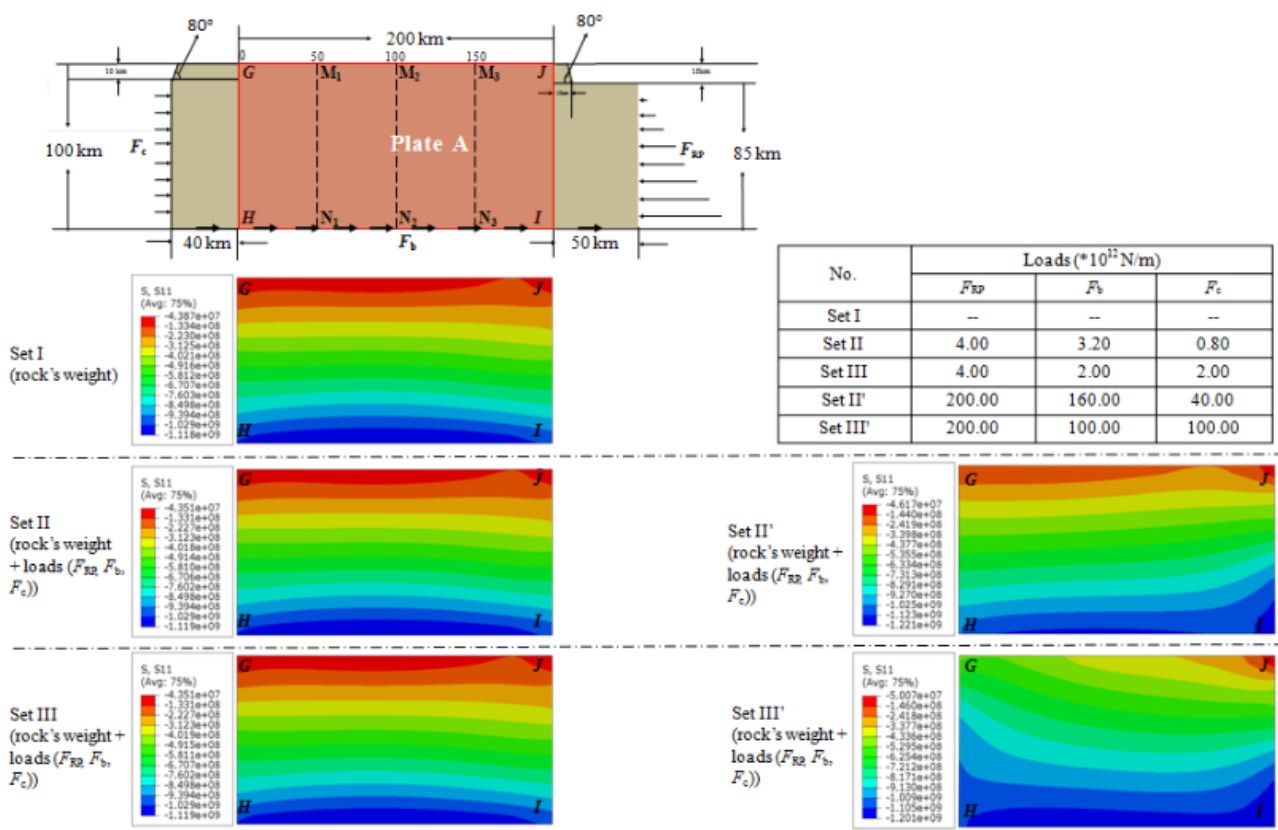

**Figure 2: Stress cloud comparison.** $F_{RP}$, $F_b$, and $F_c$ denote the ridge push force, the basal friction force, and the collisional force.



**Figure 3: Stress diagram comparison.**

## 2.2.2 Geometry and kinematics of the continental plate

In examining the plate's shape around the globe, we see that the eastern coastline of the American continent is approximately subparallel to the Atlantic ridge at the plate's boundary, and the coastline of Australia's continent is subparallel to the boundary of the Australian Plate. However, the coastline's length of the American continent is greater than that of the Atlantic ridge, whereas the coastline's length of the Australian continent is less than that of its boundary. This feature is also clear for the Indian Plate. Such fashion implies that the driving force of continental plate is likely related to the coastline.

All plates are steadily moving over the Earth's surface, this means that there would be separations and approaches between plates. Separations would result in a gap between two plates. The gap would allow magma to erupt and form a mid-ocean ridge



(MOR). In this respect, the MOR may be the result of plate motion. Currently, the MOR is treated as the cause of the plate driving force. This treatment leads to a chicken-or-egg question: which came first? In physics, the object that exerts force must be clearly differentiated from the object that accepts this force. Some argue that once subduction and spreading are initiated, plates may drive themselves as part of the large circulation of the mantle and lithosphere, which resolves the chicken-or-egg question. However, this view cannot be convincing. Ridge push force contributes to not only the oceanic plates but also the continental plates. The oceanic plates are subducting into the trenches, they take part in the large circulation. Instead, the continental plates never sink, they wouldn't take part in the large circulation, therefore, the chicken-or-egg question remains for the continental plates.

As mentioned in section 1, the latest understanding of plate dynamics is that the lithosphere, crust, and mantle compose the large-scale circulation, and that the plate itself is an integral part of this circulation. Consequently, the dynamic source of plate motion is traced back to the Earth's interior. The terrestrial planets (Venus, Mercury, and Mars) share similar formation procession and interior structure (i.e., crust, mantle, and core) with the Earth, they also have the same spatial surroundings (i.e., asteroid impact) as the Earth does. Therefore, the question remains of why there is plate motion on Earth but not on the other terrestrial planets. This discrepancy of plate motion distribution, together with the above mentioned problems of slab pull and ridge push, implies that some key factor of the Earth, which is still currently unknown, is more likely to cause plate motion.

## 3 An ocean-generating force driving mechanism for plate motion

### 3.1 Ocean-generating force

Ocean water covers approximately 71% of the Earth's surface, and its total volume is almost 1.35 billion $km^3$, with an average depth of nearly 3,700 meters. Geochemical study of zircons suggests that liquid water has existed for more than 4 Gy ago (Mojzsis et al., 2001; Bercovici et al., 2015; Valley et al., 2002). Ocean water is supported by the upper part of the lithosphere, this loading allows the weight of the ocean water to be vertically transferred to the lithosphere. The impact of ocean water on the isostatic balance and heat process of the lithosphere has been widely discussed (Bercovici et al., 2015; Fleitout and Froidevaux, 1983; Osei Tutu et al., 2018; Ricard et al., 1984; Steinberger et al., 2001; Lithgow-Bertelloni, 2014; Ghosh and Holt, 2012; Naliboff et al., 2012). The absence of plate tectonics on the other terrestrial planets (e.g., Mars and Venus) is already recognized; this absence is typically ascribed to the lack of liquid water because the water on Mars possibly appeared in liquid form in the planet's early history and water on Venus has disappeared through a runaway greenhouse (Squyres et al., 2004). Nevertheless, the mechanism by which liquid water contributes to plate tectonics remains enigmatic (Bercovici et al., 2015). A view is that the Earth's surface is cooled by liquid water since the Earth's temperature needs to be stabilized by a negative feedback in the formation of plate tectonics (Bercovici et al., 2015; Walker et al., 1981; Berner, 2004).

Liquid exerts pressure on the wall of a container that holds it. The pressure generated on the wall of a cubic container may be written as $P=\rho gy/2$, and the application of this pressure across the wall yields a force. This force may be expressed as



$F=PS=\rho g y^2 x/2$, where $S$ is the wall area, g and $\rho$ are the gravitational acceleration and liquid density, respectively, and $x$ and $y$ are the liquid width and depth, respectively, in the container. Referring to the real world, ocean basins are naturally gigantic

containers, and their depths are often more than a few kilometers and vary from one place to another. All of these factors imply that oceans can generate enormous pressure everywhere and that this pressure is unequal among oceans. Furthermore, the application of pressure against oceanic basin walls, which consist of the continents, can yield enormous unequal forces on the continents. Geometrically, ocean pressure is always exerted vertically to the continental slope, by which a normal force is formed. This normal force is called the ocean-generated force, which may be further decomposed into a horizontal force and

a vertical force. The horizontal force may be further written as $F =0.5\rho g L h^2$, where $\rho$, g, $L$, and $h$ are the density of water, gravitational acceleration, ocean width that fits the continent's width, and ocean depth, respectively.

In practice, the continent's side is not flat, and the continent's base is generally wider than its top, making the continent appear more like a circular truncated cone standing in the ocean. As the horizontal force is related to the ocean's width (i.e., the continent side's width), we need to horizontally project the continent onto a polygonal column, dissect the whole side of this

column into a series of smaller rectangular sides connecting one to another and subsequently calculate the horizontal force generated at each of these rectangular sides. Figure 4 exhibits the horizontal forces generated on the continents. The horizontal forces and the parameters for calculating them are listed in Table 1. Figure 4 exhibits the horizontal forces generated on the continents. The horizontal forces and the parameters for calculating them are listed in Table 1.

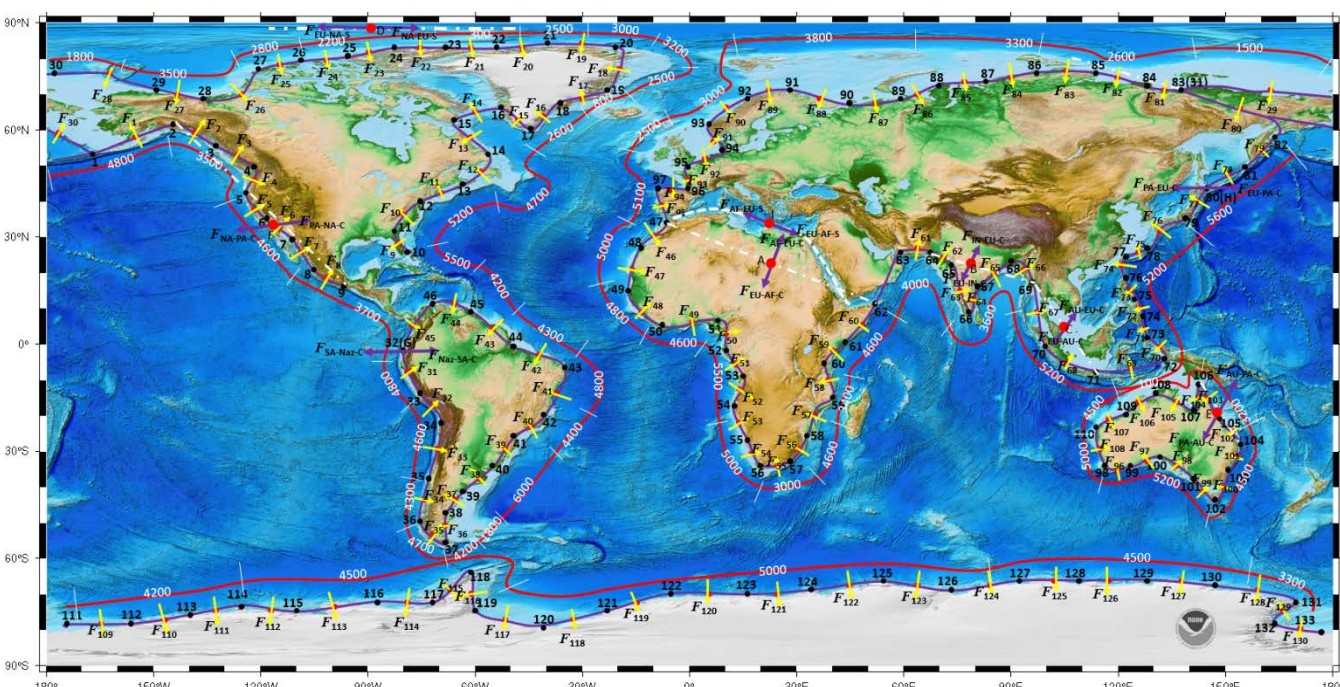

**Figure 4: Geographic treatment of the control sites on the continents and the horizontal forces generated on them.** *F* (yellow arrows) denotes the horizontal force. The red lines depict the ocean depths. The black dots denote the control sites;



the small red dots denote the geometric centers of the sides. The large red dots denote the exerting centers of the collisional and shearing forces (large purple arrows) between plates. The background map is ETOPO1 Global Relief Model (Amanteand Eakins, 2009). Note that the ocean depths were artificially resolved through the NOAA bathymetric data viewer.









**Table 1(A). Basic information for the horizontal forces**

| j | d$_j$ | q$_j$ | i | L$_i$ | α$_i$ | β$_i$ | h$_{i\text{-ocean}}$ | F$_i$ | Inclination to latitude, east (γ$_i$) |
|---|---|---|---|---|---|---|---|---|---|
| | Longitude | Latitude | | km | Longitude | Latitude | m | N (*10$^{17}$) | Degrees (°) |
| 1 | 194.09° | 54.23° | 1 | 1747.90 | 204.23° | 60.39° | 4,800 | 1.9733 | 140.74° |
| 2 | 214.37° | 66.55° | 2 | 1071.92 | 220.79° | 62.71° | 4,800 | 1.2102 | 127.52° |
| 3 | 227.21° | 58.86° | 3 | 1190.03 | 232.02° | 54.29° | 3,500 | 0.7143 | 121.53° |
| 4 | 236.82° | 49.71° | 4 | 931.50 | 236.29° | 45.54° | 3,500 | 0.5591 | 354.92° |
| 5 | 235.76° | 41.36° | 5 | 921.16 | 238.93° | 38.05° | 4,600 | 0.9551 | 36.94° |
| 6 | 242.09° | 34.73° | 6 | 1319.66 | 246.74° | 30.35° | 4,600 | 1.3683 | 42.46° |
| 7 | 251.39° | 25.96° | 7 | 662.22 | 253.02° | 23.38° | 4,000 | 0.5192 | 30.03° |
| 8 | 254.64° | 20.80° | 8 | 1092.31 | 259.13° | 18.34° | 3,700 | 0.7327 | 59.98° |
| 9 | 263.61° | 15.88° | | | | | | | |
| 10 | 280.94° | 25.22° | 9 | 700.05 | 279.36° | 28.05° | 5,500 | 1.0377 | 206.26° |
| 11 | 277.78° | 30.87° | 10 | 1125.90 | 281.21° | 35.10° | 5,200 | 1.4918 | 146.46° |
| 12 | 284.63° | 39.32° | 11 | 1126.57 | 290.39° | 42.06° | 5,200 | 1.4927 | 122.57° |
| 13 | 296.15° | 44.79° | 12 | 1032.68 | 300.23° | 48.58° | 5,200 | 1.3683 | 144.54° |
| 14 | 304.31° | 52.37° | 13 | 1254.00 | 299.14° | 57.29° | 3,000 | 0.5530 | 209.60° |
| 15 | 293.97° | 62.21° | 14 | 701.87 | 300.48° | 63.47° | 4,700 | 0.7597 | 113.29° |
| 16 | 306.98° | 64.72° | 15 | 708.93 | 311.51° | 62.32° | 3,000 | 0.3126 | 41.23° |
| 17 | 316.04° | 59.91° | 16 | 870.41 | 320.23° | 63.36° | 2,600 | 0.2883 | 151.37° |
| 18 | 324.42° | 66.80° | 17 | 662.56 | 330.75° | 68.72° | 800 | 0.0208 | 129.72° |
| 19 | 337.08° | 70.63° | 18 | 1360.30 | 337.57° | 76.75° | 3,200 | 0.6825 | 178.95° |
| 20 | 338.06° | 82.87° | 19 | 275.43 | 327.87° | 83.08° | 2,800 | 0.1058 | 260.40° |
| 21 | 317.67° | 83.29° | 20 | 198.31 | 311.52° | 82.83° | 2,500 | 0.0607 | 239.16° |
| 22 | 305.37° | 82.37° | 21 | 231.66 | 297.64° | 82.75° | 300 | 0.0010 | 249.11° |
| 23 | 289.90° | 83.12° | 22 | 238.55 | 281.92° | 82.74° | 2,200 | 0.0566 | 249.47° |
| 24 | 273.94° | 82.36° | 23 | 336.45 | 267.96° | 81.15° | 2,200 | 0.0798 | 322.66° |
| 25 | 261.98° | 79.94° | 24 | 294.67 | 255.30° | 79.43° | 2,200 | 0.0699 | 292.70° |





| 26 | 248.62° | 78.91° | 25 | 313.81 | 244.23° | 77.84° | 2,800 | 0.1206 | 319.08° |
| 27 | 239.83° | 76.77° | 26 | 997.49 | 229.64° | 73.31° | 3,500 | 0.5987 | 319.52° |
| 28 | 219.44° | 69.84° | 27 | 523.95 | 212.59° | 70.37° | 3,500 | 0.3145 | 257.09° |
| 29 | 205.73° | 70.90° | 28 | 753.32 | 195.39° | 72.17° | 3,500 | 0.4522 | 291.66° |
| 30 | 185.05° | 73.44° | 29 | 1601.37 | 171.87° | 68.03° | 1,500 | 0.1766 | 317.22° |
| 31 | 158.68° | 62.61° | 30 | 2225.18 | 176.39° | 58.42° | 50 | 0.0003 | 293.80° |

Notes: all geographic sites refer to Figure 4.


**Table 1(B). Basic information for the horizontal forces (continued)**

| | Control site | | | | Side | | | | |
| | | | | Length | Hypothetical geocentric center | | Ocean depth | Horizontal force | |
| $j$ | $d_j$ | $q_j$ | $i$ | $L_i$ | $\alpha_i$ | $\beta_i$ | $h_{i\text{-ocean}}$ | $F_i$ | Inclination to latitude, east ($\gamma_i$) |
| | Longitude | Latitude | | km | Longitude | Latitude | m | N ($*10^{17}$) | Degrees |
| 32 | 278.96° | -2.20° | 31 | 1290.88 | 281.25° | -7.55° | 4,800 | 1.4573 | 22.92° |
| 33 | 283.53° | -12.90° | 32 | 1163.08 | 286.57° | -17.26° | 4,600 | 1.2059 | 33.59° |
| 34 | 289.60° | -21.62° | 33 | 1737.42 | 287.89° | -29.30° | 4,600 | 1.8014 | 349.01° |
| 35 | 286.17° | -36.97° | 34 | 1405.78 | 284.78° | -43.22° | 4,300 | 1.2737 | 350.81° |
| 36 | 283.39° | -49.46° | 35 | 876.16 | 287.57° | -52.48° | 4,700 | 0.9484 | 40.11° |
| 37 | 291.74° | -55.50° | 36 | 977.06 | 291.92° | -51.11° | 1,800 | 0.1551 | 181.43° |
| 38 | 292.09° | -46.71° | 37 | 731.33 | 294.03° | -43.73° | 1,800 | 0.1161 | 154.87° |
| 39 | 295.96° | -40.75° | 38 | 1014.55 | 300.13° | -37.59° | 6,000 | 1.7897 | 133.75° |
| 40 | 304.29° | -34.43° | 39 | 1161.02 | 307.71° | -30.12° | 6,000 | 2.0480 | 145.55° |
| 41 | 311.13° | -25.81° | 40 | 1086.71 | 315.40° | -22.90° | 5,200 | 1.4398 | 126.58° |
| 42 | 319.66° | -19.98° | 41 | 1571.58 | 322.47° | -13.46° | 4,600 | 1.6295 | 157.31° |
| 43 | 325.28° | -6.93° | 42 | 1744.37 | 318.22° | -3.48° | 4,300 | 1.5804 | 243.89° |
| 44 | 311.15° | -0.02° | 43 | 1602.67 | 305.40° | 4.36° | 4,200 | 1.3853 | 232.65° |
| 45 | 299.64° | 8.73° | 44 | 292.31 | 300.11° | 7.50° | 4,800 | 0.3300 | 200.75° |
| 46 | 300.58° | 6.27° | 45 | 2576.75 | 289.77° | 2.04° | 3,700 | 1.7285 | 291.42° |
| 47 | 353.22° | 34.24° | 46 | 908.04 | 350.19° | 31.09° | 5,000 | 1.1123 | 320.52° |



| 48 | 347.15° | 27.93° | 47 | 1462.59 | 345.17° | 21.61° | 5,000 | 1.7917 | 343.79° |
| 49 | 343.19° | 15.29° | 48 | 1482.74 | 347.41° | 10.07° | 4,800 | 1.6739 | 38.46° |
| 50 | 351.63° | 4.84° | 49 | 1689.12 | 359.25° | 5.29° | 4,600 | 1.7514 | 93.36° |
| 51 | 6.87° | 5.73° | 50 | 898.57 | 8.32° | 1.96° | 4,600 | 0.9317 | 20.99° |
| 52 | 9.77° | -1.82° | 51 | 1051.19 | 11.94° | -6.03° | 5,500 | 1.5581 | 27.12° |
| 53 | 14.11° | -10.24° | 52 | 887.15 | 13.10° | -14.11° | 5,500 | 1.3150 | 345.81° |
| 54 | 12.09° | -17.98° | 53 | 1157.53 | 14.03° | -22.88° | 5,000 | 1.4180 | 19.99° |
| 55 | 15.96° | -27.77° | 54 | 779.24 | 17.82° | -30.90° | 5,000 | 0.9546 | 26.98° |
| 56 | 19.67° | -34.02° | 55 | 669.59 | 23.27° | -33.69° | 3,000 | 0.2953 | 83.72° |
| 57 | 26.87° | -33.36° | 56 | 1010.77 | 29.68° | -29.52° | 4,600 | 1.0480 | 147.57° |
| 58 | 32.48° | -25.68° | 57 | 1416.91 | 36.40° | -20.46° | 4,000 | 1.1109 | 144.93° |
| 59 | 40.31° | -15.24° | 58 | 1063.87 | 39.56° | -10.51° | 3,400 | 0.6026 | 188.85° |
| 60 | 38.81° | -5.78° | 59 | 880.15 | 41.14° | -2.58° | 4,600 | 0.9126 | 144.02° |
| 61 | 43.47° | 0.63° | 60 | 1479.88 | 47.38° | 6.04° | 4,600 | 1.5344 | 144.34° |
| 62 | 51.29° | 11.45° | | | | | | | |

Notes: all geographic sites refer to Figure 4.

**Table 1(C). Basic information for the horizontal forces (continued)**

| | Control site | | | Side | | | | | |
| | | | | Length | Hypothetical geocentric center | | Ocean depth | Horizontal force | |
| $j$ | $d_j$ | $q_j$ | $i$ | $L_i$ | $\alpha_i$ | $\beta_i$ | $h_{i\text{-ocean}}$ | $F_i$ | Inclination to latitude, east $(\gamma i)$ |
| | Longitude | Latitude | | km | Longitude | Latitude | m | N $(*10^{17})$ | Degrees |
|---|---|---|---|---|---|---|---|---|---|
| 69 | 95.41° | 16.55° | 67 | 1802.58 | 97.12° | 8.62° | 5,000 | 2.2082 | 11.96° |
| 70 | 98.82° | 0.68° | 68 | 1806.5 | 105.79° | -3.54° | 5,200 | 2.3935 | 60.27° |
| 71 | 112.76° | -7.75° | 69 | 2151.54 | 122.43° | -6.51° | 100 | 0.0011 | 97.36° |
| 72 | 132.09° | -5.27° | 70 | 1075.97 | 130.03° | -0.89° | 5,200 | 1.4256 | 204.11° |
| 73 | 127.96° | 3.49° | 71 | 485.05 | 127.35° | 5.59° | 5,200 | 0.6427 | 196.28° |
| 74 | 126.73° | 7.68° | 72 | 435.29 | 126.29° | 9.59° | 5,200 | 0.5767 | 167.20° |
| 75 | 125.85° | 11.5° | 73 | 947.45 | 123.79° | 15.27° | 5,200 | 1.2553 | 152.16° |





| 76 | 121.72° | 19.04° | 74 | 554.03 | 121.85° | 21.53° | 5,200 | 0.7341 | 177.22° |
| 77 | 121.98° | 24.02° | 75 | 614.22 | 124.74° | 25.22° | 5,200 | 0.8138 | 115.61 |
| 78 | 127.49° | 26.41° | 76 | 1399.63 | 132.94° | 30.63° | 5,200 | 1.8545 | 131.96° |
| 79 | 138.39° | 34.84° | 77 | 1133.3 | 141.69° | 39.26° | 5,600 | 1.7415 | 150.02° |
| 80 | 144.98° | 43.68° | 78 | 954.18 | 149.84° | 46.38° | 5,600 | 1.4662 | 128.80° |
| 81 | 154.69° | 49.07° | 79 | 1196.64 | 159.22° | 53.76° | 5,600 | 1.8388 | 150.27° |
| 82 | 163.74° | 58.44° | 80 | 1136.53 | 161.21° | 60.53° | 3,800 | 0.8042 | 266.56° |
| 83 | 158.68° | 62.61° | 81 | 1393.86 | 143.60° | 67.44° | 1,500 | 0.1537 | 254.05° |
| 84 | 128.51° | 72.26° | 82 | 603.43 | 120.52° | 73.84° | 2,600 | 0.1999 | 234.78° |
| 85 | 112.52° | 75.42° | 83 | 469.11 | 104.03° | 75.62° | 2,600 | 0.1554 | 275.25° |
| 86 | 95.54° | 75.81° | 84 | 440.81 | 89.21° | 74.73° | 3,300 | 0.2352 | 302.95° |
| 87 | 82.88° | 73.64° | 85 | 490.57 | 75.85° | 72.86° | 3,300 | 0.2618 | 290.54° |
| 88 | 68.82° | 72.08° | 86 | 529.37 | 63.55° | 70.47° | 3,800 | 0.3746 | 312.33° |
| 89 | 58.27° | 68.86° | 87 | 603.56 | 51.38° | 68.00° | 3,800 | 0.4271 | 288.45° |
| 90 | 44.49° | 67.13° | 88 | 546.62 | 38.16° | 67.67° | 3,800 | 0.3868 | 257.5° |
| 91 | 31.83° | 68.20° | 89 | 753.91 | 23.04° | 67.60° | 3,800 | 0.5334 | 280.17° |
| 92 | 14.25° | 66.99° | 90 | 733.5 | 9.68° | 64.34° | 300 | 0.0032 | 323.21° |
| 93 | 5.11° | 61.69° | 91 | 926.31 | 6.17° | 57.56° | 5,100 | 1.1806 | 7.80° |
| 94 | 7.22° | 53.43° | 92 | 708.24 | 183.36° | 51.35° | 3,000 | 0.3123 | 310.79° |
| 95 | 359.49° | 49.26° | 93 | 591.35 | 359.14° | 46.61° | 5,100 | 0.7537 | 5.18° |
| 96 | 358.79° | 43.96° | 94 | 761.67 | 354.04° | 43.83° | 5,100 | 0.9707 | 267.83° |
| 97 | 349.29° | 43.70° | 95 | 1104.32 | 351.26° | 38.97° | 5,100 | 1.4074 | 17.89° |
| 98 | 133.29° | -38.42° | 96 | 1032.06 | 128.33° | -36.08° | 5,200 | 1.3674 | 120.28° |
| 99 | 123.36° | -33.73° | 97 | 774.89 | 127.27° | -32.59° | 5,200 | 1.0267 | 109.16° |
| 100 | 131.18° | -31.44° | 98 | 1089.81 | 135.80° | -34.55° | 5,200 | 1.444 | 50.76° |
| 101 | 140.41° | -37.65° | 99 | 875.86 | 143.60° | -40.77° | 5,200 | 1.1605 | 37.70° |
| 102 | 146.78° | -43.89° | 100 | 958.73 | 148.49° | -39.78° | 4,500 | 0.9513 | 162.28° |
| 103 | 150.20° | -35.67° | 101 | 877.51 | 152.09° | -32.06° | 4,500 | 0.8707 | 156.08° |
| 104 | 153.98° | -28.45° | 102 | 943.16 | 151.56° | -24.82° | 3,200 | 0.4732 | 211.22° |

Notes: all geographic sites refer to Figure 4.





**Table 1(D). Basic information for the horizontal forces (continued)**

| | | | | | Side | | | | |
|---|---|---|---|---|---|---|---|---|---|
| | Control site | | | Length | Hypothetical geocentric center | | Ocean depth | Horizontal force | |
| $j$ | $d_j$ | $q_j$ | $i$ | $L_i$ | $\alpha_i$ | $\beta_i$ | $h_{i\text{-ocean}}$ | $F_i$ | Inclination to latitude, east ($\gamma_i$) |
| | Longitude | Latitude | | km | Longitude | Latitude | m | N ($*10^{17}$) | Degrees |
| 105 | 149.13° | -21.19° | 103 | 1359.76 | 145.75° | -16.01° | 3,200 | 0.6823 | 212.08° |
| 106 | 142.36° | -10.82° | 104 | 802.72 | 141.57° | -14.35° | 100 | 0.0004 | 347.77° |
| 107 | 140.78° | -17.88° | 105 | 1216.8 | 135.51° | -15.81° | 100 | 0.0006 | 247.79° |
| 108 | 130.24° | -13.74° | 106 | 1109.61 | 125.89° | -16.48° | 100 | 0.0005 | 303.28° |
| 109 | 121.53° | -19.22° | 107 | 864.63 | 117.80° | -20.95° | 4,500 | 0.8579 | 296.31° |
| 110 | 114.06° | -22.67° | 108 | 2529.37 | 123.68° | -30.55° | 5,000 | 3.0985 | 46.39° |
| 111 | 188.47° | -78.56° | 109 | 391.5 | 196.72° | -78.09° | 4,200 | 0.3384 | 285.48° |
| 112 | 204.97° | -77.61° | 110 | 519.06 | 213.35° | -76.35° | 4,200 | 0.4487 | 302.42° |
| 113 | 221.73° | -75.08 | 111 | 461.58 | 229.17° | -74.47° | 4,200 | 0.399 | 286.94° |
| 114 | 236.61° | -73.86° | 112 | 401.97 | 243.06° | -74.38° | 4,500 | 0.3989 | 253.39° |
| 115 | 249.51° | -74.90° | 113 | 780.47 | 261.99° | -74.03° | 4,500 | 0.7744 | 284.02° |
| 116 | 274.46° | -73.16° | 114 | 477.75 | 281.74° | -72.91° | 4,500 | 0.4741 | 263.24° |
| 117 | 289.02° | -72.65° | 115 | 1595.2 | 298.54° | -66.48° | 4,500 | 1.5828 | 328.21° |
| 118 | 308.06° | -60.30° | 116 | 1702.14 | 305.75° | -67.92° | 5,000 | 2.0851 | 186.49° |
| 119 | 303.40° | -75.53° | 117 | 418.96 | 309.48° | -76.83° | 5,000 | 0.5132 | 226.90° |
| 120 | 315.53° | -78.12° | 118 | 850.21 | 327.66° | -75.67° | 5,000 | 1.0415 | 308.82° |
| 121 | 339.79° | -73.22° | 119 | 644.56 | 347.35° | -71.56° | 5,000 | 0.7896 | 304.72° |
| 122 | 354.90° | -69.89° | 120 | 919.54 | 186.86° | -69.65° | 5,000 | 1.1264 | 266.74° |
| 123 | 18.81° | -69.41° | 121 | 674.24 | 26.90° | -68.64° | 5,000 | 0.8259 | 284.57° |
| 124 | 34.98° | -67.87° | 122 | 819.5 | 44.05° | -66.87° | 5,000 | 1.0039 | 285.64° |
| 125 | 53.12° | -65.86° | 123 | 838.69 | 62.26° | -67.22° | 5,000 | 1.0274 | 249.11° |
| 126 | 71.40° | -68.58° | 124 | 983.34 | 82.48° | -67.32° | 5,000 | 1.2046 | 286.27° |
| 127 | 93.55° | -66.06° | 125 | 675.92 | 101.11° | -66.29° | 4,500 | 0.6707 | 265.79° |
| 128 | 108.67° | -66.51° | 126 | 908.21 | 119.04° | -66.82° | 4,500 | 0.9012 | 267.30° |



| 129 | 129.41° | -66.90° | 127 | 654.62 | 136.97° | -67.01° | 4,500 | 0.6495 | 267.88° |
| 130 | 144.52° | -67.12° | 128 | 1098.65 | 157.08° | -69.49° | 3,900 | 0.8188 | 242.07° |
| 131 | 169.63° | -71.85° | 129 | 631.15 | 166.64° | -74.58° | 3,300 | 0.3368 | 163.76° |
| 132 | 163.65° | -77.31° | 130 | 645.18 | 177.36° | -78.01° | 3,300 | 0.3443 | 256.53° |
| 133 | 191.07° | -78.70° | | | | | | | |
| A | 26.50° | 22.40° | AF-EU-C (EU-AF-C) | | | | | 4.2000 | 66.13° (246.13°) |
| B | 21.05° | 80.92° | IN-EU-C (EU-IN-C) | | | | | 1.3000 | 71.47° (251.47°) |
| C | 12.15° | 104.09° | AU-EU-C (EU-AU-C) | | | | | 1.5000 | 54.74° (234.74°) |
| D | 88.00° | 84.59° | NA-EU-S (EU-NS-S) | | | | | 4.0000 | 0.00° (180.00°) |
| E | -19.64° | 148.17° | AU-PA-C (PA-AU-C) | | | | | 1.0000 | 55.00° (235.00°) |
| F | 41.22° | 238.99° | NA-PA-C (PA-NA-C) | | | | | 3.0000 | 185.00° (5.00°) |
| G | -2.20° | 278.96° | Naz-SA-C (SA-Naz-C) | | | | | 0.2000 | 0.00° (180.00°) |
| H | 43.68° | 144.98° | PA-EU-C (EU-PA-C) | | | | | 0.1000 | 180.00° (0.00°) |
| I | 20.68° | 35.70° | AF-EU-S (EU-AF-S) | | | | | 0.8000 | 170.00° (350.00°) |

Notes: all geographic sites refer to Figure 4.






## 3.2 Resultant movements of plates

The continents are fixed on the top of the lithosphere, and the lithospheric plates connect to each other, this relationship allows the ocean-generated force to be laterally transferred to the lithospheric plates. Subsequently, we list the plausible forces that

act on a sample continental plate (Figure 5) and discuss the physical nature of these forces.

These forces can be classified into two categories: the forces acting on the parts of the continent that connect to the oceans and those acting at both the bottom surface of the continental plate and the parts of the continental plate that connect to adjacent plates. The forces acting on the parts of the continent that connect to the ocean derive from ocean pressure and are treated as ocean-generated forces, denoted as $F_R$ on the right and $F_L$ on the left. The horizontal forces decomposed from these forces are

denoted as $F_R$' on the right and $F_L$' on the left. The force acting on the bottom surface of the continental plate arises from a coupling between the plate and underlying viscous asthenosphere. This force is called the basal friction force and is denoted as $f_{base}$. According to Forsyth & Uyeda (1975), if there is thermal convection in the asthenosphere, $f_{base}$ would be a driving force (Runcorn, 1962a, b; Turcotte & Oxburgh, 1972; Morgan, 1972). If, instead, the asthenosphere is passive relative to plate motion, $f_{base}$ would be a resistive force. Here, we assume $f_{base}$ to be a resistive force. Given that the continental plate moves

toward the right, the forces acting on the parts of the continental plate that connect to adjacent plates include the collisional force from the plate on the right side and the push force from the ridge on the left side, they are denoted as $F_C$ and $F_{ridge}$, respectively.

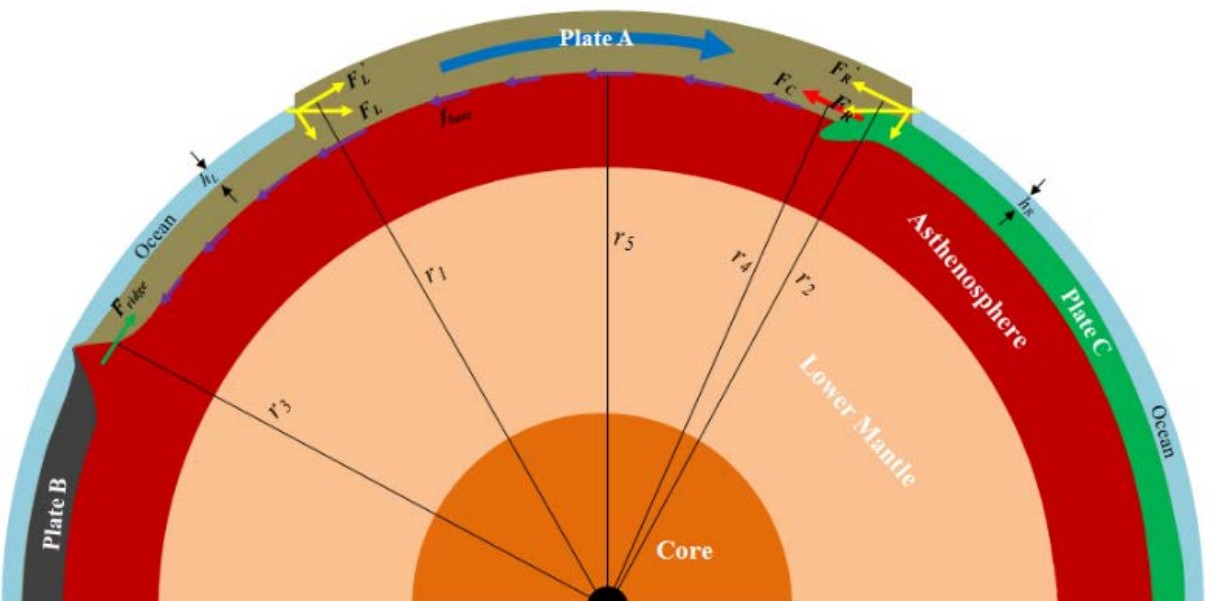

**Figure 5: Modeling the dynamics of a continental plate.** $F_L(F_R)$ represents the ocean-generated force on the left (right) side

of Plate A, while $F_L$'($F_R$') and $F_L$''($F_R$'') denote the horizontal and vertical forces decomposed from the ocean-generated force,





respectively. $f_{base}$ denotes the basal friction force exerted by the underlying asthenosphere, while $F_C$ and $F_{ridge}$ denote the collisional force from Plate C on the right side and the push force from the ridge on the left side, respectively. $h_L$ and $h_R$ are the ocean depths on the left and right, respectively. $r_1$, $r_2$, $r_3$, $r_4$, and $r_5$ denote the distances of these forces to the Earth's center. Note that the ocean depth and lithospheric plate thickness are highly exaggerated.


Plate motion is conventionally understood as a rigid plate rotating about an axis that penetrates the Earth's center, and this rotation must be a consequence of the integrated effect of all torques acting on the plate (e.g., Richardson, 1992; Forsyth & Uyeda, 1975). Following this understanding, we use torque balance to discuss the movement caused by these forces. According to Figure 5, a combined torque for Plate A may be written as

$\tau = (r_1F_L' - r_2F_R') + r_3F_{ridge} - (r_4F_C + r_5f_{base})$ (1)

where the first term $(r_1F_L' - r_2F_R')$ denotes the torque yielded by the final horizontal force, the second term $r_3F_{ridge}$ denotes the torque yielded by the ridge push force, both of them represent the driving torque for the continental plate, and the third term $(r_4F_C + r_5f_{base})$ denotes the resisting torque, which hinders the movement of the continental plate. Taking into consideration the reality that the plate is too thin (e.g., less than few hundred kilometers) relative to the Earth's radius (e.g., more than six

thousand kilometers), we approximate $r_1 = r_2 = r_3 = r_4 = r_5$. $F_L'$ and $F_R'$ may be further written as $F_L' = 0.5\rho gLh_L^2$ and $F_R' = 0.5\rho gLh_R^2$, where $\rho$, g, L, $h_L$, and $h_R$ are the density of water, gravitational acceleration, ocean width that fits the continent's width, ocean depth at the left, and ocean depth at the right, respectively.

Equation (1) provides three possibilities for the continental plate. If the driving torque is greater than the resisting torque, the combined torque is greater than zero, and the continental plate is subjected to an accelerating motion. Practically, it is

impossible for the continental plate to undergo such a movement. If the driving torque is equal to the resisting torque, the combined torque is zero, and the continental plate would be subjected to a steady motion. If the driving torque is less than the resisting torque, the combined torque is less than zero, and the continental plate remains motionless.

Plate A's movement exhibited in Figure 5 is parallel to that of Plate B and Plate C, this situation is rather idealized. Practically, the movements of most plates intersect with each other. For instance, the South American Plate moves northwest, the Nazca

Plate moves eastward, the African Plate moves northeast, the Eurasian Plate moves eastward. These nonparallel movements would yield additional collisional forces and shearing forces between plates. If two plates are not moving in the opposite direction, the collisional and shearing forces between them may be driving; and if the two plates are moving in the opposite direction, the collisional and shearing forces between them may be resisting. Below, we develop two semi-analytic methods (I and II) to independently resolve plate motion.


*Method I*

It is assumed that the Earth's surface is covered with Plate A, Plate B, Plate C, Plate D, and others, and that Euler pole of each plate has been established (Figure 6). For Plate A, the horizontal force $F_i$ (i=1, 2, 3, 4, and 5) acts on the side of the continent



that is fixed on top of Plate A. The horizontal force ($F_1$, for instance) yields a component ($F_1{}'$, for instance) that is orthogonal to the rotation axis of the plate; this component then yields a torque ($\tau_1{}'$, for instance). The torques yielded by all the components decomposed from the horizontal forces are summed into first driving torque. The ridge push force $F_{r-i}$ ($i$=1, 2, 3, and 4) acts on the edge of the plate, this force also yields a component that is orthogonal to the rotation axis; this component also yields a torque. The torques yielded by all the components decomposed from the ridge push forces are summed into second driving torque. Given that Plate A, Plate B, Plate C, and Plate D move eastward, southward, westward, and eastward, respectively, and that Plate D moves faster than Plate A, these make Plate A undergo a collisional driving force $F_{B-c}$ from Plate B, a shearing driving force $F_{D-s}$ from Plate D, a collisional resistive force $F_{C-c}$ from Plate C, a shearing resistive force $F_{B-s}$ from Plate B, and a basal friction force $F_{basal}$ from the underlying viscous asthenosphere. The collisional driving force $F_{B-c}$ and the shearing driving force $F_{D-s}$ also yield two components that are orthogonal to the rotation axis of the plate; these components also yield torques. The torques yielded by these two components are summed into third driving torque. The collisional resistive force $F_{C-c}$ and the shearing resistive force $F_{B-s}$ also yield two components that are orthogonal to the rotation axis; these components also yield torques. The torques yielded by these two components are summed into first resistive torque. The basal friction force $F_{basal}$ yields second resistive torque.

Then, we divide these five sets of torques into two exerting parts: one, which includes the second driving torque and a little portion of the first and third driving torque, balances out the first resistive torque, and the other, which including the remaining portion of the first and third driving torque, balances the second resistive torque. Consequently, all these torque balances allow Plate A to be steadily rotated under the assumption that the acceleration and inertia of the plate are neglected. The remaining portion of the first and third driving torque is called the net driving torque, and the second resistive torque is called the net resistive torque. The balance between the net driving torque and the net resistive torque may be written as

$$\tau_{driving} - \tau_{basal} = 0 \qquad\qquad (2)$$

where $\tau_{driving}$ is the net driving torque, $\tau_{driving} = \varepsilon\tau$, and $\varepsilon$ is the ratio of the net driving torque to the first and third driving torque. As shown in Figure 6(A), the component decomposed from a force (the horizontal force, for instance) may be written as $F_{i'} = F_i \cos\eta_i$, and $\eta_i = \gamma_i - \lambda_i$, where $\gamma_i$ is the inclination of this force to latitude. $\lambda_i$ is the azimuth of arc $P_iE$ with respect to latitude. This component yields a torque $\tau_i$ with respect to the rotation axis, i.e., $\tau_i = r_i F_{i'}$, where $r_i$ denotes the lever arm distance of the component $F_{i'}$, $r_i = R_{earth}\sin\varphi_{Pi}$, $R_{earth}$ is the Earth's radius and $R_{earth}$=6,371 km, and $\varphi_{Pi}$ is the angle of site $P_i$ and the Euler pole. A sum of the torques yielded by the components, which are decomposed from the horizontal forces, collisional driving forces, and shearing driving forces, forms the first and third driving torque $\tau$. $\tau_{basal}$ denotes the net resistive torque yielded by the basal friction force, it can be written as $\tau_{basal} = r_K F_{basal}$, where $r_K$ denotes the lever arm distance of the basal friction force. According to the principle of viscous fluid mechanics, the basal friction force may be expressed as $F_{basal} = \mu Au/y$, $\mu$, A, u, and y are the viscosity of the asthenosphere, the plate's area, the plate's speed, and the thickness of the asthenosphere, respectively. Therefore, $u = y\tau_{driving}/\mu A$, this speed represents average level of the plate's movement.





In general, the largest speed of a plate occurs at the plate's equator, while the smallest speed occurs at the location whose angle distance to the Euler pole is minimal or maximal. As shown in Figure 6(B), we assume that the geometric center (i.e., location $K$) of Plate A moves at the average speed, namely, $u=u_k$. And then, the speed of any location $S$ within this plate may be expressed with $u_s=u_k\sin\varphi_s/\sin\varphi_k$, where $\varphi_s(\varphi_k)$ is the angle distance of location $S$ ($K$) to the Euler pole (i.e., location $E$) relative to the Earth's center. The speed $u_s$ can be further decomposed into the longitudinal speed $u_{s-lo}$ and latitudinal speed $u_{s-la}$, and $u_{s-lo}=u_s\sin(\lambda_s-90^\circ)$, $u_{s-la}=u_s\cos(\lambda_s-90^\circ)$, where $\lambda_s$ is the azimuth of arc $SE$ with respect to latitude. The azimuth of the movement is then calculated through the longitudinal and latitudinal speeds. All these angles and distances (i.e., $\eta_i$, $\gamma_i$, $\lambda_i$, $\lambda_s$, $\varphi_{Pi}$, $\varphi_k$, $\varphi_s$, $r_i$, $r_K$) may be further calculated through the latitudes and longitudes of related locations.



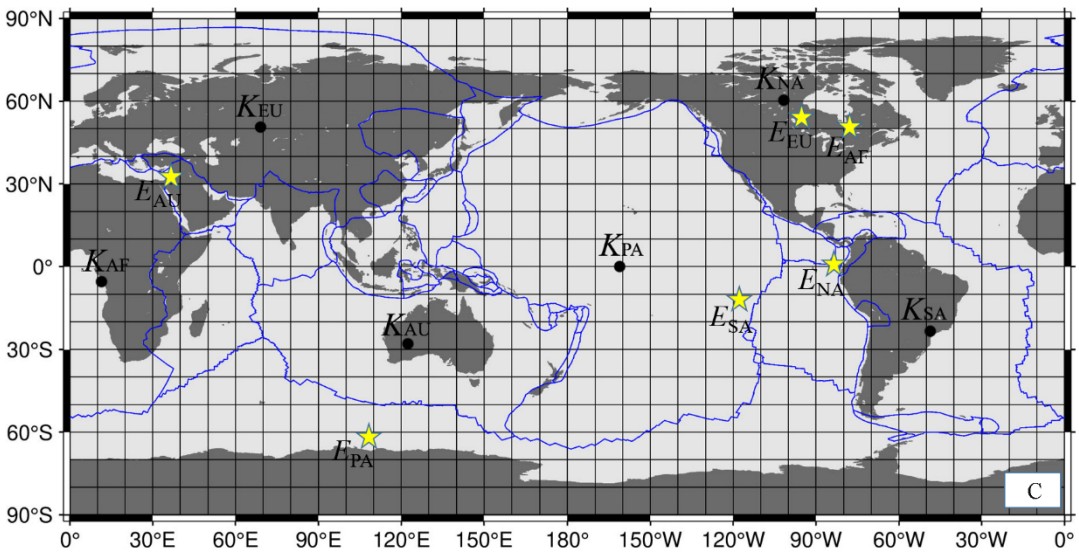

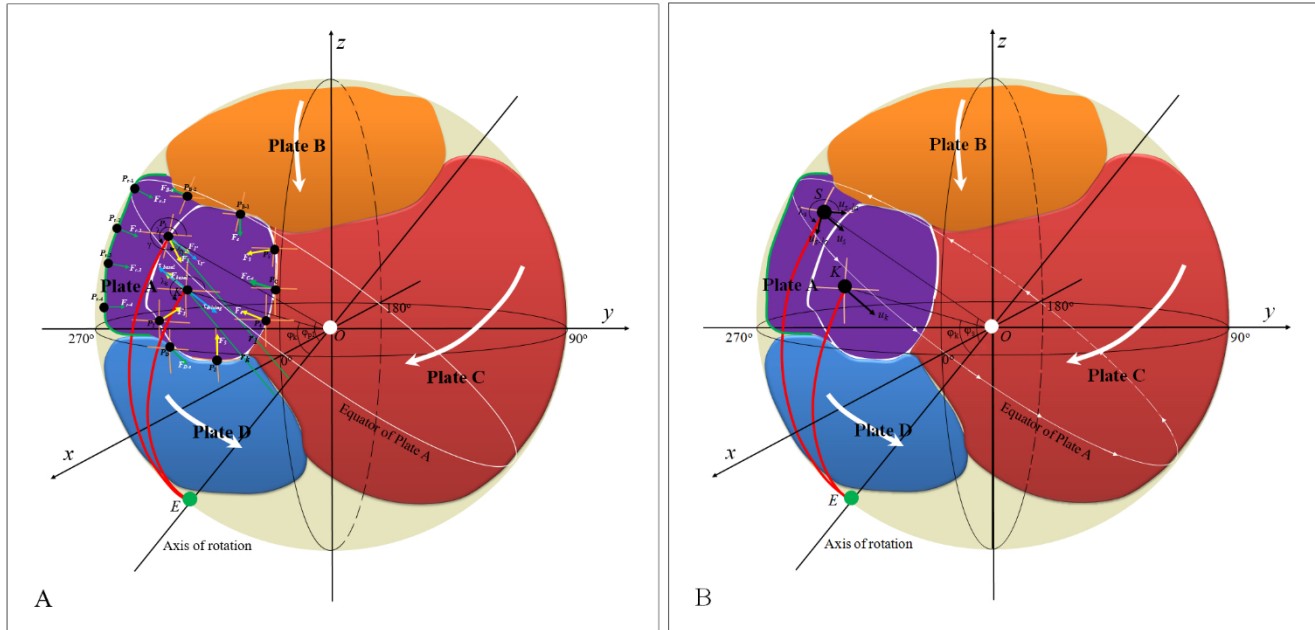


**Figure 6: Modeling the torque balances for plate motion. (A) Geometry of horizontal forces, ridge push forces, collisional forces, and shearing forces over a spherical frame.** The white oval within Plate A represents the scope of the continent. The green line denotes the oceanic ridge. The pink lines beneath the locations (black dots) denote the latitudinal and longitudinal directions. **(B) Decomposing the average movement of the plate into the movement of any location.** The





location $K$ and $E$ are the geometric center of Plate A and its Euler pole, respectively. **(C) Exhibiting the geometric centers (black dots) of the six selected plates and the established Euler pole locations (yellow stars) over a planar frame.**

Here, we use six plates (South American, African, Eurasian, North American, Australian, and Pacific plates) to demonstrate their movements. In order to simplify the following deduction, we plot globally tectonic plates into a grid of $10°×10°$ and use

these grid nodes, which are within plate, to obtain the geometric center of each plate. The geometric center is approximately calculated through the average of the latitudes and longitudes of these nodes. The Euler pole location of each of these plates is cited from the GSRM v.2.1 (Kreemer et al., 2014). Both the geometric center of each plate and its Euler pole location are exhibited in Figure 6(C).

Besides the horizontal forces, the other possible forces (i.e., collisional and shearing) for these plates must be considered. For

example, the African, Indian, and Australian plates provide the collisional driving forces $F_{AF\text{-}EU\text{-}C}$, $F_{IN\text{-}EU\text{-}C}$, and $F_{AU\text{-}EU\text{-}C}$ for the Eurasian Plate, respectively. The Nazca plate provides a collisional driving force $F_{Naz\text{-}SA\text{-}C}$ for the South American Plate. The Eurasian Plate provides a shearing resistive force $F_{EU\text{-}NA\text{-}S}$ for the North American Plate; vice versa, the North American Plate provides a shearing driving force $F_{NA\text{-}EU\text{-}S}$ for the Eurasian Plate. The Australian, North American, and Eurasian Plates provide the collisional driving forces $F_{AU\text{-}PA\text{-}C}$, $F_{NA\text{-}PA\text{-}C}$, and $F_{EU\text{-}PA\text{-}C}$ for the Pacific Plate, respectively. Taking into

consideration the long argument of slab pull that is listed in section 2.1, we presently neglect slab pull. And if this force can be confirmed in the future, it can be added into this model. The details of these forces are exhibited in Figure 4 and listed Table 1. The resultant torques from all related forces are listed in Table 2.








**Table 2 (A). Parameters for the torques of six selected plates in the method I**

| Plate | Eule pole | | No. | Angle between horizontal force and its decomposed force | Decomposed force | Angle of site $P_i$ and Eule pole | Earth's radius | Lever arm distance | Torque |
|---|---|---|---|---|---|---|---|---|---|
| | *E* | | | $\eta_i$ | $F_{i'}$ | $\varphi_{pi}$ | $R_{earth}$ | $r_i$ | $\tau_i$ |
| | Latitude | Longitude | *i* | Degrees | N (*$10^{17}$) | Degrees | m (*$10^3$) | m (*$10^3$) | N·m (*$10^{23}$) |
| | | | 1 | 97.27° | 0.2497 | 79.30° | 6371.00 | 6260.27 | -1.5632 |
| | | | 2 | 92.72° | 0.0574 | 72.92° | 6371.00 | 6089.89 | -0.3493 |
| | | | 3 | 86.94° | 0.0381 | 63.32° | 6371.00 | 5692.78 | 0.2170 |
| | | | 4 | 316.47° | 0.4054 | 55.68° | 6371.00 | 5261.94 | 2.1332 |
| | | | 5 | 353.88° | 0.9497 | 49.51° | 6371.00 | 4845.13 | 4.6012 |
| | | | 6 | 358.11° | 1.3675 | 39.66° | 6371.00 | 4066.52 | 5.5611 |
| | | | 7 | 343.60° | 0.4981 | 30.91° | 6371.00 | 3272.51 | 1.6300 |
| | | | 8 | 14.05° | 0.7108 | 23.29° | 6371.00 | 2519.47 | 1.7909 |
| | | | 9 | 212.77° | 0.8725 | 26.03° | 6371.00 | 2795.38 | -2.4389 |
| | | | 10 | 115.64° | 0.6455 | 33.23° | 6371.00 | 3490.93 | -2.2536 |
| North America | 2.19° | 276.25° | 11 | 129.96° | 0.9586 | 41.83° | 6371.00 | 4249.32 | -4.0733 |
| | | | 12 | 101.72° | 0.2780 | 50.75° | 6371.00 | 4933.75 | -1.3715 |
| | | | 13 | 227.96° | 0.3704 | 58.02° | 6371.00 | 5404.09 | -2.0014 |
| | | | 14 | 129.97° | 0.4880 | 63.81° | 6371.00 | 5717.10 | -2.7900 |
| | | | 15 | 65.38° | 0.1302 | 65.61° | 6371.00 | 5802.46 | 0.7556 |
| | | | 16 | 90.08° | 0.0004 | 69.11° | 6371.00 | 5952.09 | -0.0023 |
| | | | 17 | 109.75° | 0.0010 | 75.74° | 6371.00 | 6174.81 | -0.0061 |
| | | | 18 | 207.62° | 0.6048 | 81.54° | 6371.00 | 6301.68 | -3.8112 |
| | | | 19 | 281.86° | 0.0218 | 83.53° | 6371.00 | 6330.41 | 0.1377 |
| | | | 20 | 254.21° | 0.0165 | 81.97° | 6371.00 | 6308.49 | -0.1042 |
| | | | 21 | 258.36° | 0.0002 | 81.06° | 6371.00 | 6293.59 | -0.0013 |
| | | | 22 | 251.94° | 0.0175 | 80.59° | 6371.00 | 6285.20 | -0.1102 |





| | | No. | | | | | |
|---|---|---|---|---|---|---|---|
| | | 23 | 318.90° | 0.0601 | 79.05° | 6371.00 | 6255.08 | 0.3762 |
| | | 24 | 282.85° | 0.0155 | 77.95° | 6371.00 | 6230.53 | 0.0969 |
| | | 25 | 303.63° | 0.0668 | 77.54° | 6371.00 | 6220.87 | 0.4153 |
| | | 26 | 295.24° | 0.2553 | 76.48° | 6371.00 | 6194.44 | 1.5815 |
| | | 27 | 223.62° | 0.2277 | 79.34° | 6371.00 | 6261.13 | -1.4256 |
| | | 28 | 252.92° | 0.1328 | 85.13° | 6371.00 | 6347.96 | -0.8430 |
| | | 29 | 268.96° | 0.0032 | 93.29° | 6371.00 | 6360.48 | -0.0203 |
| | | 30 | 240.52° | 0.0001 | 93.27° | 6371.00 | 6360.61 | -0.0009 |
| | | EU-NA-S | 47.10° | 2.7231 | 89.81° | 6371.00 | 6370.96 | -17.3490 |
| | | PA-NA-C | 35.15° | 1.6353 | 51.43° | 6371.00 | 4981.26 | 8.1461 |
| | | | | | | | total | -13.0728 |

Note: the negative symbol "-" beneath torque denotes counterclockwise with respect to the axis of rotation.

**Table 2 (B). Parameters for the torques of six selected plates in the method I (continued)**

| Plate | Eule pole | | No. | Angle between horizontal force and its decomposed force | Decomposed force | Angle of site $P_i$ and Eule pole | Earth's radius | Lever arm distance | Torque |
|---|---|---|---|---|---|---|---|---|---|
| | $E$ | | | $\eta_i$ | $F_{i'}$ | $\varphi_{pi}$ | $R_{earth}$ | $r_i$ | $\tau_i$ |
| | Latitude | Longitude | $i$ | Degrees | N ($*10^{17}$) | Degrees | m ($*10^3$) | m ($*10^3$) | N·m ($*10^{23}$) |
| South America | -14.10° | 242.14° | 31 | 94.63° | 0.1175 | 38.23° | 6371.00 | 3942.62 | -0.4635 |
| | | | 32 | 117.64° | 0.5594 | 40.64° | 6371.00 | 4149.63 | -2.3212 |
| | | | 33 | 75.71° | 0.4446 | 45.73° | 6371.00 | 4561.87 | 2.0280 |
| | | | 34 | 68.24° | 0.4722 | 42.53° | 6371.00 | 4306.96 | 2.0337 |
| | | | 35 | 107.25° | 0.2812 | 41.10° | 6371.00 | 4188.48 | -1.1777 |
| | | | 36 | 247.03° | 0.0605 | 48.92° | 6371.00 | 4802.51 | -0.2907 |
| | | | 37 | 214.43° | 0.0958 | 50.48° | 6371.00 | 4914.25 | -0.4707 |

deep





| | | | 38 | 188.74° | 1.7689 | 54.93° | 6371.00 | 5214.44 | -9.2238 |
|---|---|---|---|---|---|---|---|---|---|
| | | | 39 | 200.24° | 1.9216 | 61.42° | 6371.00 | 5594.69 | -10.7506 |
| | | | 40 | 181.06° | 1.4396 | 66.20° | 6371.00 | 5829.25 | -8.3917 |
| | | | 41 | 212.82° | 1.3693 | 71.37° | 6371.00 | 6037.15 | -8.2667 |
| | | | 42 | 301.17° | 0.8180 | 74.55° | 6371.00 | 6140.76 | 5.0232 |
| | | | 43 | 286.96° | 0.4041 | 66.24° | 6371.00 | 5831.18 | 2.3565 |
| | | | 44 | 253.52° | 0.0936 | 58.47° | 6371.00 | 5430.40 | -0.5084 |
| | | | 45 | 347.68° | 1.6887 | 58.25° | 6371.00 | 5417.83 | 9.1491 |
| | | | Naz-SA-C | 71.58° | 0.0632 | 38.26° | 6371.00 | 3945.32 | 0.2493 |
| | | | | | | | | total | -21.0251 |
| | | | 46 | 70.14° | 0.3779 | 53.21° | 6371.00 | 5102.38 | 1.9283 |
| | | | 47 | 102.95° | 0.4016 | 56.52° | 6371.00 | 5314.13 | -2.1344 |
| | | | 48 | 164.92° | 1.6163 | 66.57° | 6371.00 | 5845.72 | -9.4486 |
| | | | 49 | 218.52° | 1.3702 | 77.78° | 6371.00 | 6226.76 | -8.5319 |
| | | | 50 | 145.39° | 0.7668 | 86.18° | 6371.00 | 6356.85 | -4.8747 |
| | | | 51 | 154.94° | 1.4114 | 94.61° | 6371.00 | 6350.43 | -8.9632 |
| | | | 52 | 117.56° | 0.6083 | 101.46° | 6371.00 | 6243.91 | -3.7983 |
| | | | 53 | 155.92° | 1.2945 | 108.55° | 6371.00 | 6039.87 | -7.8188 |
| Africa | 49.66° | 281.92° | 54 | 165.93° | 0.9259 | 116.64° | 6371.00 | 5694.53 | -5.2727 |
| | | | 55 | 137.16° | 0.2165 | 121.92° | 6371.00 | 5407.37 | -1.1709 |
| | | | 56 | 283.40° | 0.2429 | 123.18° | 6371.00 | 5331.97 | 1.2950 |
| | | | 57 | 275.06° | 0.0979 | 121.18° | 6371.00 | 5450.79 | 0.5337 |
| | | | 58 | 313.40° | 0.4141 | 115.74° | 6371.00 | 5738.80 | 2.3762 |
| | | | 59 | 264.22° | 0.0918 | 110.48° | 6371.00 | 5968.21 | -0.5481 |
| | | | 60 | 259.19° | 0.2878 | 107.05° | 6371.00 | 6090.91 | -1.7529 |
| | | | EU-AF-C | 7.26° | 4.1663 | 76.43° | 6371.00 | 6193.10 | 25.8022 |
| | | | EU-AF-S | 81.71° | 0.1154 | 88.57° | 6371.00 | 6369.01 | -0.7349 |
| | | | | | | | | | -23.1140 |

Note: the negative symbol "-" beneath torque denotes counterclockwise with respect to the axis of rotation.




**Table 2 (C). Parameters for the torques of six selected plates in the method I (continued)**

| late | Eule pole | | No. | Angle between horizontal force and its decomposed force | Decomposed force | Angle of site $P_i$ and Eule pole | Earth's radius | Length of lever arm | Torque |
|---|---|---|---|---|---|---|---|---|---|
| | $E$ | | | $\eta_i$ | $F_{i'}$ | $\varphi_{pi}$ | $R_{earth}$ | $r_i$ | $\tau_i$ |
| | Latitude | Longitude | $i$ | Degrees | N ($*10^{17}$) | Degrees | m ($*10^3$) | m ($*10^3$) | N·m ($*10^{23}$) |
| | | | 67 | 258.40° | 0.4439 | 115.16° | 6371.00 | 5766.75 | -2.5600 |
| | | | 68 | 149.71° | 2.0668 | 125.41° | 6371.00 | 5192.58 | -10.7320 |
| | | | 69 | 334.71° | 0.0010 | 122.62° | 6371.00 | 5365.89 | 0.0051 |
| | | | 70 | 83.52° | 0.1608 | 114.29° | 6371.00 | 5806.87 | 0.9336 |
| | | | 71 | 79.27° | 0.1196 | 109.58° | 6371.00 | 6002.67 | 0.7181 |
| | | | 72 | 37.66° | 0.4566 | 106.33° | 6371.00 | 6114.00 | 2.7916 |
| | | | 73 | 49.55° | 0.8144 | 102.01° | 6371.00 | 6231.63 | 5.0750 |
| | | | 74 | 21.11° | 0.6848 | 96.81° | 6371.00 | 6326.02 | 4.3322 |
| | | | 75 | 81.02° | 0.1270 | 92.43° | 6371.00 | 6365.29 | 0.8082 |
| | | | 76 | 62.48° | 0.8568 | 84.59° | 6371.00 | 6342.61 | 5.4341 |
| Eurasia | 55.38° | 264.59° | 77 | 40.27° | 1.3288 | 73.63° | 6371.00 | 6112.77 | 8.1227 |
| | | | 78 | 57.60° | 0.7856 | 64.43° | 6371.00 | 5747.16 | 4.5148 |
| | | | 79 | 31.07° | 1.5750 | 54.93° | 6371.00 | 5214.08 | 8.2124 |
| | | | 80 | 171.95° | 0.7962 | 49.33° | 6371.00 | 4832.06 | -3.8475 |
| | | | 81 | 154.76° | 0.1390 | 49.63° | 6371.00 | 4854.19 | -0.6747 |
| | | | 82 | 133.27° | 0.1370 | 48.52° | 6371.00 | 4772.97 | -0.6539 |
| | | | 83 | 174.32° | 0.1546 | 48.39° | 6371.00 | 4763.58 | -0.7366 |
| | | | 84 | 23.23° | 0.2161 | 49.86° | 6371.00 | 4870.34 | -1.0527 |
| | | | 85 | 11.67° | 0.2564 | 51.62° | 6371.00 | 4994.15 | -1.2803 |
| | | | 86 | 34.19° | 0.3098 | 53.25° | 6371.00 | 5104.74 | -1.5816 |
| | | | 87 | 11.01° | 0.4192 | 54.21° | 6371.00 | 5167.62 | -2.1662 |
| | | | 88 | 175.74° | 0.3857 | 52.24° | 6371.00 | 5036.62 | 1.9426 |





| No. | η | F' | φ | R | r | τ |
|---|---|---|---|---|---|---|
| 89 | 0.46° | 0.5334 | 48.88° | 6371.00 | 4799.38 | -2.5601 |
| 90 | 45.07° | 0.0023 | 47.33° | 6371.00 | 4684.66 | -0.0107 |
| 91 | 5.85° | 1.1744 | 50.70° | 6371.00 | 4930.18 | -5.7901 |
| 92 | 44.78° | 0.2217 | 45.84° | 6371.00 | 4570.16 | 1.0132 |
| 93 | 102.94° | 0.1687 | 55.45° | 6371.00 | 5247.62 | -0.8854 |
| 94 | 81.57° | 0.1424 | 54.98° | 6371.00 | 5217.66 | 0.7429 |
| 95 | 122.84° | 0.7633 | 57.09° | 6371.00 | 5348.90 | -4.0826 |
| AF-EU-C | 174.37° | 4.1797 | 82.54° | 6371.00 | 6317.01 | -26.4034 |
| IN-EU-C | 178.65° | 1.2996 | 103.51° | 6371.00 | 6194.68 | -8.0509 |
| AU-EU-C | 166.77° | 1.4602 | 110.51° | 6371.00 | 5967.18 | -8.7133 |
| NA-EU-S | 106.31° | 1.1233 | 36.62° | 6371.00 | 3800.34 | -4.2691 |
| PA-EU-C | 97.84° | 0.0136 | 68.57° | 6371.00 | 5930.61 | 0.0808 |
| AF-EU-S | 29.84° | 0.6940 | 93.37° | 6371.00 | 6359.96 | 4.4136 |
| | | | | | total | -36.9103 |

Note: the negative symbol "-" beneath torque denotes counterclockwise with respect to the axis of rotation.

**Table 2 (D). Parameters for the torques of six selected plates in the method I (continued)**

| Plate | Eule pole | | No. | Angle between horizontal force and its decomposed force | Decomposed force | Angle of site $P_i$ and Eule pole | Earth's radius | Length of lever arm | Torque |
|---|---|---|---|---|---|---|---|---|---|
| | $E$ | | | $\eta_i$ | $F_{i'}$ | $\varphi_{pi}$ | $R_{earth}$ | $r_i$ | $\tau_i$ |
| | Latitude | Longitude | $i$ | Degrees | N (*10^17) | Degrees | m (*10^3) | m (*10^3) | N·m (*10^23) |
| | | | 96 | 254.20° | 0.3723 | 110.26° | 6371.00 | 5976.79 | -2.22515 |
| | | | 97 | 241.45° | 0.4908 | 107.86° | 6371.00 | 6063.97 | -2.97597 |
| Australia | 33.31° | 36.38° | 98 | 182.17° | 1.4429 | 115.09° | 6371.00 | 5769.94 | -8.32561 |
| | | | 99 | 170.88° | 1.1458 | 123.09° | 6371.00 | 5337.79 | -6.11606 |
| | | | 100 | 294.07° | 0.3880 | 126.38° | 6371.00 | 5129.41 | 1.990312 |





| | | | | 101 | 283.24° | 0.1994 | 126.78° | 6371.00 | 5102.73 | 1.01746 |
|---|---|---|---|---|---|---|---|---|---|---|
| | | | | 102 | 334.60° | 0.4275 | 123.59° | 6371.00 | 5307.41 | 2.268856 |
| | | | | 103 | 331.54° | 0.5998 | 114.69° | 6371.00 | 5788.35 | 3.471845 |
| | | | | 104 | 106.97° | 0.0001 | 110.38° | 6371.00 | 5972.19 | -0.0007 |
| | | | | 105 | 261.10° | 0.0001 | 106.09° | 6371.00 | 6121.32 | -0.0006 |
| | | | | 106 | 66.81° | 0.0002 | 98.56° | 6371.00 | 6300.01 | 0.0013 |
| | | | | 107 | 64.56° | 0.3686 | 94.58° | 6371.00 | 6350.68 | 2.340567 |
| | | | | 108 | 178.47° | 3.0974 | 104.19° | 6371.00 | 6176.62 | -19.1313 |
| | | | | EU-AU-C | 15.71° | 1.4440 | 64.82° | 6371.00 | 5765.39 | 8.325007 |
| | | | | PA-AU-C | 3.92° | 0.9977 | 118.47° | 6371.00 | 5600.47 | 5.587377 |
| | | | | | | | | | total | -13.7727 |
| Pacific | -63.09° | 109.63° | | AU-PA-C | 87.49° | 0.0437 | 50.72° | 6371.00 | 4931.80 | 0.2157 |
| | | | | NA-PA-C | 39.99° | 2.2984 | 143.46° | 6371.00 | 3793.07 | -8.7181 |
| | | | | EU-PA-C | 12.55° | 0.0976 | 110.42° | 6371.00 | 5970.70 | 0.5828 |
| | | | | | | | | | total | -7.9195 |

Note: the negative symbol "-" beneath torque denotes counterclockwise with respect to the axis of rotation.





The asthenosphere viscosity is not yet exactly determined. Many numerical studies using glacial isostatic adjustment and geoid modeling have shown that asthenospheric viscosity ranges from $10^{17}$ to $10^{20}$ Pas (e.g., Steinberger, 2016; Hager and Richards, 1989; Mitrovica, 1996; King, 1995; Kido et al., 1998; James et al., 2009; Pollitz et al., 1998; Berker, 2017; Kaufmann and

Lambeck, 2000; Hu et al., 2016). Laboratory experiments, however, suggested that the magnitude of the asthenospheric viscosity could be substantially different from those constrained by numerical studies. The viscosity is variable and likely related to the thermodynamic state, grain size, composition of the medium, and state of stress (Bercovici et al., 2015). Both the melt contents of the asthenosphere and the water in the asthenosphere may greatly affect the viscosity (Mei et al., 2002; Hirth and Kohlstedt, 1996). Hirth and Kohlstedt (1996) reported a variable viscosity profile for a melt-free oceanic lithosphere

with a mean value of $\sim 10^{18}$ Pas. These authors (e.g., Doglioni et al., 2011; Scoppola et al., 2006) concluded that, in consideration of the water- and melt-rich layers characterized by much lower viscosities, a strong vertical variability of viscosity may be more realistic. The asthenosphere's effective viscosity can be greatly lowered to $10^{15}$ Pas if the water content in the case of both diffusion and dislocation creep is included (Korenaga and Karato, 2008). Scoppola et al. (2006) conducted a more detailed review of asthenospheric viscosity and concluded that the presently accepted values of viscosity might be

reduced through a combined experiment including these parameters (i.e., melt content, water content, mechanical anisotropy, and shear localization). A "superweak", low-viscosity asthenosphere supported by recent observations is being accepted by the geophysical community (Kawakatsu et al., 2009; Hawley et al., 2016; Holtzman, 2016; Naif et al., 2013; Freed et al., 2017; Hu et al., 2016; Stern et al., 2015; Bercker, 2017). Jordan (1974) treated the asthenospheric thickness as 300 km.

Taking into account the present status of the viscosity and thickness of the asthenosphere above, we adopt y=300 km for each

of the six selected plates, $\mu=10^{18}$ Pas for the South American, African, North American, and Eurasian plates, $\mu=0.6\times10^{18}$ Pas for the Australian Plate, and $\mu=0.12\times10^{18}$ Pas for the Pacific Plate.

The other parameters (i.e., plate area, the ratio of the net driving torque and the first and third driving torque) and the resultant average movements of these six plates are listed in Table 3.

There have been many plate motion models (i.e., GSRM, NUVEL-1, and MORVEL) that include global navigation satellite

systems (GNSS) and paleomagnetic data. For instance, GSRM v.2.1 includes more than 6,739 continuous GPS velocity measurements (Kreemer et al., 2014). The movements reproduced by these models may approximately represent observations. Here, the movements of 450 locations (41 for the South American Plate, 70 for the African Plate, 93 for the North American Plate, 95 for the Eurasian Plate, 47 for the Australian Plate, and 104 for the Pacific Plate) are reproduced by GSRM v.2.1. The calculated and reproduced movements are then compared in Figure 7. It can been seen that the calculated movements for these

locations are well consistent with the reproduced movements in both speed and azimuth, the RMS of the calculated speed against the reproduced speed for the South American, African, North American, Eurasian, Australian, and Pacific plates is 0.91, 3.76, 2.77, 2.31, 7.43, and 1.95 mm/yr, respectively.



**Table 3. The net driving torques and their resultant movements for these six selected plates in the method I**

| Plate | Area | Ratio | Net driving torque | Geometric center of the plate | | Lever arm distance for basal friction force | Movement |
|---|---|---|---|---|---|---|---|
| | A | $\varepsilon$ | $\tau_{driving}$ | $K$ | $\varphi_k$ | $r_k$ | u |
| | km² | | N·m ($10^{23}$) | Latitude | Longitude | Degrees | m ($10^3$) | mm/yr |
| South America | 43,600,000 | 0.15 | 3.18 | -24.39° | 313.66° | 67.63° | 5891.49 | 11.73 |
| Africa | 61,300,000 | 0.64 | 14.74 | -5.57° | 13.43° | 95.22° | 6344.59 | 35.85 |
| North America | 75,900,000 | 0.79 | 10.39 | 59.57° | 256.88° | 59.31° | 5478.57 | 23.64 |
| Eurasia | 67,800,000 | 0.31 | 11.51 | 50.32° | 68.84° | 73.49° | 6108.39 | 26.29 |
| Australia | 47,000,000 | 0.91 | 12.55 | -28.30° | 122.98° | 102.51° | 6219.65 | 67.72 |
| Pacific | 103,300,000 | 0.75 | 5.98 | 0.10° | 198.65° | 89.64° | 6370.88 | 71.61 |






**Figure 7: The reproduced movements from GSRM v.2.1 (black arrows) verse the calculated movements from our model (red arrows) in the method I.** a), b), c), d), e), and f) are the South American, African, North American, Eurasian, Australian, and Pacific plates, respectively.



*Method II*

We assume that the Earth's surface is covered with Plate A, Plate B, Plate C, Plate D, and others (Figure 8). For Plate A, it undergoes the horizontal force $F_i$ ($i$=1, 2, 3, 4, and 5), the ridge push force $F_{r-i}$ ($i$=1, 2, 3, and 4), the collisional driving force $F_{B-c}$, the shearing driving force $F_{D-s}$, the collisional resistive force $F_{C-c}$, the shearing resistive force $F_{B-s}$, and the basal friction

force $F_{basal}$. One horizontal force ($F_1$, for instance) yields a torque ($\tau_1$, for instance), another horizontal force ($F_2$, for instance) yields another torque ($\tau_2$, for instance), a combination of these two torques results in a new torque ($\tau_{1-2}$, for instance), this new torque then combines the torque yielded by third horizontal force to form another new torque. Subsequently, the torques yielded by all the horizontal forces are combined into a final torque. The collisional driving force $F_{B-c}$ yields a torque, the shearing driving force $F_{D-s}$ yields a torque, the final torque combines these two torques to firm first driving torque. The collisional

resistive force $F_{C-c}$ and the shearing resistive force $F_{B-s}$ also yield two torques, they combine to firm first resistive torque. The basal friction force $F_{basal}$ yields second resistive torque. The ridge push force $F_{r-i}$ ($i$=1, 2, 3, and 4) also yields a torque, the torques yielded by all the ridge push forces are combined into second driving torque.

Then, we divide these four sets of torques into two exerting parts: one, which includes the second driving torque and a portion of the first driving torque, balances out first the resistive torque, and the other, which including the remaining portion of the

first driving torque, balances the second resistive torque. Consequently, all these torque balances allow Plate A to be steadily rotated under the assumption that the acceleration and inertia of the plate are neglected. The remaining portion of the first driving torque is called the net driving torque, and second resistive torque is called the net resistive torque. We assume that the net driving torque exerts on the geometric center (i.e., location $K$) of Plate A, this makes the plate move along a big circle that represents the equator of this Plate. And then, the balance between the net driving torque and the net resistive torque may be

expressed with Equation (2). According to Figure 8(A), a force $F_i$ yields a torque $\tau_i$ with respect to the Earth's center, i.e., $\tau_i = R_{earth} F_i$, where $R_{earth}$ is the Earth's radius and $R_{earth}$=6,371 km. The combination of two torques follows the trigonometric principle and may be written as

$$\tau_j{}^2 = \tau_i{}^2 + \tau_{i+1}{}^2 + 2\tau_i\tau_{i+1}\cos(\gamma_i-\gamma_{i+1}) \qquad (3)$$

where $\tau_j$ is the combined torque, $\tau_i$ and $\tau_{i+1}$ are the torque yielded by the force $F_i$ and $F_{i+1}$, respectively. $\gamma_i$ and $\gamma_{i+1}$ denote the

inclination of the forces $F_i$ and $F_{i+1}$ to latitude, respectively. $\tau_{basal}$ denotes the net resistive torque yielded by the basal friction force, it can be written as $\tau_{basal} = R_{earth} F_{basal}$. According to the principle of viscous fluid mechanics, the basal friction force may be expressed as $F_{basal} = \mu A u/y$, $\mu$, $A$, $u$, and $y$ are the viscosity of the asthenosphere, the plate's area, the plate's speed, and the thickness of the asthenosphere, respectively. Therefore, $u = y\tau_{driving}/\mu A$, this speed represents average level of the plate's movement.

On the whole, the largest speed of a plate occurs at the plate's equator, while the smallest speed occurs at the location whose angle distance to the Euler pole is minimal or maximal. According to Figure 8(B), we assume that the geometric center (i.e., location $K$) of Plate A moves at a speed of $u_k = \zeta u$, where $\zeta$ is an amplification coefficient of the average speed. And then, the speed of any location $S$ within this plate may be expressed with $u_s = u_k \sin\varphi_s / \sin\varphi_k$, where $\varphi_s(\varphi_k)$ is the angle distance of location



$S$ ($K$) to the Euler pole (i.e., location $E$) relative to the Earth's center. The Euler pole is calculated through the location $K$ and the orientation of the first driving torque $\tau$. The speed $u_s$ can be further decomposed into the longitudinal speed $u_{s-lo}$ and latitudinal speed $u_{s-la}$, and $u_{s-lo}=u_s\sin(\lambda_s-90°)$, $u_{s-la}=u_s\cos(\lambda_s-90°)$, where $\lambda_s$ is the azimuth of arc *SE* with respect to latitude. The azimuth of the movement is then calculated through the longitudinal and latitudinal speeds. All these angles and distances (i.e., $\gamma_i$, $\lambda_s$, $\varphi_k$, $\varphi_s$) may be calculated through the latitudes and longitudes of related locations.



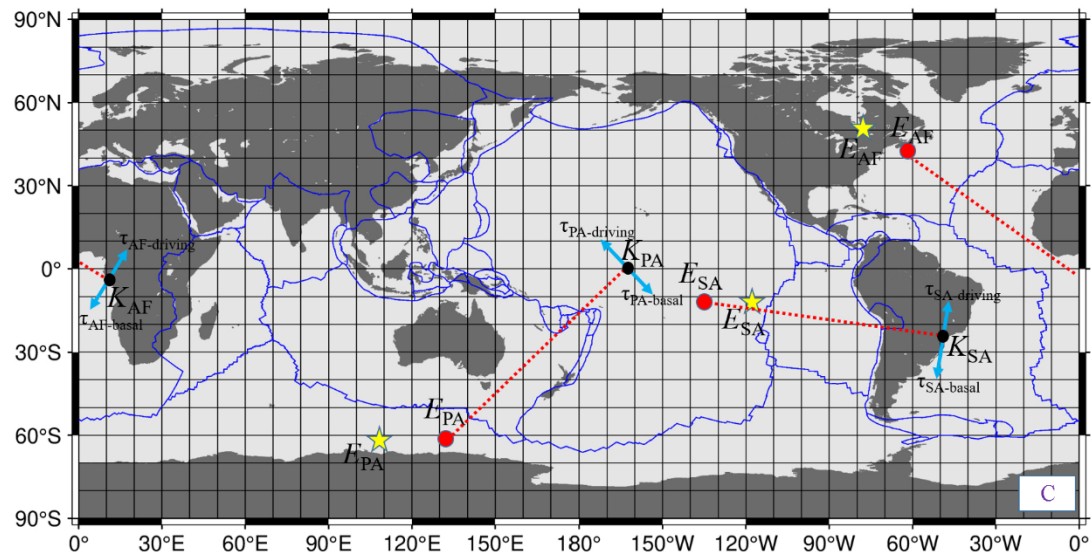

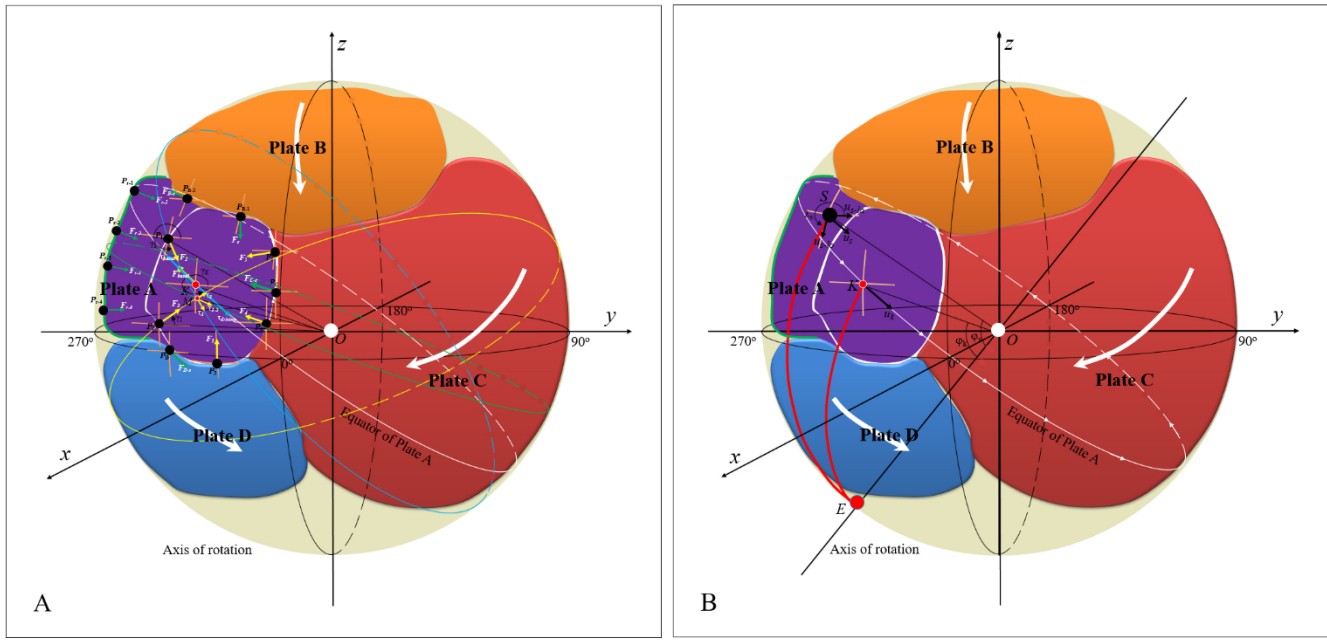


**Figure 8: Modeling the torque balances for plate motion. (A) Geometry of horizontal forces, ridge push forces, collisional forces, and shearing forces over a spherical frame.** The large light blue and yellow circles denote the orientations of the torques yielded by the horizontal force $F_1$ and $F_2$, the large green circle denotes the orientation of the combined torque of these two torques, and the large white circle denotes the possible orientation of the first driving torque. **(B) Decomposing**

**the average movement of the plate into the movement of any location.** The location $K$ and $E$ are the geometric center of



the plate and its Euler pole, respectively. **(C) Exhibiting the torque balance of the selected plates over a planar frame.** The calculated Euler pole locations (red dots) are compared to the established Euler pole locations (yellow stars).

We here use three plates (South American, African, and Pacific plates) to demonstrate the resultant movements. The geometric centers of these plates, the calculated Euler pole location, and the established Euler pole location cited from GSRM v.2.1 (Kreemer et al., 2014) are exhibited in Figure 8(C). A few other possible forces (i.e., collisional and shearing) are considered for these plates. For example, the Eurasian Plate provides a collisional driving force $F_{EU\text{-}AF\text{-}C}$ and a shearing driving force $F_{EU\text{-}AF\text{-}S}$ for the African Plate. The Nazca plate provides a collisional driving force $F_{Naz\text{-}SA\text{-}C}$ for the South American Plate. The Australian, North American, and Eurasian plates provide the collisional driving force $F_{AU\text{-}PA\text{-}C}$, $F_{NA\text{-}PA\text{-}C}$, and $F_{EU\text{-}PA\text{-}C}$ for the

Pacific Plate, respectively. Again, we neglect slab pull. And if this force can be confirmed in the future, it may be added into this model.

  The details of these forces have been exhibited in Figure 4 and listed in Table 1. The resultant torques from all related forces are listed in Table 4. The viscosity and thickness of the asthenosphere for these three plates are the same as that listed in the method I. The other parameters (i.e., plate area, the ratio of the net driving torque to the first driving torque, and the

amplification coefficient) and the resultant average movements are listed in Table 5.

  The movements of 215 locations (41 for the South American Plate, 70 for the African Plate, and 104 for the Pacific Plate) are reproduced by GSRM v.2.1. The calculated and reproduced movements are compared in Figure 9. We find that the calculated movements for these locations are basically consistent with the reproduced movements in both speed and azimuth, the RMS of the calculated speed against the reproduced speed for the South American, African, and Pacific plates is 0.98, 3.18, and 6.51

mm/yr, respectively. This result is not as good as that demonstrated in the method I. One major cause for this is that, in the method II the collisional and/or shearing forces considered are not enough. For example, the Australian, North American, and Eurasian plates collide the Pacific Plate extensively, the three collisional forces $F_{AU\text{-}PA\text{-}C}$, $F_{NA\text{-}PA\text{-}C}$, and $F_{EU\text{-}PA\text{-}C}$ are too spare relative to the long collisional zone; In addition, there may be shearing force between the North American Plate and the Pacific Plate, but we omit this force. As a result, the first driving torque that we calculate is not too accurate in both magnitude and

orientation; Another cause for this is that the geometric center of a plate is strictly not calculated through the average of the latitudes and longitudes of those nodes. The less accurate first driving torque adds to the less accurate geometric center, naturally, the calculated Euler pole location and the resultant movement of the plate cannot be accurate. Even so, our goal is realized that a combination of the ocean-generated force, the ridge push force, the collisional force, and the shearing force indeed may be responsible for plate motion.






**Table 4 (A). Parameters for the torques of three selected plates in the method II**

| Plate | Eule pole $E$ Latitude | Longitude | No. $i$ | Horizontal force $F_i$ N ($*10^{17}$) | Earth's radius $R_{earth}$ m ($*10^3$) | Torque $\tau_i$ N·m ($*10^{23}$) | No. $j$ | Combined torque $\tau_j$ N·m ($*10^{23}$) | Inclination to latitude, east ($\gamma_j$) Degrees |
|---|---|---|---|---|---|---|---|---|---|
| | | | 31 | 1.4573 | 6371.00 | 9.2848 | 31 | 9.2815 | 22.92° |
| | | | 32 | 1.2059 | 6371.00 | 7.6829 | 31-32 | 16.8927 | 27.75° |
| | | | 33 | 1.8014 | 6371.00 | 11.4769 | 31-33 | 26.8242 | 12.22° |
| | | | 34 | 1.2737 | 6371.00 | 8.1145 | 31-34 | 34.5354 | 7.29° |
| | | | 35 | 0.9484 | 6371.00 | 6.0420 | 31-35 | 39.7562 | 12.02° |
| | | | 36 | 0.1551 | 6371.00 | 0.9883 | 31-36 | 39.7462 | 13.45° |
| | | | 37 | 0.1161 | 6371.00 | 0.7397 | 31-37 | 39.1661 | 14.12° |
| South America | -14.68° | 224.18° | 38 | 1.7897 | 6371.00 | 11.4020 | 31-38 | 34.9655 | 30.59° |
| | | | 39 | 2.0480 | 6371.00 | 13.0480 | 31-39 | 31.7450 | 52.47° |
| | | | 40 | 1.4398 | 6371.00 | 9.1732 | 31-40 | 35.3756 | 66.91° |
| | | | 41 | 1.6295 | 6371.00 | 10.3814 | 31-41 | 36.7958 | 83.30° |
| | | | 42 | 1.5804 | 6371.00 | 10.0688 | 31-42 | 27.5043 | 90.28° |
| | | | 43 | 1.3853 | 6371.00 | 8.8257 | 31-43 | 21.2133 | 105.00° |
| | | | 44 | 0.3300 | 6371.00 | 2.1025 | 31-44 | 21.1033 | 110.69° |
| | | | 45 | 1.7285 | 6371.00 | 11.0123 | 31-45 | 10.0916 | 109.90° |
| | | | NA-SA-c | 0.2000 | 6371.00 | 4.4597 | 31-NA-SA-C | 9.5515 | 83.84° |






**Table 4 (B). Parameters for the torques of three selected plates in the method II (continued)**

| Plate | Eule pole $E$ | | No. $i$ | Horizontal force $F_i$ N ($*10^{17}$) | Earth's radius $R_{earth}$ m ($*10^3$) | Torque $\tau_i$ N·m ($*10^{23}$) | No. $j$ | Combined torque $\tau_j$ N·m ($*10^{23}$) | Inclination to latitude, east ($\gamma_j$) Degrees |
|---|---|---|---|---|---|---|---|---|---|
| | Latitude | Longitude | | | | | | | |
| Africa | 44.43° | 298.60° | 46 | 1.1123 | 6371.00 | 7.0867 | 46 | 7.0867 | 320.52° |
| | | | 47 | 1.7917 | 6371.00 | 11.4148 | 46-47 | 18.1425 | 334.91° |
| | | | 48 | 1.6739 | 6371.00 | 10.6648 | 46-48 | 24.8049 | 357.55° |
| | | | 49 | 1.7514 | 6371.00 | 11.1579 | 46-49 | 26.1494 | 22.67° |
| | | | 50 | 0.9317 | 6371.00 | 5.9357 | 46-50 | 32.0830 | 22.36° |
| | | | 51 | 1.5581 | 6371.00 | 9.9269 | 46-51 | 41.9838 | 23.48° |
| | | | 52 | 1.3150 | 6371.00 | 8.3777 | 46-52 | 48.8835 | 17.47° |
| | | | 53 | 1.4180 | 6371.00 | 9.0339 | 46-53 | 57.9100 | 17.86° |
| | | | 54 | 0.9546 | 6371.00 | 6.0816 | 46-54 | 63.9220 | 18.73° |
| | | | 55 | 0.2953 | 6371.00 | 1.8813 | 46-55 | 64.7399 | 20.24° |
| | | | 56 | 1.0480 | 6371.00 | 6.6768 | 46-56 | 60.9224 | 25.24° |
| | | | 57 | 1.1109 | 6371.00 | 7.0773 | 46-57 | 57.7443 | 31.35° |
| | | | 58 | 0.6026 | 6371.00 | 3.8393 | 46-58 | 54.2170 | 32.90° |
| | | | 59 | 0.9126 | 6371.00 | 5.8140 | 46-59 | 52.4030 | 38.84° |
| | | | 60 | 1.5344 | 6371.00 | 9.7756 | 46-60 | 50.6747 | 49.55° |
| | | | EU-AF-C | 4.2000 | 6371.00 | 26.7582 | 46-EU-AF-C | 26.1669 | 66.52° |
| | | | EU-AF-S | 0.8000 | 6371.00 | 5.0968 | 46-EU-AF-S | 27.8008 | 56.25° |
| Pacific | -65.25° | 136.77° | NA-PA-C | 3.0000 | 6371.00 | 19.1130 | NA-PA-C | 19.1130 | 185.00° |
| | | | *AU-PA-C | 2.2000 | 6371.00 | 14.0162 | NA-PA-C-AU-PA-C | 14.7433 | 138.26° |
| | | | EU-PA-C | 0.1000 | 6371.00 | 0.6371 | NA-PA-C-EU-PA-C | 14.2742 | 136.56° |




Note: * denotes this force is changed from $1.0 \times 10^{17}$ N listed in Table 1 to $2.2 \times 10^{17}$ N.


**Table 5. The net driving torques and their resultant movements for three selected plates in the method II**

| Plate | Area | Ratio | Net driving torque | Geometric center of the plate | | | Earth's radius | Amplification coefficient | Movement |
|---|---|---|---|---|---|---|---|---|---|
| | A | | $\tau_{driving}$ | $K$ | | $\varphi_k$ | $R_{earth}$ | | u |
| | km2 | $\varepsilon$ | N·m $(10^{23})$ | Latitude | Longitude | Degrees | m $(10^3)$ | $\zeta$ | mm/yr |
| South America | 43,600,000 | 0.30 | 2.8640 | -24.39° | 313.66° | 83.84° | 6371.00 | 1.20 | 11.71 |
| Africa | 61,300,000 | 0.40 | 11.1202 | -5.57° | 13.43° | 56.25° | 6371.00 | 1.20 | 32.33 |
| Pacific | 103,300,000 | 0.35 | 4.9960 | 0.10° | 198.65° | 136.56° | 6371.00 | 1.20 | 71.82 |






**Figure 9: The reproduced movements from GSRM v.2.1 (black arrows) verse the calculated movements from our model (red arrows) in the method II.** a), b), and c) are the South American, African, and African plates, respectively.



### 3.3 Resultant stress

As mentioned in section 2.2, the observed stresses are mostly concentrated on the uppermost part of the lithosphere (Zoback, 1992; Zoback et al., 1989; Zoback & Magee, 1991), whereas our modeling analysis suggests that the stress caused by the

existing forces (i.e., the ridge push, basal friction, and collisional) are mainly concentrated on the lower part of the lithosphere. This discrepancy indicates that other force may be responsible for the observed stresses. Ocean water is loaded on the top of the lithosphere, this allows to create a stress field associated with the upper part of the lithosphere.

To examine this expectation, we add the ocean-generated force onto the model that is exhibited in Figure 2 (top). The final model is shown in Figure 10 (top left). The inputs include the vertical pressure caused by the rock's weight and the lateral

pressures caused by these loads (i.e., $F_{RW}$, $F_{LW}$, $F_{RP}$, $F_b$, and $F_c$). $F_{RW}$ and $F_{LW}$ are the ocean-generated forces, they correspond to a result of 5 km water depth at the right and 3 km water depth at the left, respectively, and $F_{RW}=0.12\times10^{12}$ N m$^{-1}$, $F_{LW}=0.04\times10^{12}$ N m$^{-1}$. The outputs include the stress produced by the vertical pressure alone and the stress produced by a combination of the vertical and lateral pressures. Similarly, we will only discuss the horizontal stress (i.e., S11) in the following section. At this time, we first use these loads to yield Set A data and Set B data. The stress clouds of the area *GHIJ* are compared

in Figure 10 (middle left). To realize a visual impression, we magnify these loads 50 times, which yields Set A' data and Set B' data. We find that the horizontal stress caused by these loads is compressional and tends to distribute across the middle part of section *GHIJ*. We then minify $F_{RP}$ and $F_b$ 100 times, remain $F_{RW}$ and $F_{LW}$ stable, and adjust $F_c$ properly so as to sustain the horizontal force balance, this yields Set C data and Set D data. To realize a visual impression, we again magnify these revised loads 50 times, which yields Set C' data and Set D' data. A more detailed description of these loads for different sets is

exhibited in Figure 10 (top right). It can be found that, after $F_{RP}$ and $F_b$ are reduced, the horizontal stress caused by these loads are mainly concentrated on the upper part of section *GHIJ*.

The stress diagrams for three sections (i.e., M$_1$N$_1$, M$_2$N$_2$, and M$_3$N$_3$) are also collected and compared in Figure 11. When we subtract the stress caused by the rock's weight from the stress caused by the rock's weight and these loads (i.e., $F_{RW}$, $F_{LW}$, $F_{RP}$, $F_b$, and $F_c$), we obtain the stress caused by these loads, which are expressed with Set A/B/C/D - Set I. We find that the result

of the stress diagrams agrees that of the stress clouds. The continental plates are not only rigid but also curved, this allows the ocean-generated forces (i.e., the horizontal forces) to laterally penetrate across the plate. Our modeling analysis suggests that the stress caused by a combination of the ocean-generated force, ridge push force, collisional force, and basal friction force may be in accordance with the observed stresses.




| No. | Loads (*$10^{12}$ N/m) | | | | |
|---|---|---|---|---|---|
| | $F_{RP}$ | $F_b$ | $F_c$ | $F_{RW}$ | $F_{LW}$ |
| Set A | 4.00 | 3.20 | 0.87 | 0.12 | 0.04 |
| Set B | 4.00 | 2.00 | 2.08 | 0.12 | 0.04 |
| Set C | 0.04 | 0.03 | 0.09 | 0.12 | 0.04 |
| Set D | 0.04 | 0.02 | 0.10 | 0.12 | 0.04 |
| Set A' | 200.00 | 160.00 | 43.74 | 6.13 | 2.21 |
| Set B' | 200.00 | 100.00 | 103.92 | 6.13 | 2.21 |
| Set C' | 2.00 | 1.60 | 4.32 | 6.13 | 2.21 |
| Set D' | 2.00 | 1.00 | 4.92 | 6.13 | 2.21 |


**Figure 10: Stress cloud comparison.** $F_{RW}$ and $F_{LW}$ denote the ocean-generated forces, respectively. $F_{RP}$, $F_b$, and $F_c$ denote the ridge push force, the basal friction force, and the collisional force.







**Figure 11: Stress diagram comparison.**





The lithospheric plates are curved, the rocks within them are not homogeneous and isotropic, and their thickness and density also vary spatially; in addition, as seen in Figure 4, the directions of the ocean-generated force are various. We expect that the stresses caused by a combination of the ocean-generated force and all of these factors may realize a better match with the observed stresses in the WSM, and this will be included in the following research.

## 4 Discussion

### 4.1 Why may ocean-generated force drive plate motion?

Although we have demonstrated that the movements estimated from ocean-generated force are consistent with observations, many people still refuse this force to be a plate driving force for the following reasons: 1) The ocean constitutes just another deviation from the true radial density distribution of the Earth. Any "lateral" density heterogeneity creates stresses that in turn lead to deformation, and their extent is controlled by the rheological properties of the involved materials. 2) Plate motion determines the shape of the ocean basin; as a result, ocean water cannot contribute to plate motion. 3) Ocean loading on top of 670 the lithosphere does not allow ocean-generated force to drive the lithospheric plates to move along the asthenosphere, this is similar to that the water held in a container standing on the ground cannot drive the container to move along the ground. 4) Ocean-generated force is too small to drive plate motion. These issues need to be clarified here. First, the view that any "lateral" density heterogeneity would lead denser materials (i.e., rocks) to flow toward lighter materials (i.e., air or water) is rather idealized. The Himalayas are denser than the surrounding air and water, but this density heterogeneity does not allow the 675 mountains to reduce their height; instead, they increasingly rise up. The continents are also denser than the oceans, and if they flow toward the oceans, the ocean basin will be filled by the continent's rock substances. Then, the sea level would rise, causing water to submerge the coast. This would result in a decrease in the landmass area. Given the continent's volume is constant, then, the continent's height would reduce. This is evidently contrary to the continental accretion that is widely accepted by the geophysical community. The continental accretion indicates that the continents have been growing since the Archean. A further 680 review of this topic can be found in this recent research (Zhu, et al., 2021). The examples of the Himalayas and continents suggest that a system composed of ocean water and the crust is permanently disturbed by the hydrostatic pressure force and tides; As a result, it is difficult for the continents to follow the principle of "lateral" density heterogeneity to flow toward the oceans. It is an opinion of this author that ocean water compresses the crust, the elasticity of the crust's rock allows the crust to deform as a response to the hydrostatic pressure force. With the passage of time, the ocean basin expands gradually, and 685 then the water in the seas flows toward the ocean basin, and the sea level decreases to cause part of the seafloor to expose and become landmass. In addition, the continent's height increases relative to the sea level. This process simply accords with the continental accretion. Indeed, plate motion may reshape ocean basin, but ocean water is not passive, it may provide feedback through energy dissipation on the plate, and as a result, affect plate motion. Ocean loading on the lithosphere is far different from water loading in a container. Since the lithosphere has been broken into individual plates and these plates are attached to





the underlying viscous asthenosphere, this situation allows ocean-generated force to interact with the basal friction exerted by
         the asthenosphere on the plate. In contrast, a container is perfect, and the force produced by water pressure within the container
         is balanced out by the container itself and cannot interact with the basal friction exerted by the ground on the container. In
         physics, the interaction of a driving force and a resistive force is a precondition for an object to move.

         Figure 12 outlines how force balances may be created for the plates. Three plates are totally designed in the model; along the
vertical direction, the weight of each plate is balanced out by the support from the asthenosphere; thus, we will only consider
         the force along the horizontal direction. $F_{AR}$, $F_{AL}$, $F_{CL}$, and $F_{CR}$ are the horizontal forces, $F_{RL}$ and $F_{RR}$ are the ridge push forces,
         $F_{BA}$, $F_{AB}$, $F_{CB}$, and $F_{BC}$ are the collisional forces, $f_A$, $f_B$, and $f_C$ are the basal friction forces. Slab pull and trench suction are
         neglected. Each of these forces can yield a torque relative to the Earth's center, because torque is a product of force and lever
         arm, and here the lever arm may be represented with the Earth's radius since plate is too thin relative to the Earth's radius, the
lever arm length of one plate is approximately equal to that of another plate. This situation allows to simplify torque balance
         into force balance for the following discussion. We assume that Plate A rotates counterclockwise and Plate C rotates clockwise.
         For Plate A, we set $F_{RL}= 4.0\times10^{12}$ N m$^{-1}$, this magnitude is presently accepted by geophysical community (Turcotte and
         Schubert, 2004). We assume the ocean depth to be 5.00 km at the right and 3.00 km at the left, respectively. These two depths
         correspond to $F_{AR}= 0.245\times10^{12}$ N m$^{-1}$ and $F_{AL}= 0.0882\times10^{12}$ N m$^{-1}$, the final horizontal force of these two forces would be
$F_{AR}$-$F_{AL} = 0.1568\times10^{12}$ N m$^{-1}$. We set $F_{BA} = 4.05\times10^{12}$ N m$^{-1}$, and use $F_{RL} = 4.0\times10^{12}$ N m$^{-1}$ and a little portion of the final
         horizontal force, which is represented by $F_{ARL} = 0.05\times10^{12}$ N m$^{-1}$, to balance out $F_{BA}$, and use the remaining final horizontal
         force $F_{RARL}= 0.1068\times10^{12}$ N m$^{-1}$ to balance out the basal friction force $f_A$. The force balances for this plate would be $F_{BA}$- $F_{RL}$-
         $F_{ARL}= 0$ and $F_{RARL}$-$f_A$=0.

         For Plate B, we set $F_{CB}=4.0\times10^{12}$ N m$^{-1}$, due to $F_{BA}= F_{AB} = 4.05\times10^{12}$ N m$^{-1}$, thus, $F_{BA}$- $F_{CB} =0.05\times10^{12}$ N m$^{-1}$. We use this net
force to balance out the basal friction force $f_B$. The force balance for this plate would be $F_{BA}$- $F_{CB}$ -$f_B$=0.

         For Plate C, we set $F_{RR}= 3.95\times10^{12}$ N m$^{-1}$, and assume the ocean depth to be 4 km at the left and 6 km at the right, respectively.
         These two depths correspond to $F_{CR}= 0.1568\times10^{12}$ N m$^{-1}$ and $F_{CL} = 0.3528\times10^{12}$ N m$^{-1}$, the final horizontal force of these two
         forces would be $F_{CL}$ - $F_{CR}=0.196\times10^{12}$ N m$^{-1}$. Due to $F_{CB} = F_{BC} =4.0\times10^{12}$ N m$^{-1}$, we set $F_{RR}= 3.95\times10^{12}$ Nm$^{-1}$ and use a little
         portion of the final horizontal force, which is represented by $F_{CLR} = 0.05\times10^{12}$ N m$^{-1}$, to balance out $F_{BC}$, and use the remaining
final horizontal force $F_{RCLR} = 0.146\times10^{12}$ N m$^{-1}$ to balance out the basal friction force $f_C$. The force balances for this plate
         would be $F_{BC}$- $F_{RR}$-$F_{CLR} = 0$ and $F_{RCLR}$-$f_C$=0.

         These force balances allow three plates to be steadily rotated. We find, even if the ridge push force $F_{RL}$ ($F_{RR}$) is given a smaller
         amplitude (~ $10^{10}$ N m$^{-1}$, for example), so long as the collisional force $F_{BA}$ ($F_{AB}$, $F_{CB}$, and $F_{BC}$) is properly valued, these force
         balances can always be created. Nevertheless, as demonstrated in section 3.3, a ridge push force of $4.0\times10^{12}$ N m$^{-1}$ would result
in a horizontal stress that is mostly concentrated on the lower part of the lithosphere, which is not in accordance with
         observation. Hence, we prefer to accept the ridge push force to be smaller than ocean-generated force.





**Figure 12: Modeling the dynamics of the lithospheric plates.** Blue passages denote water compensation from one ocean to another. Note that the ocean depth, tide, plate's thickness, mantle, and core are highly exaggerated.



## 4.2 How does plate motion realize mechanically?

Thus far, we have concluded that ocean-generated force is able to combine the ridge push force, the collisional force, and the

shearing force to satisfy the kinematics and geometry of plate motion. Now, let us discuss how plate motion can be mechanically realized. As shown in Figure 12, it is assumed that the depth of Ocean 1 is greater than that of Ocean 2. If we use a part of Ocean 2 that connects to Plate A, which is equal in length to Ocean 1, to do comparison, the depth difference between this part of Ocean 2 and Ocean 1 creates a net gravitational potential energy relative to the asthenosphere reference level. As Plate A and Plate B move away from each other, this separation would require the Ocean 1 depth to decrease as the

basin elongates horizontally, and require the Ocean 2 depth to increase as the basin shortens horizontally. Consequently, the net gravitational potential energy decreases. Therefore, if there were no external energy inputs to compensate, the net gravitational potential energy would eventually disappear, terminating plate motion. Tides may be supplying this energy. Tides represent the regular alternations of high and low water on Earth; when high water falls, the gravitational potential energy converts into kinetic energy, then, ocean water obtains movement. As all oceans are physically connected, part of the water in

Ocean 2 may travel via passages to compensate the decreasing ocean depth of Ocean 1, thus sustaining the net gravitational potential energy. Given the basal friction force $f_{basal} = 1.62 \times 10^{18}$ N and the movement distance u=3 cm/yr for the lithosphere, an energy of $Q_1 = f_{basal} \times u = 4.86 \times 10^{16}$ J/yr is required to satisfy this movement distance. This energy also represents the net gravitational potential energy. The ocean water level often increases twice a day due to tides, and the resultant height is assumed to be h=0.3 m. Given the gravitational acceleration g=9.8 m/s, the volume $v=1.35 \times 10^9$ km$^3$ and density ρ=1000 kg/m$^3$

for the whole ocean, and consequently, the gravitational potential energy obtained by ocean water due to tides during a year would be $Q_2 = 2*365*\rho v g h = 2.9 \times 10^{21}$ J/yr. The transformation from gravitational potential energy to kinetic energy within ocean water and the energy transition between oceans must be complicated, and we believe that a small part of this tidal energy is enough to supply the net gravitational potential energy. In fact, the impact of tidal energy on plate motion has long been discussed. Wegener (1924) proposed that tides cause a slight progressive displacement of the crust. Rochester (1973) showed

that the total energy released due to tidal friction exceeds $5*10^{19}$ ergs/s. Several authors (e.g., Miller, 1966; Munk, 1968) concluded that the dissipation in both shallow seas and on the solid Earth is approximately $2*10^{19}$ ergs/s, and this amount of energy exceeds the lower bound set by seismic energy release by 2 orders of magnitude (Gutenberg, 1956) and might be driving the plate motion. Other authors (e.g., Riguzzi et al., 2010; Egbert and Ray, 2000) reevaluated the energy budget and found that the total energy released by tidal friction may reach up to $1.2*10^{20}$J/yr, and approximately $0.8*10^{20}$J/yr is dissipated

in the oceans, shallow seas, and mantle, and the remaining energy is enough to maintain the lithosphere's rotation, estimated at approximately $1.27*10^{19}$J/yr. In contrast to these studies, we provide another insight: the tidal energy obtained by ocean water may feed plate motion.





## 4.3 Does the tidal forcing relate to plate motion or seismicity?

### 4.3.1 Tidal force versus plate motion

The impact of tidal drag on plate motion has been debated for many years. Wegener (1915) attributed the continent's drift to tidal drag and centrifugal forces, but these forces were shortly found to be too weak to work. Jeffreys (1929) claimed that the mean tidal friction corresponds to a westward stress of the order of only 10-4 dyn/cm$^2$ over the earth's surface, this stress is too small to maintain the drift. The notion of tidal drag revived after the discovery of a net rotation or westward drift of the lithosphere relative to the mantle (Le Pichon, 1968; Knopoff and Leeds, 1972). Another argument in favor of this notion stems

from the assessment of energy budget, as discussed in section 4.2, it shows that tides are energetically enough for feeding plate motion. However, a satisfaction in energy cannot shield the notion of tidal drag anymore. Jordan (1974) and Jeffreys (1975) attached the theoretical basis of tidal drag, they claimed that the viscosity both related to tidal drag and necessary to allow decoupling between lithosphere and mantle (~$10^{11}$ Pas) is far less than the present-day asthenosphere viscosity. Ranalli (2000) also showed that any non-zero torque due to difference in angular velocity between the mantle shell and lithosphere shell

would be extremely transient, and cannot be a factor in the origin of the westward drift of the lithosphere. Despite these fierce objections, the advocates of tidal drag didn't give up. Scoppola et al. (2006) proposed the westward rotation of the lithosphere as a consequence of the combined effect of tidal torque, downwelling of the denser material into the mantle, and thin layers of very low viscosity hydrate channels in the asthenosphere. Several authors (e.g., Riguzzi et al., 2012; Doglioni and Panza, 2015) had suggested that, if an ultra-low viscosity layer exists in the upper asthenosphere, the horizontal component of the tidal

oscillation and torque may be able to move the lithosphere. As demonstrated in section 3.2, laboratory experiments tend to support this possibility (Bercovici et al., 2015; Mei et al., 2002; Hirth and Kohlstedt, 1996; Scoppola et al., 2006; Doglioni et al., 2011). The asthenosphere's effective viscosity can be lowered to $10^{15}$ Pas if the water content in the case of both diffusion and dislocation creep is included (Korenaga and Karato, 2008). A "superweak", low-viscosity asthenosphere is being accepted by the geophysical community (Kawakatsu et al., 2009; Hawley et al., 2016; Holtzman, 2016; Naif et al., 2013; Freed et al.,

2017; Hu et al., 2016; Stern et al., 2015; Bercker, 2017). Zaccagino et al. (2020) recently investigated a 20-year series of plate motion to conclude that plate motion relates to tidal drag in some way. For example, the lithospheric plates retain a non-zero horizontal component of the solid Earth tidal waves, and they move faster with frequencies of 8.8 and 18.6 years that correlate to lunar apsides migration and nodal precession.

We provide a few points to respond to the notion of tidal drag. First of all, the lithosphere's net rotation or westward drift is

different from plate motion. The former indicates that the lithosphere is moving in a single direction, while the latter indicates that the lithospheric plates are moving in different directions. Second, it is already established in the astronomical field that tidal drag is operated through the tractive force. This force is geometrically decomposed from a tide-generating force, and its direction always follows the Earth's surface. Apparently, this force may be divided into two symmetric fields that are aligned with the Earth-Moon system. In each field the force vector is uniformly directed to the sub-lunar points, which are projections



of the Moon on the Earth's surface. Two patterns are expected for the lithospheric plates under the tractive force. One is that, as the Earth rotates around its axis, all the plates are continuously swept from east to west. Another is that parts of the lithosphere, which are located at middle and high latitudes, would be dragged toward lower latitudes. Nevertheless, upon comparison with the plate motion vector (Figure 13), it becomes evident that the tractive force is not in accordance with the plate motion. The Pacific Plate, for instance, moves northwest, the Eurasian Plate rotates clockwise, the North American Plate

rotates counterclockwise, the African and Indian-Australian Plates move northeast. Moreover, the movements of the Pacific, Eurasian, and Indian-Australian plates intersect with each other. The diversity of plate motion implies that each plate is being operated by a set of independent forces, with a leading force controlling the direction of plate motion. Third, it has been found that the plates performed a cycle of dispersal and aggregation during a geological timescale, and that three supercontinents (i.e., Pangaea, Rodinia and Columbia) occurred over the past 2 billion years (Mitchell et al., 2021). This cycle of dispersal and

aggregation is a manifestation of plate motion. When projected on the Earth's surface, the Moon's position is mostly between 18° N and 18° S (Pugh and Woodworth, 2014), which means that the tractive force on the Earth's surface remains relatively steady. Consequently, a relatively steady tractive force is not compatible with a secular cycle of plate motions. Last, we agree with Zaccagino et al. (2020) that tidal drag contributes to plate tectonics. Although there is a difference in tidal amplitude/direction between the plates, tidal drag permanently imparts energy into the system of plates, thereby allows for a

modulation of plate tectonics.



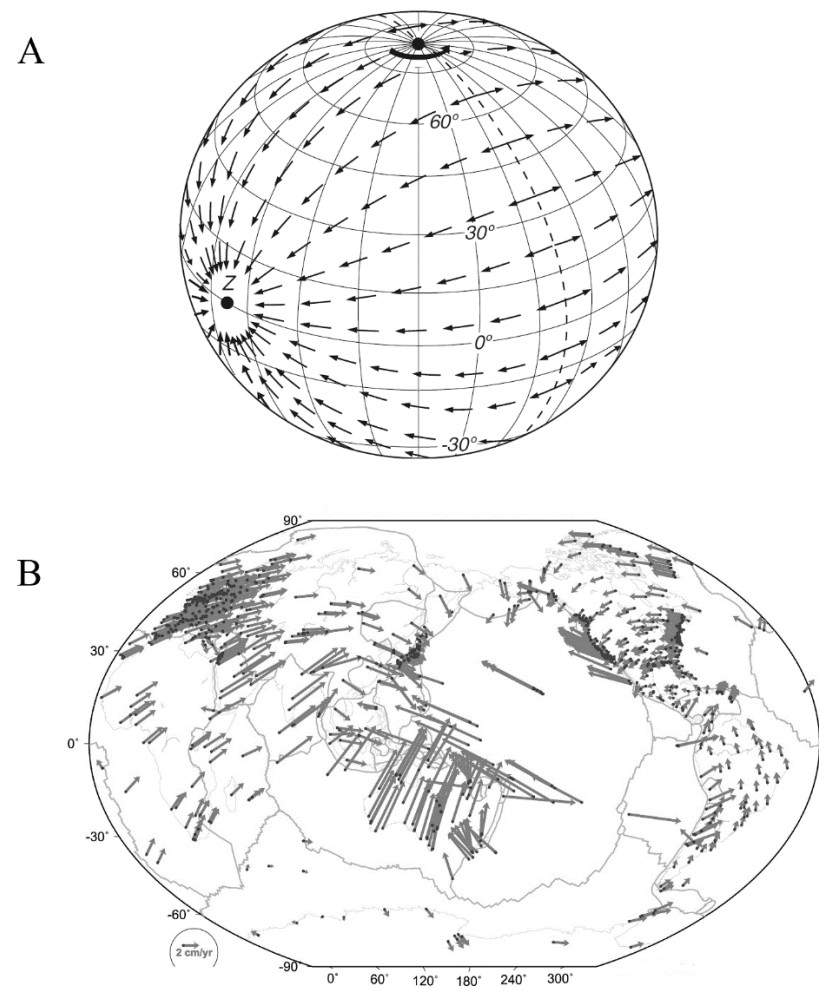

**Figure 13: The tidal tractive force (A) versus plate motion (B).** The tractive force on Earth is the situation when the Moon is above the Equator at Z (Robert, 2008). ITRF2014 horizontal velocity field of plate motion is from Altamimin et al. (2016).

### 4.3.2 Tidal force versus seismicity

Whether the tidal forcing relates to earthquake occurrence is a considerably hot topic. Most studies with global earthquake catalogs tend to show no correlation between the two (e.g., Schuster, 1897; Morgan et al., 1961; Hartzell and Heaton, 1989). Some of regional earthquake catalogs reveal a significant correlation (e,g., Young and Zurn, 1979; Ulbrich et al., 1987; Shirley, 1988), but others also revealed no correlation (Knopoff, 1964; Shlien, 1972; Shudde and Barr, 1977; Vidale et al., 1998). Nevertheless, these studies explored only the effect that is caused by solid Earth tide, the effect caused by ocean tide

(i.e., the loading) is commonly ignored. By adding ocean tide to solid Earth tide, Tsuruoka et al. (1995) reached a point that a decrease in the confining pressure due to the tidal forcing is responsible for triggering earthquake occurrence. Tanaka et al.



(2002) expanded the method taken by Tsuruoka et al. (1995), they investigated 9350 globally distributed earthquakes with magnitude 5.5 or larger to conclude that a small stress change due to the tidal forcing encourages earthquake occurrence. The results of these studies (e.g., Tsuruoka et al., 1995; Tanaka et al., 2002) suggest that ocean tide plays a crucial role in

determining the correlation.

If ocean tide really relates to earthquake occurrence, then it must depend on ocean water exerting its impact. For example, when a tide is added to the ocean, the ocean water depth will vary, the ocean water pressure will also vary. Ocean water pressure variation is not only vertically applied to the oceanic crust that is below the ocean, but also horizontally applied to the continental crust that connects to the ocean. Guillas et al. (2010) presented a link between the El Niño-Southern

Oscillation (ENSO) and earthquake occurrences on the East Pacific Rise (EPR), and proposed that a reduction in ocean-bottom pressure over the EPR may encourage seismicity. A recent study found that sea level changes affect seismicity rates in a hydrothermal system near Istanbul (Martínez-Garzón et al., 2023). As pointed out by Tanaka et al. (2002), the most likely component to control the earthquake occurrence is the stress. Since tide represents a periodic oscillation, it should be expected that the stress produced by ocean water will behave periodically. In these studies (e.g., Tsuruoka et al., 1995;

Tanaka et al., 2002; Martínez-Garzón et al., 2023), the tidal stress is theoretically estimated, but there is no modelling or experimental evidence for the tidal stress. In section 3.3, we have modelled the crust's stress through a combination of various forces (i.e., the ocean-generated force, the ridge push force, the collisional force, and the basal drag force), in which the ocean-generated force is constant. Below, we explore the stress when the ocean-generated force varies due to tide, in order to provide support for past and future studies.

The model (Figure 14) is based on that of Figure 10 (top left). The inputs include the hourly vertical pressure caused by the rock's weight and the hourly lateral pressures caused by these loads (i.e., $F_{RW}$, $F_{LW}$, $F_{RP}$, $F_b$, and $F_c$). Note, the solid body tide is neglected. $F_{RW}$ and $F_{LW}$ are the ocean-generated forces that assimilate the effect of tides. We here design a water level variation of totally 12 hours, which corresponds to a semidiurnal tide. The information of tidal height and loads is listed in Table 6. The outputs include the hourly stress produced by the vertical pressure alone and the hourly stress produced by a

combination of the vertical and lateral pressures. Using the latter to subtract the former, we obtain the hourly stress produced by the lateral pressures. Similarly, we only discuss the horizontal stress (i.e., S11). At this time, we collect the results of 6 locations (i.e., ①, ②, ③, ④, ⑤, and ⑥) to do comparison. These locations belong to the 30 km depth and 60 km depth of three sections (i.e., $M_1N_1$, $M_2N_2$, and $M_3N_3$).

The stress diagrams for these locations are compared in Figure 15. We find that the stress oscillation due to ocean tide

has laterally penetrated the crust's rock. This study has not yet considered the stress in the oceanic crust, but the result is expected. For example, ocean water exerts pressure on the oceanic crust, which produces stress for the oceanic crust; when tide is added to the ocean, the stress is mechanically entrained to oscillate. As depicted in Set C(D) - Set I of Figure 11, the stress generated over a depth of 50 km is approximately 2.0 to 6.0 Mpa. This magnitude has fallen within the range of earthquake stress drops (1~30 Mpa) (Kanamori, 1994), indicating that the ocean-generated force may closely relate to



earthquake occurrence. Please be aware of that our model is straight, the rock's materials within it are assumed to be homogeneous and isotropic. In practice, the Earth's surface is curved, the rock's materials are not only inhomogeneous but also anisotropic. The oceans circle the continents, which leads to the continents being laterally compressed inward. All of these factors allow the ocean water pressure to be amplified in the crust, resulting in a higher stress level.

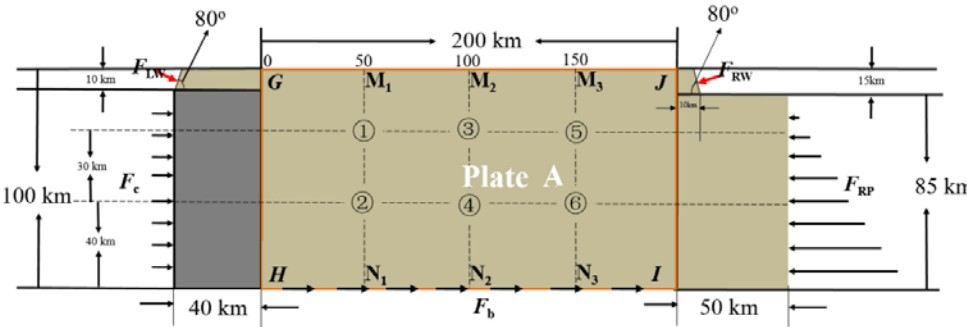


**Figure 14: Model for oscillating ocean water.**

**Table 6. Information of tidal height and loads**

| Time (h) | Tidal height (m) | Loads ($*10^{11}$N/m) | | | | |
|---|---|---|---|---|---|---|
| | | $F_{LW}$ | $F_{RW}$ | $F_{RP}$ | $F_b$ | $F_c$ |
| 0 | 0.0 | 0.4410000000 | 1.2250000000 | 0.4000000000 | 0.2000000000 | 1.0000000000 |
| 1 | 0.4 | 0.4411176078 | 1.2251960078 | 0.4000000000 | 0.2000000000 | 1.0000000000 |
| 2 | 0.8 | 0.4412352314 | 1.2253920314 | 0.4000000000 | 0.2000000000 | 1.0000000000 |
| 3 | 1.2 | 0.4413528706 | 1.2255880706 | 0.4000000000 | 0.2000000000 | 1.0000000000 |
| 4 | 0.8 | 0.4412352314 | 1.2253920314 | 0.4000000000 | 0.2000000000 | 1.0000000000 |
| 5 | 0.4 | 0.4411176078 | 1.2251960078 | 0.4000000000 | 0.2000000000 | 1.0000000000 |
| 6 | 0.0 | 0.4410000000 | 1.2250000000 | 0.4000000000 | 0.2000000000 | 1.0000000000 |
| 7 | -0.4 | 0.4408824078 | 1.2248040078 | 0.4000000000 | 0.2000000000 | 1.0000000000 |
| 8 | -0.8 | 0.4407648314 | 1.2246080314 | 0.4000000000 | 0.2000000000 | 1.0000000000 |
| 9 | -1.2 | 0.4406472706 | 1.2244120706 | 0.4000000000 | 0.2000000000 | 1.0000000000 |
| 10 | -0.8 | 0.4407648314 | 1.2246080314 | 0.4000000000 | 0.2000000000 | 1.0000000000 |
| 11 | -0.4 | 0.4408824078 | 1.2248040078 | 0.4000000000 | 0.2000000000 | 1.0000000000 |




**Figure 15: The stress variation out to time.** Note, the stress variations in A and B are too small to be perceptible.



### 4.4 How may plate motion initiate and proceed?

The dispersal and aggregation of plates represent that the ocean basin had been periodically adjusted, and this change is often called the Wilson Cycle (Wilson, 1963). Figure 16 outlines how such a cycle may be realized. It is assumed that the left end of the model is connected to its right end and that the depth of Ocean 1 depth is greater than that of Ocean 2. The greater ocean depth corresponds to greater ocean-generated force. Slab pull and trench suction are neglected. The ocean-generated force, the collisional force, and the basal friction force combine to form force balances. For instance, the force balance for Plate A is

$F_{AL}$-$F_{AR}$-$F_{CA}$-$f_A$=0, the force balance for Plate B is $F_{BR}$-$F_{BL}$-$F_{CB}$-$f_B$=0, and the force balance for Plate C is $F_{AC}$-$F_{BC}$-$f_C$=0. At time $t_1$ and $t_2$, these force balances allow Plate A and Plate B to move toward each other, Plate C is pushed to move. These movements make Ocean 2 basin shorten while make Ocean 1 basin elongate. Tides make ocean water move periodically, the passages between ocean basins allow water to travel and compensate. At time $t_3$, Plate A and Plate B meet, forming an aggregation. Meanwhile, Plate C sinks and becomes disappeared, Ocean 2 basin closes. Since plate motion stops, the forming

oceanic crusts cannot spread away from the ridge, they gradually accumulate and plug up magma eruptions, and the ridge tends to die. Once the fractures of the lithosphere are repaired, the ocean-generated force cannot interact with the basal friction force, and then, these force balances terminate. After some time, a large asteroid violently collides with the aggregated plate, forming extensive fractures on the plate, and one of the fractures penetrates down to the lower part of the plate. At time $t_4$, the large fracture induces water entry, forming a large body of water that is deeper than Ocean 1. The deeper water body

corresponds to greater ocean-generated force, this force may further expand the fracture. The large fracture also represents a mass loss of the upper part of the lithosphere; the isostasy would require the upper mantle to melt, the lower part of the aggregated plate is apparently broken. At time $t_5$, both the ocean-generated force and the molten material finally cut the plate into Plate D and Plate E. As the left end of the model is connected to its right end, the greater ocean-generated force would require the left part of Plate D to compress the right part of Plate E. Together with the basal friction exerted by the asthenosphere,

the left part of Plate D is eventually detached from the plate, forming one subduction. Similarly, the right part of Plate E is detached from the plate, forming another subduction. These detachments and subductions allow to form a new oceanic plate-Plate F. At this moment, the ocean-generated forces may interact with the basal friction force, some new force balances are created. For instance, the force balance for Plate D is $F_{DR}$-$F_{DL}$-$F_{FD}$-$f_D$=0, the force balance for Plate E is $F_{EL}$-$F_{ER}$-$F_{FE}$-$f_E$=0, the force balance for Plate F is $F_{DF}$-$F_{EF}$-$f_F$=0. These force balances allow Plate D and Plate E to move away from each other. A

new oceanic ridge gradually forms, and the increasing separation between the two plates results in a new Ocean 3 basin.

Ocean depth cannot be stable during a long geological timescale, it may change with the deepening/shallowing of basins. The ocean depth change in turn leads ocean-generated force to vary. We assume that, at time $t_1$, the Ocean 1 depth and the Ocean 2 depth are $h_1$=5,000 km and $h_2$=3,000 km, respectively, the length and width of Plate A are $D$=6,000 km and $L$=2,000 km, the water density is $\rho$=1000 kg/m$^3$, the gravitational acceleration is g=9.8 m/s. Then, the total ocean-generated force for Plate A

would be $F_{total}$= $F_{AL}$-$F_{AR}$=0.5$\rho gL(h_1^2$-$h_2^2)$=1.5680×10$^{17}$ N. We divide this total force into two exerting parts: one, as an opposing force, balances out the collisional force from Plate C, and the other, as a driving force, balances the basal friction



force. We also assume that half of the total force is used to act as the driving force, and then, according to Equation (2) exhibited in section 3, there would be $50\%*F_{total}=F_{driving}=f_{basal}=\mu Su/y$ (where $\mu$, $S$, u, and y are the viscosity of the asthenosphere, the area of Plate A, the speed of Plate A, and the thickness of the asthenosphere, respectively). Given $\mu=10^{18}$ Pas, $S=DL=1.2\times10^{13}$

$m^2$, and y=300 km, we get u=6.18 cm/yr. And now we assume that, at time $t_2$, which has passed 30,000,000 years since time $t_1$, the Ocean 1 depth reduces from 5,000 km to 4,500 km, the Ocean 2 depth increases from 3,000 km to 3,800 km, and that other parameters remain constant, and then, the speed of Plate A would be turned into u=2.24cm/yr. During a period of 30,000,000 years, the rate of Plate A is (6.18-2.24)/30000000=$1.31\times10^{-7}$ cm/yr. We again assume that, at time $t_3$, which has passed 50,000,000 years since time $t_2$, plate motion stops, and then, the speed of Plate A should be u=0.00 cm/yr. During a

period of 50,000,000 years, the rate of Plate A is (2.24-0.00)/50000000=$4.48\times10^{-8}$ cm/yr. Figure 4 and Table 1 show that the ocean-generated forces around a continental plate are various in both magnitude and orientation, the ocean depth change would require these forces to vary. As a result, the final horizontal force varies, this leads plate motion to vary with time. Our calculation of the speed of Plate A suggests that the change of plate motion is considerably slow, thus, plate motion may be treated as near-steady.

Asteroid impacts are frequent events in the solar system, and it is widely believed that the initiation of plate motion relates to large asteroid impacts (Alvarez, et al., 1980; Rampino and Stothers, 1984; Prinn and Fegley, 1987; Marzoli, et al., 1999; Hames, et al., 2000; Condie, 2001; Wan, 2018), but the details of this coupling remain elusive. Our demonstration here provides the first insight into this issue: asteroid impact fractures the lithosphere, initiating plate motion; ocean water yields force to maintain plate motion; and tides provide energy for plate motion. Ocean depth cannot be stable during a long geological

timescale, it may change with the deepening/shallowing of basins. The ocean depth change in turn leads ocean-generated force to vary. We assume that, at time $t_1$, the Ocean 1 depth and the Ocean 2 depth are $h_1$=5,000 km and $h_2$=3,000 km, respectively, the length and width of Plate A are D=6,000 km and L=2,000 km, the water density is $\rho$=1000 kg/m³, the gravitational acceleration is g=9.8 m/s. Then, the total ocean-generated force for Plate A would be $F_{total}= F_{AL}-F_{AR}=0.5\rho gL(h_1^2-h_2^2)=1.5680\times10^{17}$ N. We divide this total force into two exerting parts: one, as an opposing force, balances out the collisional

force from Plate C, and the other, as a driving force, balances the basal friction force. We also assume that half of the total force is used to act as the driving force, and then, according to Equation (2) exhibited in section 3, there would be $50\%*F_{total}=F_{driving}=f_{basal}=\mu Su/y$ (where $\mu$, $S$, u, and y are the viscosity of the asthenosphere, the area of Plate A, the speed of Plate A, and the thickness of the asthenosphere, respectively). Given $\mu=10^{18}$ Pas, $S=DL=1.2\times10^{13}$ m², and y=300 km, we get u=6.18 cm/yr. And now we assume that, at time $t_2$, which has passed 30,000,000 years since time $t_1$, the Ocean 1 depth reduces

from 5,000 km to 4,500 km, the Ocean 2 depth increases from 3,000 km to 3,800 km, and that other parameters remain constant, and then, the speed of Plate A would be turned into u=2.24cm/yr. During a period of 30,000,000 years, the rate of Plate A is (6.18-2.24)/30000000=$1.31\times10^{-7}$ cm/yr. We again assume that, at time $t_3$, which has passed 50,000,000 years since time $t_2$, plate motion stops, and then, the speed of Plate A should be u=0.00 cm/yr. During a period of 50,000,000 years, the rate of Plate A is (2.24-0.00)/50000000=$4.48\times10^{-8}$ cm/yr. Figure 4 and Table 1 show that the ocean-generated forces around a





continental plate are various in both magnitude and orientation, the ocean depth change would require these forces to vary. As a result, the final horizontal force varies, this leads plate motion to vary with time. Our calculation of the speed of Plate A suggests that the change of plate motion is considerably slow, thus, plate motion may be treated as near-steady.

Asteroid impacts are frequent events in the solar system, and it is widely believed that the initiation of plate motion relates to large asteroid impacts (Alvarez, et al., 1980; Rampino and Stothers, 1984; Prinn and Fegley, 1987; Marzoli, et al., 1999; Hames,

et al., 2000; Condie, 2001; Wan, 2018), but the details of this coupling remain elusive. Our demonstration here provides the first insight into this issue: asteroid impact fractures the lithosphere, initiating plate motion; ocean water yields force to maintain plate motion; and tides provide energy for plate motion.





**Figure 16: Modeling the dispersal and aggregation of plates.** Large white arrows denote plate motion, small yellow arrows
do ocean-generated forces, while small blue and green arrows do collisional and basal friction forces. Note that the ocean depth
and the plate's thickness are highly exaggerated.





### 4.5 Why does ocean water contribute to tectonics?

All continents are surrounded by oceans, ocean-generated forces are extensively exerted on the sides of the continents that are fixed on top of the lithospheric plates, and all plates connect to each other; consequently, all the plates may interact with each

other (Figure 17). Under the effect of ocean-generated force, a moving continental plate would ride on an oceanic plate, and the front part of the oceanic plate is forced to subduct, forming a sinking slab. Additionally, a moving plate would move away from another plate, and a gap would form between them. The gap allows magma to erupt, forming an MOR. From this point, the ridge push force may be treated as an auxiliary force for ocean-generated force. Since ocean-generated force is exerted on the continental wall (represented by coastline), and the oceanic crust is extensively connected to the continental crust, this

allows ocean-generated force to be laterally transferred to the oceanic crust, and then, the continental crust drags the oceanic crust to move, causing the plate's boundary to follow the shape of coastline.

Many people are extraordinarily perplexed as to why the Earth owns plate tectonics whereas the Venus does not. Many studies have shown that water provides the right conditions (maintaining a cool surface, for example) for plate tectonics, while the absence of water on Venus prohibits plate formation (Driscoll and Bercovici, 2013; Hilairet et al., 2007; Lenardic et al., 2008;

Korenaga, 2007; Tozer, 1985; Lenardic and Kaula, 1994; Hirth and Kohlstedt, 1996; Landuyt and Bercovici, 2009). Our understanding of ocean water provides a new perspective on this issue: no water on Venus means that there is no contribution of ocean-generated force and no further development of plate tectonics on that planet.







**Figure 17: Global view of the distribution of ocean-generated forces (yellow arrows) and ridge push forces (green**
**arrows).** The supporting tidal data are mainly from the Global Sea Level Observing System (GLOSS) database (Caldwell et al., 2015).

### 4.6 What's the trouble with mantle convection?

Although mantle convection had been given up by most of geophysicists since the late 1990s, this cannot prevent it from
becoming popular. We here restate how mantle convection cannot be realistic. The main problems with this paradigm include: 1) the convection cells proposed to exist in the asthenosphere require strong fitting to plate size. Seismic tomography shows that rising mantle material beneath ridges only extends down 200 to 400 km (Foulger, et al., 2001). This depth gives an upper limitation on the scale of the proposed cells. Most of plates (North American, Eurasian, and Pacific, for instance), however, are very wide, generally more than thousands of kilometers.



2) the movement of a large plate would yield a net mass flux in the asthenosphere to compensate the mass transport in the moving plate, this requires the plate to be treated as an integral part of the circulation (Forsyth & Uyeda, 1975). Richter (1973) employed a model to show that the asthenosphere exerts viscous resistive forces rather than driving forces on the plates, which actually opposes the mantle convection currents to act as the drivers.

3) in the scenario of mantle convection, poloidal motion involves vertical upwellings and downwellings, while toroidal motion

undertakes horizontal rotation (Bercovic, et al., 2015). The generation of torioidal motion requires variable viscosity, but numerous studies of basic 3-D convection with temperature-dependent viscosity had failed to yield the requisite toroidal flow (Bercovic, 1993, 1995b; Cadek et al., 1993; Christensen and Harder, 1991; Stein et al., 2004; Tackley, 1998; Trompert and Hansen, 1998; Weinstein, 1998).

4) mantle convection models are unsuccessful in yielding plate motions, although some of them had yielded plate-like behavior

and mathematically got solution for plate motion velocity by means of a non-Newtonian way, i.e., a balance relationship of buoyancy force and drag force (Bercovici, et al., 2015). Doglioni and Panza (2015) concluded that none of mantle convection models is really able to satisfy the constraints posed by the plate kinematics, the temperature, and the asymmetry of plate boundaries.

5) when mantle convection is treated as the plate driving force, it requires not only the mantle to couple the plates but also the

movement of mantle currents to keep consistent with that of the plates. But reality is not so. The Pacific Plate is the fastest W-ward moving one relative to the mantle and is slipping over an Low Velocity Zone (LVZ) (Doglioni et al., 2005) with low viscosity (Pollitz et al., 1998). Evidently, the Pacific is the most decoupled plate, while mantle convection requires the faster moving plates to be more coupled (higher viscosity) with the mantle. The Hawaii hot spot volcanic chain represents that the underlying mantle is moving E-SE-ward. These authors (Hammond and Toomey, 2003; Doglioni et al., 2003) modeled,

beneath the East Pacific Rise (EPR), an eastward-migrating mantle. The hot spot reference frame remains consistent with the existence of an eastward relative mantle flow beneath the South America plate (Van Hunen, van den Berg, & Vlaar, 2002). A relatively moving eastward mantle flow has been proposed also beneath North America (Silver & Holt, 2002) and beneath the Caribbean plate (Gonzalez, Alvarez, Moreno, & Panza, 2011; Negredo, Jiménez-Munt, & Villasenor, 2004). All these results indicate that the movements of the mantle currents are reverse to that of the plates, the opposed moving mantle currents provide

resistive force rather than driving force for these plates.

6) our modeling analysis suggests that, if mantle convection were considered as a driving force, what it performs is a drag on the plate's base, more like the basal friction force $F_b$ exhibited in Figure 1, and then, its resultant stress must be mainly concentrated on the lowermost part of the plate, which cannot be in accordance with observed stress.

**Competing interests**

The contact author declares no competing interests.



**Acknowledgements**

We express sincere thanks to Jinmin Chen for conducting the vector force analysis and to Bernhard Steinberger, Jeroen van Hunen, Maureen D. Long, and Thorsten Becker for their helpful comments on this research. The author declares that there is no conflict of interest. No funding was provided for this research.

**Availability of data and material**

The movement data of the 121 sample locations are extracted from GSRM v.2.1 (Kreemer et al., 2014) and are available through UNAVCO (https://www.unavco.org/software/geodetic-utilities/plate-motion-calculator/plate-motion-calculator.html). The latitude and longitude of the controlling sites on the continental plates in Figure 4 are determined through the ETOPO1 Global Relief Model (Amante and Eakins, 2009), and the ocean depths are artificially resolved through the

NOAA Bathymetric Data Viewer (https://ngdc.noaa.gov/mgg/global/global.html). The tide data in Figure 14 are from the GLOSS database (Caldwell et al., 2015).

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
