# Peer review of "What Drives Plate Motion?"

_EGUsphere, 2023_

## Author Comment (AC1)

**Author's response to the reviewer's comments**

We (as author) sincerely appreciate the reviewer for providing these comments on this manuscript. Below, we address all of these comments line by line. The reviewer's comments are displayed in bold font, while the author's response to them is displayed in nonbold font.

Reviewer's report:

**1. This is not a physically viable hypothesis.**

We will echo this comment latter.

**2. Because the density of the continents is larger than the density of water, it is the continents that would push the water, not the other way around.**

We thank the reviewer for providing this comment. Nevertheless, we acknowledge that the issue raised in this comment may require further attention.

First and foremost, fluid mechanics dictates that water pressure applied to the wall of a container generates a force that pushes the container's wall. This principle has been established for centuries. For further information on this topic, please refer to the book by Cengel and Cimbala (2014). Similarly, oceanic water pressure against the continent's wall produces a force that pushes the continent's wall. Liquid pressure differs from solid pressure in that the former arises from the weight and movement of liquid molecules, whereas the latter arises solely from the weight of solids. Additionally, the pressure at any given point within a liquid, such as water, is equal in all directions, unlike in a solid where the pressure is not uniform in all directions. Although continents are denser than oceans, their rocky materials are highly viscous and resistive, making it difficult for them to flow easily. Conversely, the low viscosity of ocean water allows water molecules to flow freely. This structural difference between the two materials explains why solid (rock) can maintain its shape, while liquid (water) conforms to the container holding it.

In general, discussing the topic of force involves two objects: the force exerting object and the object receiving the force. For a force to transfer from one object to another, the first object must move and change its position to apply force on the second object. Hence, without movement from continents, there can be no exertion of force on ocean water. Newton's third law states that when an object is pushed, it pushes back with equal force. This implies that as ocean water pushes the continents, the continents push back on the ocean water in response. However, there exists a difference between the two types of force, whereby the former is active while the latter is passive.

Second, we have engaged in private communication with Dr. John M. Cimbala regarding the question of whether ocean water is responsible for pushing the continents. We have received a response from him: "*Think about an empty tea cup sitting in a vacuum chamber with zero pressure. There is certainly internal pressure in the walls of the tea cup. However, those walls do not exert any kind of pressure or force on the surroundings (which is a vacuum). And the cup stays the same shape and holds its shape regardless of its surroundings. Now take the cup out of the vacuum chamber and into the air. Now air exerts a pressure on the cup walls. The cup walls*

*exert and equal and opposite pressure on the air due to Newton's third law. Now fill the cup with water. The pressure inside the cup increases, and the cup expands ever so slightly, but it still maintains its shape. The cup exerts pressure on the water and vice-versa. But it is the water that causes this pressure, not the cup. Water is a liquid and cannot maintain its shape unless it is in some kind of container. That is where the pressure comes from.*"

Third, the conceptual model presented below explains why fluids have the ability to push solids. Initially, two rocks that are in contact with each other on the ground experience no horizontal force between them. However, if one of the rocks were to be melted into magma, based on the principles of fluid mechanics, the magma would exert a horizontal pressure force on the other rock, even though the density of the magma is slightly less than that of the rock. Similarly, if a rock and a piece of ice were to be placed in contact on the ground, there would be no horizontal force between them. However, if the ice were to melt into water, according to the principles of fluid mechanics, the resulting water would exert a horizontal pressure force on the rock, even though its density is less than that of the rock. The reason for these fluids to push denser solids is that the components of these fluids are highly capable of flowing. If there were no obstacle from the rock, these fluids would flow or collapse towards the ground.

[Figure]

Fourth, we have developed a stress model for the rocks in the Earth's crust to demonstrate how ocean water exerts pressure on the continents. As depicted in the top figure below, our model involves a straight, ocean-loaded crust with a length of 7,500 km and a height of 50 km. We note that this model does not strictly adhere to size ratios, and the Earth's curvature has been ignored. The ocean depth ranges from 5.0 km on the left to 4.0 km on the right. The crust is composed of homogeneous and isotropic rocks, and finite element analysis software, such as Abaqus, is utilized to

produce resultant stress. The model's bottom has a remote boundary condition, while no edge boundary conditions exist for the left and right ends. The lithosphere's upper part is represented by a 50 km depth of crust, that is mostly elastic so the ductile nature is neglected. Our inputs include the crust's pressure from its weight and ocean hydrostatic pressure, while our outputs comprise two datasets: one for the stress caused by the crust's pressure alone and the other for the stress caused by the combination of the crust's and ocean's pressures. A two-dimensional frame enables us to obtain horizontal stress (S11) and vertical stress (S22). Our results, depicted in the bottom image, show that ocean water has a significant impact on the crust's stress. The stress caused by ocean water is mainly compressive and penetrates the entire crust's thickness. Notably, we observed variations in stress concentrations in the continent's upper sections where, without water, the horizontal stress in the continent is slightly compressive (weak red). However, when the water is loaded, the horizontal stress in the continent becomes strongly compressive (green).

[Figure]

Fifth, some people believe that any lateral "density difference" would cause denser substances to flow towards lighter ones, supporting the idea that the denser continents would move towards ocean water. However, it is impossible for the positions of continents and oceans to remain motionless when they are put together. This means that either continents move towards ocean water or ocean water moves towards the continents. To further investigate this matter, consider figure (B) below: compared to

figure (A), if continents were to move towards ocean water, the ocean basin would be filled with substances from the continent, causing the sea level to rise and submerge the coast, ultimately resulting in a decrease in the continent's area. However, this is contradictory to the concept of continental accretion, which has been confirmed by the geophysical community for many years. For more information on the continental accretion, please refer to the research by Zhu et al. (2021). Instead, consider figure (C) below: compared to figure (A), if ocean water were to compress the Earth's crust, the elasticity of the crust's rock would cause it to deform in response to the ocean water pressure. Consequently, the ocean basin would expand and the water in the shallow sea would flow towards it, causing parts of the seafloor to be exposed and become landmasses. As a result, the continent's area would increase, which aligns with the concept of continental accretion.

[Figure]

Sixth, the issue raised by the reviewer may be related to the difference between the internal pressure and external force acting on an object. The motion of any object is determined by its external forces, such as the force applied by two people on opposite sides of a rock on the ground. If the combined force is greater than the friction force, the rock moves; if not, the rock remains motionless. However, the object's internal pressure is not relevant to its motion. When there is an external force, there is deformation (stress) as a response. In this study (see Figure 5 of the manuscript), Plate A moves relative to Plate B, Plate C, Ocean, and Asthenosphere, so we only need to consider the forces exerted by these bodies to determine Plate A's motion. The reviewer may be referring to the continent's creep and its density comparison with ocean water, but this study focuses solely on plate motion, assuming the plate is rigid. The creep does not counteract plate motion, just as the deformation caused by the two people pushing the rock is distinct from the rock's motion relative to the ground.

[Figure]

Finally, we can offer a more practical perspective on this issue. When a reservoir is constructed and filled with water, the water pressure forces begins to compress the dam and walls of the reservoir. Initially, the deformation may be negligible, but over time it can accumulate. This is why protective-stability measurements are crucial when building a reservoir. Similarly, when the ocean water pressure force compresses the Earth's crust (including the walls of the continents), the deformation in the crust will become significant after billions of years of accumulation.

**3. Even if the authors do a first order calculation of the ocean-generated force per unit area (F = drho\*g\*z^2, drho = 2800-1000 kg/m^3, g = 9.81 m/s^2 and z = 5000 m), the force is 4e11 N/m (directed from the continent to the water), which is an order of magnitude less than the ridge push force (~2.5e12 N/m) and 2 orders of magnitude less than the slab pull force (~30e12 N/m). So, this is not a first order contribution to the plate force balance.**

Thank the reviewer for these comments. Below we have addressed each of the comments raised. The calculation "**Even if you do a first-order calculation of the force per unit area (F = drho\*g\*z^2, drho = 2800-1000 kg/m^3, g = 9.81 m/s^2, and z = 5000 m)**" may be inaccurate. One cannot use a density difference between solid and fluid to calculate the force between the two. As demonstrated in the manuscript, we agree that the ocean-generated force holds a magnitude of ~4e11N/m. We have addressed the comment "**the ocean-generated force is directed from the continent to the water**" in the author's response above. We also agree with the comment that the ocean-generated force is an order of magnitude less than the ridge push force (**~2.5e12 N/m**) and two orders of magnitude less than the slab pull force (**~30e12 N/m**). However, we argue that the comment "**So, this is not a first-order contribution to the plate force balance**" deserves further discussion.

First, we have demonstrated the issue of the plate force balance in section 4.1 of the manuscript (see lines 688~726).

Second, it is true that slab pull has a magnitude of approximately $10^{13}$ N/m. However, out of the 8 major plates (African, Antarctic, Eurasian, Indo-Australian, North American, Pacific, and South American), only the Pacific plate is attached to the slab. It is important to understand that the motion of a plate is controlled by its own force

balance. Just because the magnitude of slab pull is the largest, it does not necessarily mean that this force has contributed to the motion of the African, Antarctic, Eurasian, Indo-Australian, North American, and South American plates. On the other hand, ridge push has a magnitude of around $10^{12}$ N/m. This force is believed to combine collisional, shearing, and basal friction forces to form force balance, which controls the motion of the continental plate. In contrast, ocean-generated force has a magnitude of around $10^{11}$ N/m. As demonstrated in section 3 of the manuscript, we have arranged this force to combine with ridge push, collisional, shearing, and basal friction forces to form force balance, which controls the motion of both continental and oceanic plates. This indicates that the magnitude of the plate driving force is not the most important factor. Therefore, the force balance generated by driving forces of different magnitudes can explain an identical plate motion. This peculiarity arises from the force balance itself. For any plate, the force balance equation can be expressed as $F_{\text{net-driving}}$-$F_{\text{basal}}$=0, where $F_{\text{basal}} = \mu Au/y$. In this equation, $F_{\text{net-driving}}$ represents the net driving force that includes the plate driving force, collisional force, and shearing forces. Meanwhile, $F_{\text{basal}}$ denotes the basal friction force that the asthenosphere exerts on the plate. The variables μ, A, u, and y stand for the asthenosphere viscosity, plate area, plate speed, and thickness of the asthenosphere, respectively. Thus, u can be calculated as $yF_{\text{net-driving}}/\mu A$. It is essential to note that the force balance equation is used to replicate the observed speed of the plate motion. Although A and y are well established, the viscosity of the asthenosphere (μ) remains uncertain. According to the experiments and theoretical models of various authors presented in lines 447-466 of the original manuscript, μ can span a broad range from $10^{15}$ to $10^{20}$ Pas. In practice, if the plate driving force (e.g., slab pull or ridge push) is significant, a high viscosity value can be chosen to balance the equation. On the other hand, if the plate driving force (e.g., ocean-generated force) is small, a low viscosity value can be applied. Either choice is valid.

Last, the reviewer is arguing that the current plate driving forces (slab pull and ridge push) are still effective. However, the author believes that the reviewer should not disregard the long-standing controversy regarding these two forces, as detailed in section 2 of the manuscript. The weaknesses of these forces are numerous, and the advantage (i.e. their large magnitudes) can be greatly undermined. Furthermore, slabs are deeply buried under trenches and ridges are situated on the ocean floor - the topography, density, temperature, and rheology of these bodies have not been well established. This lack of knowledge means that our understanding of these two forces is still in the theoretical and modeling stages. On the other hand, we have a more substantial understanding of ocean-generated forces. Ocean topography has been well-measured, the density and temperature of ocean water are well-known, fluid mechanics has been well-established, and ocean bottom pressure is widely measured.

Consequently, our response to the reviewer's comments above supports that the hypothesis presented in this study is physically viable.

At the end of this response, we look forward to hearing from you regarding our manuscript. We would be glad to respond to any further questions and comments that

you may have.

---

## Author Comment (AC4)

**Author's response to the Referee 1's comments**

We (the author) gratefully thank the Referee 1 for this review of our manuscript. Below, we have addressed each of these comments in turn, with the Referee 1's comments presented in bold font and our reply in non-bold font.

**Comment 1. This is a very ambitious paper that aims to reconsider and advocate against most of the papers that have discussed the equilibrium of plate tectonics. The goal is also to reconsider the effect of tides on plate motions (and on seismicity) and to propose a scenario for the initiation of plate tectonics.**
Reply: These comments are accurate.

**Comment 2. The author insists that "mantle convection had been given up by most of geophysicists" and "mantle convection cannot be realistic". I am not totally sure what he means by that, probably that convection cannot explain plate tectonic? (although various papers involving Schmalzl, Bercovici, Tackley, Coltice... provided mantle convection models with self generated plates). I hope he does not think that mantle convection does not exist.**
Reply: These comments require the author to exercise caution. Upon reviewing section 4.6 of the manuscript and examining relevant research on mantle convection (e.g. Bercovici et al., 2015, Coltice et al., 2017), we believe that the statements "**mantle convection had been given up by most geophysicists**" and "**mantle convection cannot be realistic**" are problematic. As indicated in lines 59 to 65 of the manuscript, the geophysical community currently acknowledges the large-scale circulation of plate and mantle, while some improved models of mantle convection, as discussed in Coltice et al. (2017), are still being developed. Furthermore, we did not perform a statistical analysis to support the claim that most geophysicists reject mantle convection. Additionally, arguing the shortcomings of mantle convection is not particularly relevant to this work. Given the current state of affairs, we have decided to remove section 4.6 in the revised manuscript.

**Comment 3. The paper is very long and, for me, very difficult to follow. The concepts are often unclear. The very large bibliography is always presented as confirming the author ideas even though I would say that they often oppose his ideas.**
Reply: We apologize that the Referee 1 may have found this paper difficult to understand. We would like to address the feedback provided by Referee 1 regarding the clarity of certain concepts mentioned in the manuscript. If possible, providing more details on the specific lines or sections where the concepts are unclear would be helpful for us to make appropriate revisions.
Additionally, the comment that the bibliography may oppose our ideas is appreciated and we have carefully checked the literature cited in our manuscript, specifically in lines 50~65, lines 119~123, and lines 225~230. Although we may have missed some discrepancies, we have made revisions in the revised version of the manuscript to address these inconsistencies.

**Comment 4. I was really unable to understand exactly the theory itself; the "plates" and the "forces" are not clearly defined. For exemple, the author says that the pressure on a continent, due to the ocean, is larger when the ocean is**

**deeper, and he seems to interpret this observation as "a deep ocean pushes the continent". However, it is obvious than the crust/lithosphere has to be thicker on the side of the shallow ocean and it is rather this side, where the shallow ocean is present, that pushes the continent (i.e; continents tend to extend over the oceans). When the objects on which forces are applied are not properly defined, it is difficult to write correct force balances.**

Reply: These comments raise two key issues: first, the Referee 1 believes that the crust beneath the shallow ocean is providing force to push the continent, rather than the deeper ocean; and second, the Referee 1 suggests that the author's definition of the plates and the forces acting upon them may be inadequate. We will address each concern individually below.

For (A):

The Referee 1's view requires to carefully differentiate the crust's (continent's) deformation from plate motion. To make the issue become clear, we use a simple model below to illustrate the Referee 1's view. The Referee 1 believes that, as the crust beneath the shallow Ocean 1 is thicker than the crust beneath the deeper Ocean 2, and because the crust's density is greater than the density of water, the thicker crust beneath Ocean 1 provides a force ($F_c$, for example) on Continent; and since $F_L+F_c>F_R$, there results in a net force that pushes Continent to move toward Ocean 2. The Referee 1's implication is that the higher and denser Continent would extend towards Ocean 2 and Ocean 1(**i.e; continents tend to extend over the oceans**).

[Figure]

And now we list Figure 5 of the manuscript as below to express our view. We state in the manuscript that, the ocean provides one force ($F_L^{'}$) on the left side of the continent and another force ($F_R^{'}$) on the right side; the continent is fixed on the top of the continental plate (for example, Plate A), this attachment allows the two forces to be transferred to the plate. These two forces combine the ridge push force ($F_{ridge}$), the collisional force ($F_c$), and the basal friction ($f_{base}$) to determine Plate A's motion.

[Figure]

There is indeed a lateral density difference between continent and ocean. However, the idea that this difference makes continent tend to extend over ocean may not be realistic. If continent were forced by the lateral density difference to extend over ocean, it would cause ocean basin to be filled with substances from the continent, leading to a rise in sea level and submerging the coast. Ultimately, this would result in the decrease of the continent's area, which contradicts the concept of continental accretion that has been confirmed by the geophysical community for many years. For more information on continental accretion, please refer to a recent review article by Zhu et al. (2021) (https://doi.org/10.1029/2019RG000689). In addition, the crust's (continent's) deformation does not conflict with plate motion. As seen in the figure above, the external forces ($F_L'$, $F_R'$, $F_{ridge}$, $F_c$, and $f_{base}$) are responsible for controlling the motion of Plate A, but these forces also impact the crust (continent), leading to its deformation. Ultimately, the final deformation of the crust (continent) is determined by several factors, including the external forces, physical properties of the crust's rock, and internal pressure resulting from its weight. To demonstrate, we utilized a model to show that when ocean water is loaded on the crust, the resulting stress permeates the entire crust (continent), causing the horizontal stress to change from tensile (red) to compressive (green). This provides evidence that ocean water is responsible for pushing/compressing the continent, rather than the other way around. The reason for why the denser continent does not extend/flow towards the ocean has been extensively discussed in our response to CC1's comments, as it may relate to differences in pressure properties between rock and water and differences in flexibility between the two.

[Figure]

[Figure]

In fact, the mechanism behind continental accretion (**i.e; continents tend to extend over the oceans**) remains elusive. As demonstrated in the above model, ocean water may play a role in the deformation of the crust/continent. This leads us to propose a possible explanation for continental accretion: ocean water compresses the continent, causing the elastic continent to deform in response to the pressure. As a result, the ocean basin would expand, causing water in the shallow sea to flow towards the ocean basin. This, in turn, would expose parts of the seafloor under the shallow sea and transform them into landmasses. This occurrence would ultimately lead to an increase in the continent's area. In light of this, we have decided to address this issue raised by the Referee 1 and CC1 by adding a section 4.6 to the manuscript (see below).

**"4.6 Does the ocean force relate to the continent's movement?**
Some individuals believe that a lateral density difference results in dense continents extending over oceans. It is crucial to examine this notion, as it sheds light on how continents interact with oceans. If continents were compelled by a lateral density difference to extend over oceans, the ocean basin would be filled with substances from the continent, which would cause the sea level to rise and flood the shore, ultimately resulting in a decrease in the continent's size. This contradicts the concept of continental accretion, which has been accepted by the geophysical community for many years. More information about continental accretion can be found in a recent study by Zhu et al. (2021). Additionally, continents must deform in response to any force, such as the ocean-generated force presented in this work.
A model has been developed to demonstrate the stress generated by ocean-generated forces. The model comprises the Earth's crust, which is carrying the weight of the ocean. The Earth's curvature is neglected, and the crust's length and thickness are 7,500 km and 50 km, respectively. The depth of the ocean ranges from 5.0 km on the

left to 4.0 km on the right. The crust is made up of rocks and is assumed to be homogeneous and isotropic. We have used finite element analysis software, namely Abaqus, to analyze the resultant stress. The bottom of the model is remotely constrained, while there are no edge boundary conditions on the left and right sides of the model. The elastic modulus, Poisson ratio, and rock density of the model are set to 100,000 MPa, 0.3, and 2,690 kg/m$^3$, respectively. The inputs consist of the pressure exerted on the crust by its own weight and the hydrostatic pressure of the ocean. The outputs comprise two data sets: one for stress caused solely by the pressure of the crust, and another for stress resulting from a combination of the crust's pressure and the ocean's pressure. The two-dimensional framework enables us to calculate both horizontal stress (S11) and vertical stress (S22). Figure 19 illustrates the model and the corresponding stress values. Notably, the addition of the ocean's pressure leads to stress that fully penetrates the crust (continent). This causes the previously horizontal tensile (red) stress in the continent to shift to compressive (green).

[Figure]

**Figure 19. Modeling the stress produced by crust and ocean.** Top, geometry of the model; bottom, the stress produced by the loads, where the stress's unit is MPa. The negative symbol "-" denotes compressional.

The modelling above leads us to consider a solution for the occurrence of continental accretion: As demonstrated in the above model, ocean water may play a role in the deformation of the crust/continent. This leads us to propose a possible explanation for continental accretion: ocean water compresses the continent, causing the elastic continent to deform in response to the pressure. As a result, the ocean basin would expand, causing water in the shallow sea to flow towards the ocean basin. This, in turn, would expose parts of the seafloor under the shallow sea and transform them into landmasses. This occurrence would ultimately lead to an increase in the

continent's area. Figure 20 compares two paths of the continent's accretion: from A to B, continent extends towards ocean, then, the water's area increases whereas the continent's areas decreases; from A to C, basin expands towards continent, then, the water's area decreases whereas the continent's area increases.

[Figure]

**Figure 20. Comparing two paths of the continent's accretion.**

For (B):
The Referee 1 has raised concerns about the inadequate definition of plates and forces in the manuscript. However, this point can be tested. In section 3.1 (see lines 219-290), we firstly define the ocean-generated force that acts on a sample continent, and then, the direction of this force and its magnitude are specified for real continents (see Figure 4 and Table 1). In section 3.2 (see lines 303-304), we state "The continents are fixed on the top of the lithosphere, and the lithospheric plates connect to each other, this relationship allows the ocean-generated force to be laterally transferred to the lithospheric plates." So, the ocean-generated force is defined for the lithospheric plate. From lines 306 to lines 317, we list possible forces that act on a continental plate and further discuss the physical nature of these forces. This pattern follows Forsyth and Uyeda (1975). In Figure 5 of the manuscript, we use a model to exhibit plate distribution and the defined forces acting on the plates. For example, Plate A is treated as a continental plate, the forces acting on it include the ocean-generated force $F_L$' and $F_R$', the ridge push force $F_{ridge}$, the collisional force $F_c$, and the basal friction $f_{base}$. From line 326 to line 349, we discuss the torques resulting from the forces we have defined. Moving onto lines 351 to 485, we use Figure 6 of the manuscript to plot these forces onto a spherical frame, examine the torque balances that these forces create, and then use these balances to resolve the movements of the six plates. However, please note that in both lines 360-361 and lines 409-417, additional forces (such as collisional and shearing forces) were added to the defined forces. Unfortunately, we could not exhibit these forces in Figure 5, as the frame is only two-dimensional. We have also defined a few collisional forces for the Pacific Plate, while neglecting slab pull. We have provided reasons for this on lines 414-416.
Even so, we're still afraid that our definition of the ocean-generated force isn't too clear. To improve this, we add a figure (as below) in the revised manuscript. Of

course, if the definition of the plates and forces we have used is not acceptable to the Referee 1, we would appreciate some feedback, including references or examples so that we can make proper adjustments.

[Figure]

**Figure 4. Modeling the ocean-generated forces acting on the continent.** $F_L(F_R)$ represents the ocean-generated force on the left (right) side of the continent, while $F_L'(F_R')$ and $F_L''(F_R'')$ denote the horizontal and vertical forces decomposed from the ocean-generated force, respectively. $L$ denotes the width of the continent's side; $h_L$ and $h_R$ are the ocean's depth on the left and right, respectively. Note that the Earth's curvature is neglected.

**Comment 5. I had also difficulties with the numerical applications. To take an exemple, around lines 700. Yongfeng Yang computes an "ocean force" F_AR= 0.245e12 N/m for a d=5 km ocean. I would compute this force as 1/2 rho g d^2=0.1225e12 N/m. I may be wrong but it seems that a factor 1/2 is missing. The same factor seems to be also missing in F_AL (the opposite force of a shallow ocean of 3km), and of course on the resulting force F_AR-F_AL (Yongfeng Yang uses 0.1568e12 N/m when it should be 0.0784e12 N/m).**
Reply: These comments are correct. We did, in fact, overlook the factor of 1/2. As a result, we have made improvements to the numerical analysis in the revised version of the manuscript.

**Comment 6. But already 5 km is an unrealistically large depth: the average depth of oceans is only 3.7 km and people looking for a potential 'dynamic topography, do not seem to see any difference in ocean bathymetries larger than say 1 km (and this is already a very generous value, by isostasy a h=2 km difference of bathymetry implies under the shallow ocean a crustal root of r=h (rho_crust-rho_water)/(rho_mantle-rho_crust)=9 km, so a crust thicker by 9+2=11 km under the shallow ocean). Therefore the ocean force between an ocean of depth d1=3200 m and an ocean of depth d2=4200 m, is only 0.036e12 N/m, 4-5 times smaller than the value chosen by the author.**
Reply: These comments suggest that we may have overestimated the strength of ocean forces by focusing solely on deeper ocean sections. Connecting to the previous comments (**To take an exemple, around lines 700. Yongfeng Yang computes an**

**"ocean force" F_AR= 0.245e12 N/m for a d=5 km ocean.**), we will argue that the magnitude of this force is not crucial for section 4.1. As illustrated by Figure 12 in the manuscript (see the figure below), our model estimates ocean depths of 5 km and 3 km to the right and left of the continent respectively, but readers may consider depths of 4 km and 3 km on the right and left instead. This variation in estimation serves to demonstrate that force balance can be achieved with related forces. Further details can be found in our response below.

[Figure]

**Comment 7. It is already difficult for me to understand how a force of 0.1568e12 N/m could play a significant role against a ridge push of 4e12 N/m (using the author numbers, i.e. against a force 27 times larger), but it seems that the ratio is in fact larger than a factor 100.**

Reply: Based on our previous response, it appears that the Referee 1 is arguing that an ocean force of 0.1568e12 N/m is insufficient to counterbalance a ridge push force of 4e12 N/m. However, we do not use the ocean force to counteract the ridge push force. This can be seen in several parts of the manuscript such as lines 13~14, lines 314~317, lines 326~332, lines 356~358, lines 351~385, line 497~524, and lines 694~721. In our approach, the ridge push force is always treated as a plate driving force while the ocean force and ridge push force are combined to balance the collisional, shearing, and basal friction forces. This balancing act results in force balance and plate movement resolution. Let's go back to the manuscript to see the thinking line of the paper. In Section 2.2, we model the existing forces (ridge push,

collisional, and basal friction) and find that they are not sufficient to account for observed stress, implying the need for additional force. In Sections 3.1 and 3.2, we introduce the ocean-generated force and use this force in tandem with the ridge push force to balance the collisional, shearing, and basal friction forces. In Section 3.3, we use modeling to demonstrate that even with a ridge push force of 4e12 N/m, the combination of the ocean-generated force, ridge push force, collisional force, and shearing force is still insufficient to account for observed stress. However, reducing the ridge push force to below the ocean-generated force can reach the observed stress (refer to lines 618-631). In Section 4.1 (refer to lines 694-721), we show that force balances can be achieved with these combined forces. It is important for the Referee 1 to note that the ridge push force is approximately symmetric around the ridge crest, as shown in the figure above. The ridge push force $F_{RL}$ pushes Plate A at its right side, at the same time, $F_{RR}$ pushes Plate C, Plate C pushes Plate B, and Plate B also pushes Plate A at its left side, consequently, $F_{RL}$ is properly balanced out by $F_{RR}$. Therefore, regardless of whether the ridge push force is valued at 4e12 N/m or below, the force balance can still be achieved. Moreover, we have mentioned readers in the manuscript (see lines 717~721), " …, even if the ridge push force $F_{RL}$ ($F_{RR}$) is given a smaller amplitude (~ $10^{10}$ N m$^{-1}$, for example), so long as the collisional force $F_{BA}$ ($F_{AB}$, $F_{CB}$, and $F_{BC}$) is properly valued, these force balances can always be created. Nevertheless, as demonstrated in section 3.3, a ridge push force of $4.0 \times 10^{12}$ N m$^{-1}$ would result in a horizontal stress that is mostly concentrated on the lower part of the lithosphere, which is not in accordance with observation. Hence, we prefer to accept the ridge push force to be smaller than ocean-generated force."

This comment, "**the ratio is actually greater than 100 times,**" is accurate but lacks significant value. The Referee 1's comments are related to some of CC1's comments which have already been addressed by the author. Therefore, it is unnecessary to further emphasize this matter.

**Comment 8. I note that the slab traction is generally estimated about 10 times larger than the ridge push (the author mentions a slab traction of about 3.3e13 N/m which is the right amount). Quoting Bercovici et al, (AGU monograph, 2000) "As demonstrated by Forsyth and Uyeda [1975], the correlation between the connectivity of a plate to a slab (i.e., the percent of its perimeter taken by subduction zones) and the plate's velocity argues rather conclusively for the dominance of slab pull as a plate driving force".**

Reply: These comments merit further discussion.

On the other hand, slab pull (traction) is widely accepted as the greatest. However, out of the 8 major plates (African, Antarctic, Eurasian, Indo-Australian, North American, Pacific, and South American), only the Pacific Plate is attached to slab that produces the pull force. Please be aware, these 7 plates (African, Antarctic, Eurasian, Indo-Australian, North American, and South American) don't attach to any slab. So, slab pull is only suitable for the oceanic plate like the Pacific Plate. Just because "**the correlation between the connectivity of a plate to a slab (i.e., the percent of its perimeter taken by subduction zones) and the plate's velocity**" is strong, it does not necessarily mean that the slab has contributed to plate motion. For example, we will argue that, as readers have seen in the manuscript (see lines 41~415, Table 2(D), Figure 7(f), lines 548~551, Table 4(B), Figure 9(c)), we have successfully produced a movement for the Pacific Plate by means of a combination of three collisional forces. In short, without the contribution of slab pull, the Pacific Plate's movement can still be reached. Perhaps, the strong correlation between the **connectivity and the plate's**

**velocity** is just a coincidence. See the figure below (Figure 13(B) of the manuscript is a copy of this feature), the Eurasian Plate rotates clockwise, the North American Plate rotates counterclockwise, the Indian-Australian Plates move northeast, and the Pacific Plate moves northwest, this situation allows the former three plates to circle the Pacific Plate. As a result, the Pacific Plate's margin is forced to subduct, forming very long **subduction zones** in the boundaries between the Pacific Plate and these three plates. From this point, the slab itself may be a consequence of plate motion. These comments require further discussion. While slab pull (traction) is widely accepted as the greatest force driving plate motion, it only applies to the Pacific Plate. Out of the 8 major plates (African, Antarctic, Eurasian, Indo-Australian, North American, Pacific, and South American), the Pacific Plate is the only one attached to a slab that produces the pull force. It is important to note that a strong correlation between the connectivity of a plate to a slab (i.e., the percent of its perimeter taken by subduction zones) and the plate's velocity does not necessarily indicate that the slab has contributed to plate motion. As shown in the manuscript (see lines 41~415, Table 2(D), Figure 7(f), lines 548~551, Table 4(B), Figure 9(c)), a combination of three collisional forces has successfully produced movement for the Pacific Plate without the contribution of slab pull. In fact, perhaps the strong correlation between connectivity and plate velocity is merely a coincidence. The figure below (Figure 13(B) of the manuscript is a copy of this feature) shows that the Eurasian Plate rotates clockwise, the North American Plate rotates counterclockwise, the Indian-Australian Plates move northeast, and the Pacific Plate moves northwest, resulting in the former three plates circling the Pacific Plate. As a result, the margin of the Pacific Plate is forced to subduct, forming very long subduction zones at the boundaries between the Pacific Plate and these three plates. This suggests that the slab itself may be a consequence of plate motion.

[Figure]

On the flip side, the Referee 1 has utilized the work of Bercovici et al. (AGU monograph, 2000) to insinuate that slab pull still holds significance. However, this is not entirely accurate. The Referee 1 ought to recognize the longstanding controversy surrounding this force that's delineated in section 2.1 of the manuscript. There are numerous loopholes in this force, which can significantly undermine its advantages (e.g., magnitude of 3.3e13 N/m). That being said, our model does leave room for slab pull, as evidenced by lines 414-416 in the manuscript "Taking into consideration the

long argument of slab pull that is listed in section 2.1, we presently neglect slab pull. And if this force can be confirmed in the future, it can be added into this model."

**Comment 9. Yongfeng Yang does not believe in ridge push although, the same halfspace cooling model is used to estimate ridge push and slab pull.**
Reply: As the Referee 1 has read our response above, this is not true that **Yongfeng Yang does not believe in ridge push although**. This comment "**the same halfspace cooling model is used to estimate ridge push and slab pull**" is right but not relevant to this research. If the Referee 1 has any specific concerns about this issue, please elaborate on them so that we can incorporate your feedback in the revised version.

**Comment 10. I do not think the paper is clear, convincing and rigorous enough to be accepted for publication.**
Reply: Along with this response, we have made revisions to the manuscript. Other modifications include changes to the literature, figure sequence, and references.
We look forward to hearing from you regarding our manuscript. We would be glad to respond to any further questions and comments that you may have.

---

## Author Comment (AC6)

Response to reviewers

We (as author) are very grateful that the reviewers (i.e., Chuanliang Li, Referee 1, and Peter Malin) have paid high attention to this manuscript. The comments from these reviewers are valuable and helpful for improving the quality of this manuscript. In the open discussion phase, we have made response to the comments. In this final response phase, we improve the response by addressing all of the comments line by line. The comments are displayed in bold font, while the author's responses to them is displayed in nonbold font.

Response to Chuanliang Li

**1. This is not a physically viable hypothesis.**

Reply: We will echo this comment latter.

**2. Because the density of the continents is larger than the density of water, it is the continents that would push the water, not the other way around.**

Reply: Thank the reviewer for providing this comment. Nevertheless, we acknowledge that the issue raised in this comment may require further attention.

First and foremost, fluid mechanics dictates that water pressure applied to the wall of a container generates a force that pushes the container's wall. This knowledge has been established for centuries. For further information on this topic, please refer to a book by Cengel and Cimbala (2014). Similarly, oceanic water pressure against the continent's wall generates a force that pushes the continent's wall. Liquid pressure differs from solid pressure in that the former arises from the weight and movement of liquid molecules, whereas the latter arises solely from the weight of solids. Additionally, the pressure at any given point within a liquid, such as water, is equal in all directions, unlike in a solid where the pressure is not uniform in all directions. Although continents are denser than oceans, the rocky materials within them are highly viscous and resistive, making it difficult for them to flow easily. Conversely, the low viscosity of ocean water allows water particles to flow freely. This difference in physics between the two materials explains why solid (rock) can maintain its shape, while liquid (water) conforms to the container holding it. In general, discussing the topic of force involves two objects: the force exerting object and the object receiving the force. For one object using force to push another object, the first object must positively move and change its position to apply force on the second object. Hence, without a positive movement of the continents relative to ocean water, there can be no exertion of force on ocean water. Newton's third law states that when an object is pushed, it pushes back with equal force. This implies that, when ocean water pushes the continents, the continents push back on the ocean water in response. However, there exists a difference between the two types of push, whereby the former is active while the latter is passive.

Second, we have discussed with Dr. John M. Cimbala regarding the question of whether ocean water is responsible for pushing the continents. There is a message from him: "*Think about an empty tea cup sitting in a vacuum chamber with zero pressure. There is certainly internal pressure in the walls of the tea cup. However, those walls do not exert any kind of pressure or force on the surroundings (which is a vacuum). And the cup stays the same shape and holds its shape regardless of its surroundings. Now take the cup out of the vacuum chamber and into the air. Now air exerts a pressure on the cup walls. The cup walls exert and equal and opposite pressure on the*

*air due to Newton's third law. Now fill the cup with water. The pressure inside the cup increases, and the cup expands ever so slightly, but it still maintains its shape. The cup exerts pressure on the water and vice-versa. But it is the water that causes this pressure, not the cup. Water is a liquid and cannot maintain its shape unless it is in some kind of container. That is where the pressure comes from."*

According to a report from National Geographic Society (https://education.nationalgeographic.org/resource/formation-earth/), the Earth's infancy is extremely hot, to the point that the planet likely consisted almost entirely of molten magma. Over the course of a few hundred million years, the planet began to cool and oceans of liquid water formed. So, the Earth's crust formed earlier than ocean water did, the loading of ocean water onto the Earth's crust is like that the water is filled into the cup. So, it is the ocean water that causes pressure to compress/push the crust (continents), not the crust (continents).

Third, the conceptual model presented below explains why fluids have ability to push solids. Initially, two rocks that are in contact with each other on the ground experience no horizontal force between them. However, if one of the rocks were melted into magma, based on the principles of fluid mechanics, the magma would exert a horizontal pressure force on the other rock, even though the density of the magma is slightly less than that of the rock. Similarly, if a rock and a piece of ice are placed in contact on the ground, there would be no horizontal force between them. However, if the ice is melt into water, according to the principles of fluid mechanics, the resulting water would exert a horizontal pressure force on the rock, even though its density is less than that of the rock. The reason for these fluids to push denser solids is that the components of these fluids are highly capable of flowing. If there were no obstacle from the rock, these fluids would flow or collapse towards the ground.

[Figure]

Fourth, we have developed a model below to demonstrate how ocean water exerts pressure on the continents. As depicted in the top figure below, the model involves a straight, ocean-loaded crust with a length of 7,500 km and a height of 50 km. Please note that this model does not strictly adhere to size ratios, and the Earth's curvature has been ignored. The ocean depth ranges from 5.0

km on the left to 4.0 km on the right. The crust is composed of homogeneous and isotropic rocks, and finite element analysis software, such as Abaqus, is utilized to produce resultant stress. The model's bottom has a remote boundary condition, while no edge boundary conditions exist for the left and right ends. The lithosphere's upper part is represented by a 50 km depth of crust, which is mostly elastic so the ductile nature is neglected. The inputs include the crust's pressure from its weight and ocean hydrostatic pressure, while the outputs comprise two sets of data: one is the stress caused by the crust's pressure alone and the other is the stress caused by a combination of the crust's and ocean's pressures. A two-dimensional frame enables us to obtain horizontal stress (S11) and vertical stress (S22). Our results, depicted in the bottom image, show that ocean water has a significant impact on the crust's stress. The stress caused by ocean water is mainly compressive and penetrates the entire crust's thickness. Notably, we found variations in stress concentrations in the continent's upper sections where, without water, the horizontal stress in the continent is slightly tensile (weak red). However, when the water is loaded, the horizontal stress in the continent becomes strongly compressive (green).

[Figure]

Fifth, some people believe that any lateral "density difference" would cause denser substances to move/flow towards lighter ones, supporting the idea that the denser continents would move/extend towards ocean water. Of course, it is impossible for the positions of continents and oceans to remain motionless when they are put together. This means that either continents move/extend towards ocean water or ocean water moves towards the continents. To investigate this issue, consider figure (B) below: compared with figure (A), if continents were to move/extend towards ocean water, the ocean basin would be filled with substances from the continent, causing the sea

level to rise and submerge the coast, ultimately resulting in a decrease in the continent's area. Imagine it, putting a stone into a cup of water, the water level rises accordingly. So, putting the continent's rocks into ocean basin, the water level also rises accordingly. However, this situation is entirely contradictory to the concept of continental accretion (i.e., the area of the continent increases with time), which has been confirmed by geophysical community for many years. For more information about the continental accretion, please refer to a recent review research by Zhu et al. (2021) (https://doi.org/10.1029/2019RG000689). Instead, consider figure (C) below: compared with figure (A), if ocean water compresses (pushes) the Earth's crust, the elastic crust would deform in response to the ocean water pressure. Consequently, the ocean basin expands and the water in the shallow sea flows towards it, causing parts of the seafloor to be exposed and become landmasses. As a result, the continent's area increases, which aligns with the concept of the continental accretion.

[Figure]

Sixth, the issue raised by the reviewer may be related to a relationship between internal force and external force. The motion of any object is determined by the external forces that it undergoes, such as the forces applied by two people on opposite sides of a rock on the ground. If the combined force is greater than the basal friction force, the rock moves. However, the object's internal force is not relevant to this resulting motion. If there is an external force, there must be deformation (stress) as a response, and the magnitude of the deformation is determined by the rock's rigidity. In this study (see figure below, which is a copy of Figure 5 in the manuscript), Plate A connects to Plate B, Plate C, Ocean, and Asthenosphere, so we need to consider the forces exerted by these bodies to determine Plate A's motion. The reviewer's comment may be referring to the continent's creep, which mainly relates to the internal force. However, this study focuses on plate motion, assuming the plate is rigid. In fact, the continent's creep does not counteract plate motion, just as the deformation caused by the two people pushing the rock is distinct from the rock's motion relative to the ground.

[Figure]

Finally, we may offer a more practical perspective on this issue. When a reservoir is constructed and filled with water, the water pressure forces begins to compress the dam and walls of the reservoir. Initially, the deformation is negligible, but over time it can accumulate. This is why protective-stability facilities are considered when building a reservoir. Similarly, when ocean water was loaded on the Earth's crust, the deformation in the crust will become significant after billions of years of accumulation.

**3. Even if the authors do a first order calculation of the ocean-generated force per unit area (F = drho\*g\*z^2, drho = 2800-1000 kg/m^3, g = 9.81 m/s^2 and z = 5000 m), the force is 4e11 N/m (directed from the continent to the water), which is an order of magnitude less than the ridge push force (~2.5e12 N/m) and 2 orders of magnitude less than the slab pull force (~30e12 N/m). So, this is not a first order contribution to the plate force balance.**

Reply: Thank the reviewer for these comments. The calculation presented by the reviewer doesn't follow the principle of fluid mechanics. The comment "**the ocean-generated force is directed from the continent to the water**" has been addressed earlier. We agree that the ocean-generated force is an order of magnitude less than the ridge push force and two orders of magnitude less than the slab pull force. However, the comment "**So, this is not a first-order contribution to the plate force balance**" deserves further discussion.

First, we have demonstrated in the manuscript (see lines 688~726) how the plate force balance can be created by a combination of the ocean-generated force, the ridge push force, the collisional force, and the basal friction force.

Second, among of 7 major plates (African, Antarctic, Eurasian, Indo-Australian, North American, Pacific, and South American), only the Pacific plate, which is oceanic, is attached to slab that produces slab pull force. So, just because slab pull is the largest, it does not necessarily mean that this force contributes to the movements of other plates (African, Antarctic, Eurasian, Indo-Australian, North American, and South American). Moreover, ridge push may combine other forces (i.e., collisional, shearing, and basal friction) to determine the movements of the continental plates. In contrast, as demonstrated in section 3 of the manuscript, we have combined the ocean-generated force with the ridge push force to balance the collisional, shearing, and basal friction forces, by which the movements of both continental and oceanic plates are realized. This result indicates that the force balance leaded by the driving forces of different magnitudes may explain

identical plate motion. This peculiarity arises from the force balance itself. According to the principle of fluid mechanics (Cengel and Climbala, 2014), the force balance for a plate can be expressed as $F_{\text{net-driving}}-F_{\text{basal}}=0$, where $F_{\text{basal}} = \mu Au/y$. In this equation, $F_{\text{net-driving}}$ represents the net driving force, which may include the plate driving force, collisional force, and shearing force. Meanwhile, $F_{\text{basal}}$ denotes the basal friction force exerted by the asthenosphere on the plate. The variables $\mu$, A, u, and y stand for the asthenosphere viscosity, plate area, plate speed, and thickness of the asthenosphere, respectively. Thus, u can be expressed as $u=yF_{\text{net-driving}}/\mu A$. It is important to note that the force balance equation is used to replicate the observed motion. Although plate area and thickness of the asthenosphere are well established, the viscosity of the asthenosphere remains uncertain. According to the experiments and theoretical models of various authors (see lines 447-466 of the original manuscript), the viscosity can span a broad range from $10^{15}$ to $10^{20}$ Pas. In practice, if a plate driving force (e.g., slab pull or ridge push) is large, one may use a high viscosity value to match the equation. And if a plate driving force (e.g., ocean-generated force) is small, one may use a low viscosity value to match the equation. Either choice is valid.

Last, the reviewer may argue that slab pull and ridge push are effective. However, we remind that the long controversy regarding the kinematics and geometry of these two forces, which is well documented in Section 2 of the manuscript, shouldn't be disregarded. In addition to this, slabs are deeply buried under trenches, ridges are situated on the ocean floor - the topography, density, temperature, and rheology of these bodies have not yet been well established. This lack of knowledge means that the current understanding of these two forces is still in the theoretical and modeling stages. Contrary to this, we have obtained a more substantial knowledge of ocean. Ocean topography has been well-measured, the density and temperature of ocean water are well-known, fluid mechanics has been well-established, and ocean bottom pressure is widely measured. Such a comparison allows readers to easily determine which force is reliable.

Consequently, our responses to the reviewer's comments above support that the hypothesis presented in this study is physically viable.

Response to Referee 1

**Comment 1. This is a very ambitious paper that aims to reconsider and advocate against most of the papers that have discussed the equilibrium of plate tectonics. The goal is also to reconsider the effect of tides on plate motions (and on seismicity) and to propose a scenario for the initiation of plate tectonics.**

Reply: Thanks a lot. These evaluations are correct.

**Comment 2. The author insists that "mantle convection had been given up by most of geophysicists" and "mantle convection cannot be realistic". I am not totally sure what he means by that, probably that convection cannot explain plate tectonic? (although various papers involving Schmalzl, Bercovici, Tackley, Coltice... provided mantle convection models with self generated plates). I hope he does not think that mantle convection does not exist.**

Reply: These comments require the author to exercise caution. Upon reviewing Section 4.6 of the manuscript and examining relevant research on mantle convection (e.g. Bercovici et al., 2015, Coltice et al., 2017), we believe that these statements "**mantle convection had been given up by**

**most geophysicists**" and "**mantle convection cannot be realistic**" are flawed. As mentioned in lines 59 to 65 of the manuscript, the geophysical community currently acknowledges the large-scale circulation of plate and mantle, while some improved models of mantle convection are still being developed (Coltice et al., 2017). Furthermore, we never perform a statistical analysis to conclude that most geophysicists reject mantle convection. Additionally, arguing the weakness of mantle convection is not particularly relevant to this present work. Given the current state of affairs, we have removed Section 4.6 in the revised manuscript.

**Comment 3. The paper is very long and, for me, very difficult to follow. The concepts are often unclear. The very large bibliography is always presented as confirming the author ideas even though I would say that they often oppose his ideas.**

Reply: We apologize that Referee 1 may have found this paper difficult to understand. Providing more details on the specific lines or sections where the concepts are unclear would be helpful for us to make appropriate revisions. Additionally, the comment that the bibliography may oppose our ideas is appreciated. We have checked manuscript to find that such inconsistencies occur mostly in lines 50~65, lines 119~123, and lines 225~230. It is possible that we have missed other discrepancies that have been noted by Referee 1. Accordingly, we have made significant improvement on the structure and writing style of the manuscript.

**Comment 4. I was really unable to understand exactly the theory itself; the "plates" and the "forces" are not clearly defined. For exemple, the author says that the pressure on a continent, due to the ocean, is larger when the ocean is deeper, and he seems to interpret this observation as "a deep ocean pushes the continent". However, it is obvious than the crust/lithosphere has to be thicker on the side of the shallow ocean and it is rather this side, where the shallow ocean is present, that pushes the continent (i.e; continents tend to extend over the oceans). When the objects on which forces are applied are not properly defined, it is difficult to write correct force balances.**

Reply: These comments raise two key issues: (A), Referee 1 believes that the crust beneath the shallow ocean provides force to push the continent, rather than the deeper ocean; and (B), Referee 1 found that the author's definition of the plates and the forces acting upon them is inadequate. We address each concern individually below.

For (A):

Referee 1's comment requires to carefully differentiate the crust's (continent's) deformation from plate motion. To make the issue become clear, we use a simple model below to illustrate Referee 1's view. Referee 1 argues that, as the crust beneath the shallow Ocean 1 is thicker than the crust beneath the deeper Ocean 2, and as the crust's density is greater than the density of water, the thicker crust beneath Ocean 1 provides a force ($F_c$, for example) on Continent; and since $F_L+F_c>F_R$, there results in a net force that pushes Continent to move toward Ocean 2. Referee 1's implication is that the higher and denser Continent extends towards Ocean 2 and Ocean 1(**i.e; continents tend to extend over the oceans**).

[Figure]

Unfortunately, what Referee 1 addresses is different from what we address in this study. As exhibited in Figure 5 of the manuscript, the ocean provides one force ($F_L^{'}$) on the left side of the continent and another force ($F_R^{'}$) on the right side. Since the continent is fixed on the top of the continental plate (for example, Plate A), this attachment allows the two forces to be transferred to the plate. These two forces combine the ridge push force ($F_{ridge}$) to balance the collisional force ($F_c$) and the basal friction ($f_{base}$), which determines Plate A's motion.

It is true that there is a lateral density difference between continent and ocean. However, the proposition that the denser continent tend to extend over ocean may not be realistic. Imagine it, if continent extends over ocean, this would cause ocean basin to be filled with substances from the continent, leading to a rise in sea level and submerging the coast. Ultimately, this would result in a decrease of the continent's area. This result entirely contradicts the concept of continental accretion that has been confirmed by geophysical community for many years. For more information about continental accretion, please refer to a recent review article by Zhu et al. (2021) (https://doi.org/10.1029/2019RG000689). In addition, as stated in the author's response to Chuanliang Li, the crust's (continent's) deformation does not conflict with plate motion. See Figure 5 of the manuscript, the external forces ($F_L^{'}$, $F_R^{'}$, $F_{ridge}$, $F_c$, and $f_{base}$) are responsible for determining Plate A's motion, but at the same time these forces compress the crust (continent), leading the crust (continent) to deform. Essentially speaking, the deformation of the crust (continent) is determined by several factors, including the external forces, physical properties of the crust's rock, and internal pressure resulted from its weight. See the author's response to the comments made by Chuanliang Li earlier, we have used a model to show that it is the ocean water that compresses/pushes the continent, rather than the other way around.

In fact, the mechanism behind continental accretion remains poorly understood. As we demonstrated above, ocean water may have played a role in the deformation of the crust/continent. This leads us to propose a solution for the occurrence of continental accretion: ocean water compresses the Earth's crust, causing the crust to deform in response to the pressure. As a result, the ocean basin expands, causing water in the shallow sea to flow towards the ocean basin. This, in turn, exposes parts of the seafloor and transform them into landmasses. Ultimately, the continent's area is increased. In light of this, we have added a Section 4.6 (see below) in the revised manuscript to address the continental accretion.

**"4.6 Does the ocean relate to the continental accretion?**

Some peoples believe that, since the continents are higher and denser than the oceans, they extend over the oceans. It is crucial to examine this notion, because it may shed light on how the continents interact with the oceans. On the one hand, if the continents extend over the oceans, the ocean basin would be filled with substances from the continent, causing the sea level to rise and flood the shore, ultimately, the continent's size decreases. This contradicts the concept of continental accretion (i.e., the continent's area increases over time). More information about continental accretion can refer to a recent study by Zhu et al. (2021). On the other hand, the continents must deform in response to any external force, such as the ocean-generated force presented in this study. To explore the effect, we develop a model to demonstrate the stresses generated by the ocean-generated forces. The model comprises the Earth's crust, which is carrying the weight of the ocean. The Earth's curvature is neglected, and the crust's length and thickness are 7,500 km and 50 km, respectively. The depth of the ocean ranges from 5.0 km on the left to 4.0 km on the right. The crust is made up of rocks and is assumed to be homogeneous and isotropic. We have used finite element analysis software, namely Abaqus, to analyze the resultant stress. The bottom of the model is remotely constrained, while there are no edge boundary conditions on the left and right sides of the model. The elastic modulus, Poisson ratio, and rock density of the model are set to 100,000 MPa, 0.3, and 2,690 kg/m$^3$, respectively. The inputs consist of the pressure exerted on the crust by its own weight and the hydrostatic pressure of the ocean. The outputs comprise two data sets: one for stress caused solely by the pressure of the crust, and another for stress resulting from a combination of the crust's pressure and the ocean's pressure. The two-dimensional framework enables us to obtain both horizontal stress (S11) and vertical stress (S22). Figure 19 illustrates the model and the resultant stress distribution. Notably, the ocean's pressure leads to stress that fully penetrates the crust (continent). This causes the previously horizontal tensile (red) stress in the continent to shift to compressive (green).

[Figure]

**Figure 19. Modeling the stress produced by crust and ocean.** Top, geometry of the model; bottom, the stress produced by the loads. The stress's unit is MPa, and the negative symbol "-" denotes compressional.

The modelling above leads us to consider a solution for the occurrence of continental accretion: ocean water compresses the Earth's crust, the crust deforms in response to the ocean water pressure. And then, the ocean basin expands, causing water in the shallow sea to flow towards the ocean basin. This, in turn, exposes parts of the seafloor under the shallow sea and transform them into landmasses. This ultimately leads to an increase in the continent's area. Figure 20 compares two paths of the continent's accretion: from A to B, continent extends towards ocean, then, the water's area increases whereas the continent's areas decreases; from A to C, basin expands towards continent, then, the water's area decreases whereas the continent's area increases."

[Figure]

**Figure 20. Comparing two paths of the continent's accretion.**"

For (B):

Referee 1 has raised concerns about the inadequate definition of plates and forces in the manuscript. However, this point can be tested. In Section 3.1 (see lines 219-290 of the manuscript), we define the ocean-generated force that acts on a sample continent, and then, the direction of this force and its magnitude are specified for real continents (see Figure 4 and Table 1). In Section 3.2 (see lines 303-304), we state "The continents are fixed on the top of the lithosphere, and the lithospheric plates connect to each other, this relationship allows the ocean-generated force to be laterally transferred to the lithospheric plates." So, the ocean-generated force is defined also for the lithospheric plate. From lines 306 to lines 317, we list possible forces that act on a continental plate and further discuss the physical nature of these forces. This method follows Forsyth and Uyeda (1975). As exhibited in Figure 5, we use a model to show the plate distribution and the forces acting on the plates. For example, Plate A is treated as a continental plate, the forces acting on it include the ocean-generated force $F_L$' and $F_R$', the ridge push force $F_{ridge}$, the collisional force $F_c$, and the basal friction $f_{base}$. From line 326 to line 349, we discuss the torques resulting from the forces we have defined. Moving onto lines 351 to 485, we use Figure 6 to plot these forces onto a spherical frame, examine the torque balances created by these forces, and then use these balances to reproduce the movements of six plates. Please note that in both lines 360-361 and lines 409-417, additional forces (such as collisional and shearing forces) were added to these defined forces. We have also defined a few collisional forces for the Pacific Plate, while slab pull is not considered. We have provided reason for this treatment on lines 414-416. Even so, we really found that the computation of the ocean-generated force isn't too clear. To improve this, we add a figure (as below) and some literature (as below) in the revised manuscript.

[Figure]

**Figure 4. Modeling the ocean-generated forces acting on the continent.** $F_L(F_R)$ represents the ocean-generated force on the left (right) side of the continent, while $F_L'(F_R')$ and $F_L''(F_R'')$ denote the horizontal and vertical forces decomposed from the ocean-generated force, respectively. $L$ denotes the width of the continent's side; $h_L$ and $h_R$ are the ocean's depth on the left and right, respectively. Note that the Earth's curvature is neglected.

"Referring to Figure 5, initially, we design numerous geographical sites located along the edges of the continents. The latitudes and longitudes of these sites were extracted from the ETOPO1 Global Relief Model. The distance between adjacent sites is calculated using their respective latitudes and longitudes and is treated as the width of a smaller rectangular side. Using the NOAA bathymetric data viewer, we obtain the corresponding ocean depth of each rectangular side. Subsequently, we employ the aforementioned equation to compute the horizontal forces present around the continents."

**Comment 5. I had also difficulties with the numerical applications. To take an exemple, around lines 700. Yongfeng Yang computes an "ocean force" F_AR= 0.245e12 N/m for a d=5 km ocean. I would compute this force as 1/2 rho g d^2=0.1225e12 N/m. I may be wrong but it seems that a factor 1/2 is missing. The same factor seems to be also missing in F_AL (the opposite force of a shallow ocean of 3km), and of course on the resulting force F_AR-F_AL (Yongfeng Yang uses 0.1568e12 N/m when it should be 0.0784e12 N/m).**
Reply: These comments are correct. We did, in fact, overlook the factor of 1/2. As a result, we have made improvements on the numerical analysis in the revised manuscript.

**Comment 6. But already 5 km is an unrealistically large depth: the average depth of oceans is only 3.7 km and people looking for a potential 'dynamic topography, do not seem to see any difference in ocean bathymetries larger than say 1 km (and this is already a very generous value, by isostasy a h=2 km difference of bathymetry implies under the shallow ocean a crustal root of   r=h (rho_crust-rho_water)/(rho_mantle-rho_crust)=9 km, so a crust thicker by 9+2=11 km under the shallow ocean). Therefore the ocean force between an ocean**

**of depth d1=3200 m and an ocean of depth d2=4200 m, is only 0.036e12 N/m, 4-5 times smaller than the value chosen by the author.**

Reply: Thank the reviewer for these comments. The method used by the reviewer is different from the method we use. See lines 231-248 of the manuscript, we have made use of the principle of fluid mechanics to constrain the ocean-generated force: applying ocean water pressure to the wall of the continent yields the ocean-generated force. This force can be decomposed into a horizontal force and a vertical force. We express the horizontal force as $F= 0.5\rho g L h^2$, where $\rho$, g, $L$, and $h$ are the density of water, gravitational acceleration, ocean width that fits the continent's width, and ocean depth, respectively. In practice, we at first approximate the continent as a polygonal column that stands in the ocean, and dissect the whole side of this column into a series of smaller rectangular sides connecting one to another, and finally use the formula above to compute the horizontal force for each side of the column. In fact, the ocean force computed through the reviewer's method is the same as that computed through our method. Refer to lines 703-704 of the manuscript, an ocean depth of 5 km yields an ocean force of 0.245e12 N/m (as the reviewer mentioned earlier, we have missed a factor 0.5, so, the correct value is 0.1225e12 N/m).The formula used to calculate is $F=\rho g h^2$, so, using this formula to compute the ocean force for **an ocean depth d1=3200 m and an ocean depth d2=4200 m**, the result is 0.050e12 N/m and 0.086e12 N/m, respectively, the difference between the two is 0.036e12 N/m, which is the same result computed through the reviewer's method.

**Comment 7. It is already difficult for me to understand how a force of 0.1568e12 N/m could play a significant role against a ridge push of 4e12 N/m (using the author numbers, i.e. against a force 27 times larger), but it seems that the ratio is in fact larger than a factor 100.**

Reply: We understand that Referee 1 is implying that an ocean force of 0.1568e12 N/m is too small to counterbalance a ridge push force of 4e12 N/m. Unfortunately, this is a serious misunderstanding. We never use the ocean-generated force to counteract the ridge push force. This can be witnessed in several parts of the manuscript such as lines 13~14, lines 314~317, lines 326~332, lines 356~358, lines 351~385, line 497~524, and lines 694~721. In this study, we treat the ridge push force as a plate driving force, both the ocean force and ridge push force are combined together to balance the collisional, shearing, and basal friction forces. We feel it is rather necessary to now look over the thinking line of the manuscript. In Section 2.2, we use a model to show that a combination of the ridge push, collisional, and basal friction forces is unable to account for the vertical distribution of observed stresses, implying a need of other force. In Sections 3.1, we present the ocean-generated force. In Section 3.2, we use the ocean-generated force in tandem with the ridge push force to balance the collisional, shearing, and basal friction forces, by which the movements of six plates are determined. In Section 3.3, we use a modeling to find that, given the ridge push force has a magnitude of 4e12 N/m, a combination of the ocean-generated force, ridge push force, collisional force, and shearing force is still unable to account for the vertical distribution of observed stresses. However, reducing the ridge push force to less than the ocean-generated force can satisfy the observation (see lines 618-631 of the manuscript). In Section 4.1 (see lines 694-721), we show that the force balances can be achieved with a combination of these forces. It is important to note that the ridge push force is nearly symmetric around the ridge crest, as shown in the figure below (which is a copy of Figure 12) . The ridge push force $F_{RL}$ pushes Plate A at its right side, at the same time, $F_{RR}$ pushes Plate C, Plate C

pushes Plate B, and Plate B also pushes Plate A at its left side, consequently, $F_{RL}$ is properly balanced out by $F_{RR}$. Therefore, regardless of whether the ridge push force is valued at 4e12 N/m or below, the force balance can be achieved anytime. Moreover, we have mentioned readers in the manuscript (see lines 717~721), " …, even if the ridge push force $F_{RL}$ ($F_{RR}$) is given a smaller amplitude (~ $10^{10}$ N m$^{-1}$, for example), so long as the collisional force $F_{BA}$ ($F_{AB}$, $F_{CB}$, and $F_{BC}$) is properly valued, these force balances can always be realized. Nevertheless, as demonstrated in section 3.3, a ridge push force of $4.0 \times 10^{12}$ N m$^{-1}$ would result in a horizontal stress that is mostly concentrated on the lower part of the lithosphere, which is not in accordance with the observed stress. Hence, we prefer to accept the ridge push force to be smaller than ocean-generated force." This comment "**the ratio is actually greater than 100 times**" is correct but not too meaningful. See the author's response to the comments made by Chuanliang Li earlier.

[Figure]

**Comment 8. I note that the slab traction is generally estimated about 10 times larger than the ridge push (the author mentions a slab traction of about 3.3e13 N/m which is the right amount). Quoting Bercovici et al, (AGU monograph, 2000) "As demonstrated by Forsyth and Uyeda [1975], the correlation between the connectivity of a plate to a slab (i.e., the percent of its perimeter taken by subduction zones) and the plate's velocity argues rather conclusively for the dominance of slab pull as a plate driving force".**

Reply: These comments are somewhat repetitive with those made by Chuanliang Li, which we have addressed earlier. We would like to emphasize a little bit more that, while slab pull is considered the largest, it is solely suitable for oceanic plate such as the Pacific Plate, and not for continental plates like African, Antarctic, Eurasian, Indo-Australian, North American, and South American plates. As seen in the figure below, the directions of movements of plates are various, implying that the motion of each plate is controlled by a dynamic system. Therefore, simply arguing the magnitude of slab pull is not enough. Furthermore, even though there is strong correlation between a plate's velocity and its connectivity to a slab (represented as the percentage of its perimeter taken by subduction zones), it does not guarantee that the slab has a significant contribution to plate motion. In this study, we have successfully reproduced the movements of the

Pacific Plate by using a combination of three collisional forces (as depicted in lines 41~415, Table 2(D), Figure 7(f), lines 548~551, Table 4(B), and Figure 9(c) of our manuscript). This result suggests that the Pacific Plate's movement can realize without slab pull. Actually, the strong correlation between connectivity and velocity could be coincidental. For example, as seen in the figure below, the Eurasian Plate rotates clockwise, the North American Plate rotates counterclockwise, the Indian-Australian Plates moves northeast, and the Pacific Plate moves northwest. The pattern of these movements allows the Pacific Plate to be circled by the Eurasian, North American, and Indian-Australian plates. As a result, the Pacific Plate is forced to subduct, forming very long subduction zones in the boundaries between the Pacific Plate and the three plates. From this point, the slab itself may be a consequence of plate motion.

Last, Referee 1 uses Bercovici et al. (AGU monograph, 2000) to endorse slab pull. However, the long controversy regarding the physics and geometry of this force, which was fully documented by Doglioni and Panza (2015) and is outlined in Section 2.1 of this manuscript, shouldn't be disregarded by Referee 1. Even so, our model does leave room for slab pull, as mentioned in lines 414-416 of the manuscript "Taking into consideration the long argument of slab pull that is listed in section 2.1, we presently neglect slab pull. And if this force can be confirmed in the future, it can be added into this model."

[Figure]

**Comment 9. Yongfeng Yang does not believe in ridge push although, the same halfspace cooling model is used to estimate ridge push and slab pull.**
Reply: Please see the author's response above, this is not true that "**Yongfeng Yang does not believe in ridge push although**". The comment "**the same halfspace cooling model is used to estimate ridge push and slab pull**" may be right but not relevant to this study.

**Comment 10. I do not think the paper is clear, convincing and rigorous enough to be accepted for publication.**
Reply: Along with the response above, we have made significant improvement in the revised

manuscript, which includes changes in the text, figures, and references.

Response to Peter Malin
**Comment 1. I do not think that the underlying premises of this paper are physically valid.**
Reply: Thank Peter Malin for this concern. The comment seems overly conclusive but lacks detail. The reviewer is likely to imply that the ocean-generated force is too small to fight against the existing plate driving forces (i.e., ridge push and slab pull). Thus, we encourage the reviewer to look over the author's responses to the comments made by Chuanliang Li and Referee 1 earlier. Comparing the comments made by three reviewers, we find that they all express suspicion on the ocean-generated force. This may arise from a lack of understanding of plate kinematics. So, what amount of force is necessary for driving the lithospheric plate to move over the underlying asthenosphere?

A plate moving over a fluid (see figure (A) below) is constrained by the basic principle of fluid mechanism (Cengel and Climbala, 2014) as $F = \mu Au/y$, where $F$, $\mu$, A, u, and y are the driving force, the fluid's viscosity, the plate's area, the plate's speed, and the depth of the fluid, respectively. The assumption for this formula is that the dimension of the plate is much greater than the fluid's depth. The plate moves over the fluid, the fluid provides resistance force to counteract the driving force, therefore, it is the force balance that maintains the movement of the plate. This equation means that, for given y, $\mu$, and A, the velocity u is proportional to the force $F$. According to Britannica (https://www.britannica.com/science/lithosphere), the lithosphere consists of the crust and the upper mantle, and extends to a depth of about 100 km. Also according to Britannica (https://www.britannica.com/science/asthenosphere), the asthenosphere is a zone of Earth's mantle lying beneath the lithosphere and believed to be more fluid than the lithosphere, and it extends from about 100 km to about 700 km below Earth's surface. These features allow us to apply the equation above to constrain the movement of the lithosphere over the asthenosphere (see figure (B) below), where $F$, $\mu$, A, u, and y now denote the driving force, the asthenosphere's viscosity, the lithosphere's area, the lithosphere's speed, and the asthenosphere's depth, respectively. The lithosphere's area and depth have been well established, the speed of the lithospheric plates is measured to be about a few centimeters per year. However, there is a large uncertainty regarding the asthenosphere's viscosity. According to the experiments and theoretical models from various authors (see lines 447-466 of the original manuscript), the viscosity spans a broad range from $10^{15}$ to $10^{20}$ Pas. Given y=300 km, A= 510,000,000 km$^2$, u= 3 cm/yr, and $\mu = 10^{15} \sim 10^{20}$ Pas, then, $F = \mu Au/y = 1.6172 \times 10^{15} \sim 1.6172 \times 10^{20}$ N. This result indicates that, for a lithosphere's movement of 3 cm per year, it would require a driving force of $10^{15} \sim 10^{20}$ N to fit, and that the driving force is equal to the resistive force exerted by the asthenosphere, which forms force balance to maintain that movement. And now, we consider the lithosphere as individual plates, and apply a driving force to one of these plates. The resistance force undergone by this plate comes from adjacent plates and from underlying asthenosphere. Since all the plates are attached to the underlying asthenosphere, so, the upper limitation of the resistance force undergone by this plate will be $10^{15} \sim 10^{20}$ N, which is estimated above. Therefore, if a driving force has the magnitude that falls into the range of the upper limitation of the resistance force, the force balance may be formed to maintain the movement of a plate. As exhibited in Table 1(A, B, C, and D) of the manuscript, the horizontal force $F_i$ (i.e., the ocean-generated force) generally has a magnitude of $10^{17}$ N, which has fallen into the range of the upper limitation of the resistance

force. This indicates that the ocean-generated force is capable of driving the lithospheric plate to move.

[Figure]

A                                                    B

Taking into account this present status, we add a paragraph as below in Section 4.1 of the revised manuscript to address the plate's kinematics.

"Some people are puzzled as to how ocean-generated force can drive the lithospheric plates to move over the asthenosphere. This confusion stems from a lack of understanding of plate kinematics. So, what amount of force is actually necessary to maintain the movement of a lithospheric plate? According to the principle of fluid mechanics (Çengel and Climbala, 2014), a plate moving over a fluid is constrained by the equation $F = \mu Au/y$, where $F$, $\mu$, A, u, and y represent the driving force, the fluid's viscosity, the plate's area, the plate's speed, and the depth of the fluid, respectively. This equation requires that the size of the plate is much greater than the fluid's depth and the fluid provides a resistance force, which balances the driving force to maintain the plate's motion. The lithosphere consists of the crust and the upper mantle and extends about 100 km below Earth's surface. Beneath the lithosphere is the more fluid asthenosphere, which extends from roughly 100 km to 700 km below Earth's surface. These features allow us to apply the equation above to constrain the movement of the lithosphere plate over the asthenosphere, where $F$, $\mu$, A, u, and y now denote the driving force, the asthenosphere's viscosity, the lithosphere's area, the lithosphere's speed, and the asthenosphere's depth, respectively. While the lithosphere's area and depth, and the speed of plates have been well established, there remains a high uncertainty concerning the viscosity of the asthenosphere, which varies widely from $10^{15}$ to $10^{20}$ Pas based on experimental and theoretical models (see a description of this issue in Section 3.2). Given y = 300 km, A = 510,000,000 km$^2$, u = 3 cm/yr, and $\mu = 10^{15} \sim 10^{20}$ Pas, the driving force estimated through the equation ranges from $1.6172 \times 10^{15}$ to $1.6172 \times 10^{20}$ N. This result suggests that in order for a lithosphere to move 3 cm per year, a driving force of $10^{15} \sim 10^{20}$ N is necessary, and this force is countered by the resistive force exerted by the asthenosphere, creating a force balance that enables the movement to be sustained. And now we consider the lithosphere as individual plates, and apply a driving force to one of these plates, the plate encounters resistance from adjacent plates and the underlying asthenosphere. Since all plates are attached to the underlying asthenosphere, the upper limit of the resistance force that the plate can undergo will be $10^{15} \sim 10^{20}$ N, which is estimated above. Thus, for force balance to be achieved and for the lithospheric plate to move, a driving force must fall into the range of the upper limit of the resistance force. Table 1(A, B, C, and D) shows that the ocean-generated force (i.e., the horizontal

force $F_i$) is generally at a magnitude of $10^{17}$ N, which just falls within the range of the upper limit of the resistance force. This indicates that the ocean-generated force is capable of driving the lithospheric plate to move."

**Comment 2. I do not think that the forces claimed to explain plate motion are accounted for in manner that makes physical sense. For example the edges of plates etc are not represented with geologically accurate structures: e.g. Plate boundary faults and continental-ocean interfaces.**

Reply: These comments deserve author's attention. Firstly, having a precise understanding of the continental-ocean interfaces is crucial in calculating the ocean-generated force. As demonstrated in Section 3.1 (see lines 242-247 of the manuscript), we have made assumptions for the continent: "In practice, the continent's side is not flat, and the continent's base is generally wider than its top, making the continent appear more like a circular truncated cone standing in the ocean. As the horizontal force is related to the ocean's width (i.e., the continent side's width), we need to horizontally project the continent onto a polygonal column, dissect the whole side of this column into a series of smaller rectangular sides connecting one to another and subsequently calculate the horizontal force generated at each of these rectangular sides." The horizontal force is expressed as $F = 0.5\rho gLh^2$, which relates to the density of water, gravitational acceleration, ocean width, and ocean depth. These factors totally determine the ocean-generated forces around a continent and rely heavily on the accuracy of the polygonal column, which is projected from a real continent. In general, the greater the number of smaller rectangular sides dissected from the whole side of the column, the more precise the calculation of the column's whole side, and the more precise the estimation of ocean depth that corresponds to a rectangular side. In this study, we utilize the geographical locations of the sites controlling column dimensions presented in Figure 4 to compute the lengths of smaller rectangular sides. Additionally, we estimate the corresponding ocean depths through the use of the NOAA bathymetric data viewer. Despite the absence of external funding, we acknowledge that this approach may not yield a comprehensive solution. Nonetheless, we are confident that the model and assumptions presented herein serve as a useful foundation for future research on this topic.

On the flip side, a precise knowledge of plate boundary is crucial for understanding the dynamic interactions between plates. We have considered collisional and shearing forces (e.g., $F_{AF-EU-C}$, $F_{IN-EU-C}$, $F_{AU-EU-C}$) in Section 3.2 (lines 409-417 of the manuscript) and illustrated them in Figure 4 and Table 1(D). These forces relate to plate boundaries, but their magnitudes and directions are artificially assigned, which may lead to discrepancies when compared to reality. For accurate determination of these forces, a detailed investigation of the structures at plate boundary is necessary. Unfortunately, our present sources cannot support this task, but we would like to remind readers as below in the revised manuscript.

"It is worth noting that the magnitude and direction of the collisional and shearing forces are artificially assigned, potentially causing discrepancies with reality. A precise determination of these forces requires in-depth investigation of the structures at plate boundary, but is not feasible within the scope of this study due to individual effort constraints."

**Comment 3. Moreover I do not think the models and their physical and structural characteristics are representative of the Earth's upper mantle, asthenosphere, and**

**lithosphere.**

Reply: The reviewer's lack of belief in the author's models and their characteristics, without providing any supporting details or evidence, presents a challenge in accurately addressing his concerns. However, we would like to respectfully counter this viewpoint. The current understanding of the Earth's structure indicates that the lithosphere is comprised of plates steadily moving over the underlying asthenosphere, and the Earth's layers can be divided into lithosphere (encompassing the crust and upper mantle), asthenosphere, lower mantle, and core. The author's models and corresponding physical characteristics are largely depicted in Figures 5 and 12 of the manuscript, which align with this current understanding. Certainly, it is possible that we overlooked some detail. Therefore, any clarification from the reviewer regarding weaknesses in the author's models would be greatly appreciated.

**Comment 4.The ms leaves out references to the most recent and well supported studies of the ocean crust, lithosphere, and asthenosphere.  (E.G. for a completely different and physically and geologically acceptable alternative see: Morgan, J. P., Jörg Hasenclever, and C. Shi. "New observational and experimental evidence for a plume-fed asthenosphere boundary layer in mantle convection." Earth and Planetary Science Letters 366 (2013): 99-111.)**

Reply: These comments point out that the recent works haven't been noted by the author. Sorry, we are also difficult to accurately address this concern. Many studies of geophysics that involve crust, lithosphere, and asthenosphere are published day after day, we have examined recent works but cannot find ones that closely relate to this study. Certainly, it is possible that we have missed some works. Therefore, any clarification from the reviewer regarding this issue would be greatly appreciated.

However, the reviewer's proposed work (Morgan et al., 2013) appears to be irrelevant to this present study. These authors proposed a hypothesis of PFA boundary layer, which is not yet confirmed so far. This hypothesis is also based on another hypothesis of mantle plumes that remains highly controversial among geophysicists (Koppers et al., 2021). Our standard to cite a work is that it must closely relate to the topics included in this study. As indicated in the title of the manuscript, which is "What Drives Plate Motion?", we aim at exploring a plate driving mechanism. We ask ourselves a few questions before we intend to discuss the work (Morgan, et al., 2013) in this study: are mantle plumes a real plate driving force? does mantle plumes contribute to the existing force balance? What's a coupling between vertical upwelling mantle plumes and horizontal moving plates? And how does the PFA boundary layer contribute to force balance and plate motion? …The answers for these questions are presently not clear.

We would like to recommend several works (e.g., Forte, 2020; Bercovici et al., 2015; Coltice et al., 2017) for the reviewer to examine. These authors gave a detailed description of the plate driving forces but never mentioned the work by Morgan et al. (2013).

Koppers, A. A. P., et al.: Mantle plumes and their role in Earth processes, Nature Reviews Earth & Environment, 2, 382-401, 2021.

Forte, A. M.: Plate Driving Forces, Encyclopedia of Geophysics, 2$^{nd}$, edited by H.K. Cupta, 2020.

Bercovici, D., Tackeley, P. J., and Ricard, Y. : The generation of plate tectonics from mantle dynamics: Reference Module in Earth Systems and Environmental Science, Treatise on Geophysics (Second Edition), 7, 271-318, 2015.

Coltice, N., Gerault, M., and Ulvrova, M.: A mantle convection perspective on global tectonics, Earth-Science Reviews, 165, 120-150, 2017.

**Comment 5. The length, structure and writing style of this paper ends up hiding the demonstration promised in the abstract.**
Reply: This comment overlaps some of the comments made by Referee 1, suggesting a significant revision of the manuscript's structure. We accept this comment and have made significant improvement on the structure and writing of the manuscript.